

# Review article: Feature tracing in radio-echo sounding products of terrestrial ice sheets and planetary bodies

Hameed Moqadam[1,2] and Olaf Eisen[1,3]

[1]Glaciology, Alfred Wegener Institute Helmholtz Centre for Polar and Marine Research, Bremerhaven, Germany

[2]Constructor University, Bremen, Germany

[3]Department of Geosciences, University of Bremen, Bremen, Germany

**Correspondence:** Hameed Moqadam (hameed.moqadam@awi.de)

**Abstract.** Radio-echo sounding (RES) is a useful technique for measuring the subsurface properties of ice sheets and glaciers. One of the most important and unique outcomes is the mapping of ice sheets' englacial layer stratigraphy, mainly consisting of isochronous reflection horizons. Mapping those is still a labour-intensive task. This review provides an overview of state-of-the art (semi-)automated methods for identifying ice surface, basal, and internal reflection horizons from radargrams in radioglaciology. We discuss a variety of methods which were developed or applied to RES data over the last decades, including image processing, statistical techniques, and deep learning approaches. For each approach, we briefly summarize their procedures, challenges, and potential applications. Despite major advances, we conclude that gaps remain in effectively mapping internal reflection horizons in an automated way, but with deep learning representing a potential advancement. This paper aims to inform researchers and practitioners in radioglaciology about current and future trends in mapping the englacial stratigraphy of ice sheets.

## 1 Introduction

Radio-echo sounding (RES) is a powerful technique which has been used in radioglaciology for more than 50 years to investigate subsurface properties of polar ice sheets (Schroeder et al., 2020). It has proven useful for determining widespread basal topography and ice thickness on glaciers as well as in inaccessible regions such as the Antarctica and Greenland Ice Sheets. However, RES data not only reveal information about the base of the sheet and ice thickness, but also provide insights into their internal structure. Such insights are obtained from the presence of englacial reflections and backscatter characteristics in RES data, most prominently internal reflection horizons (IRH), also known as internal radar reflections (Schlegel et al., 2022). These IRHs are a result of variations in the dielectric properties of the ice, which can be attributed to changes in density, impurity content, acidity, or crystal orientation fabric (Moore and Paren, 1987; Eisen et al., 2007).

It has been shown that IRHs, caused by changes in conductivity, are generally isochronous, i.e. one horizon has the same age everywhere (Fujita et al., 1999; Eisen et al., 2006), serving as indicators for paleoglaciology (Siegert, 1999; Fahnestock et al., 2001; Miners, 2002; Jansen et al., 2024). Englacial horizons observed in RES datasets have also been utilized to investigate ice dynamics, calibrate ice–flow models, estimate past accumulation rates, and constrain layer ages from ice cores (Siegert et al., 2004; Rippin et al., 2006; Conway et al., 1999; Waddington et al., 2007; Schroeder et al., 2020; Sutter et al., 2021).





Geometry of isochronal radar reflection horizons, in conjunction with ice–flow modeling, can provide significant perspectives into ice dynamics, basal sliding, surface accumulation history, and englacial folding (Waddington et al., 2007; Nereson and Raymond, 2001; Hindmarsh et al., 2009; Catania and Neumann, 2010; Leysinger Vieli et al., 2011; Lenaerts et al., 2014; Jenkins et al., 2016; Holschuh et al., 2017; Born and Robinson, 2021; Bons et al., 2016; Sutter et al., 2021; Jouvet et al., 2020; Jansen et al., 2016). Additionally, stratigraphic information provided by englacial layers complement ice-core analyses,

improving interpretation of climate changes recorded in ice cores by revealing flow paths and irregularities that may affect age stratigraphy at ice-core sites (Fahnestock et al., 2001; NEEM community members, 2013; Parrenin, 2004). To support joint international and collaborative exploitation of the available radar data sets, the Scientific Committee on Antarctic Research (SCAR) has endorsed the AntArchitecture Action Group, specifically dedicated to cataloging IRHs across the entire Antarctic ice sheet (Bingham et al., 2019).

One of the earliest publications on internal reflections by Bailey et al. (1964) observe a continuous echo at the depth of 500 meters as well as a 97% continuous basal layer after a series of measurement campaigns in Greenland. They noted that compacted annual accumulation is the cause of such echoes (reflections). Moreover, other early works such as works of Gud-mandsen (1975) and Robin (1975) discuss exclusively RES measurements over ice sheets and their interpretations (Paren and Robin, 1975; Clough, 1977). IRH are traditionally identified by manually or semi-automatically tracing individual reflections

within RES datasets, a laborious and time-consuming process (Nereson et al., 2000; Waddington et al., 2007). It has been shown that tracing 20 IRHs in 20,000 km of data in such a way would take 10 operator-years to complete (Sime et al., 2011). To overcome the slow way of manual tracing, already since the 1980s, some commercial softwares have been used for semi-automated mapping IRHs. Some others were complemented by open-source software modules provided by the community, in addition to processing and analyzing RES data. Some examples include software packages such as MATLAB (MathWorks,

2022), GSSI Radan (GSSI), ReflexW (Sandmeier, 2016), Sensors & Software EKKO Project (Sensors Software Inc) and open-source software packages such as ImpDAR (Lilien et al., 2020) library for Python and RGPR package (Huber and Hans, 2018) for R.

Obtained age stratigraphy of the Antarctic ice sheet, unlike the Greenland ice sheet (MacGregor et al., 2015), has been limited to specific regions (MacGregor et al., 2015; Siegert et al., 1998; Eisen et al., 2004; Siegert et al., 2004; Steinhage et al.,

2001; Leysinger Vieli et al., 2011; Cavitte et al., 2016; Winter et al., 2019), resulting in an incomplete picture of its englacial architecture. Several challenges slow down the achievement of a continent-wide stratigraphy. The considerable time required for tracing IRHs, limited spatial coverage of available data, and a lack of integration between stratigraphic information from different RES systems (Cavitte et al., 2016; Winter et al., 2017) are among the reasons. However, the primary challenge remains to be the imbalance between the amount of available data and the amount of time required with available methods to map the

stratigraphy. In terms of size, the Antarctic ice sheet surpasses the Greenland ice sheet by more than sixfold. Consequently, there exists a significantly larger volume of unexplored data from Antarctica compared to that of Greenland.

This limited advancement of methodologies for assessing the structural configuration of the stratigraphy across large spatial scales challenges exploration of englacial architecture in Antarctica (Delf et al., 2020). To overcome the difficulties associated with manual picking of IRHs, there has been a growing interest in developing (semi–)automatic methods for tracing IRHs in





RES echograms (also known as "radargrams"), in particular from airborne operations. The motivation behind these efforts is to reduce the amount of human labour required for data analysis, particularly as radar datasets have expanded over large spatial scales (Medley et al., 2014; MacGregor et al., 2015; Cavitte et al., 2016; Koenig et al., 2016; Bingham et al., 2019; Delf et al., 2020), as well as to reduce subjectivity of interpretations of IRHs (Dossi et al., 2015). Automated horizon-picking techniques have shown some potential, but they still require some operator input, and are yet to effectively map IRHs.

In the past two decades, there have been various research attempts on automatically tracing ice–bed boundary, mapping reflections, tracing firn layer boundaries and segmenting regions of radargrams from both ice sheets and planetary radargrams by several research groups. Yet, a complete account of this long-lasting endeavor which contains a comprehensive overview of all the methods—and regions these methods were applied to—has been missing.

In this review paper, we present an overview of the available methods for tracing layer boundaries and IRHs in radargrams.
By presenting various studies and approaches, we aim to provide insights into the advancements, challenges, and future directions. In section 2, we briefly discuss the RES technology, and terminology that is necessary for understanding radar products. Section 3 introduces the methods that have been employed by various research groups in a timeline of method development for the task of stratigraphy mapping. A comprehensive timeline of the published works with a short summary of each publication, remarking the more relevant information of each of the publications, is discussed in section 4. Finally, we provide discussion,
conclusion and outlook in sections 5 and 6, respectively, highlighting the need for automatic methods to fully exploit the extensive datasets and labor-intensive nature of manual picking and analyzing the recent trends with potential directions of future research.

## 2 Background

In this section, we provide necessary concepts and information related to RES. We start with introducing radioglaciology and
go on to describe radargrams and IRH representations. For further details on radar physics and applications, we refer the reader to available radar literature Bogorodsky et al. (1985); Plewes and Hubbard (2001b); Dowdeswell et al. (2008); Bingham and Siegert (2007); Pellikka and Rees (2010); Woodward and Burke (2007).

### 2.1 Radioglaciology

Radioglaciology is the scientific field that employs radar (*ra*dio *d*etection *a*nd *r*anging) systems to explore the cryosphere. RES
is an active remote sensing method which, unlike satellite imagery, could give a picture of the cross section of ice sheet. An electromagnetic waveform is emitted from a transmitter antenna, penetrates the ice and is reflected by changes in the complex-valued permittivity of ice. The reflection travels back to a receiving antenna. Reflective properties are influenced by various factors such density (presence of bubbles), orientation of ice crystal, inhomogeneities, impurities and geometry of the materials. Applications range from determining ice thickness, identifying englacial and subglacial properties, e.g. lakes, reconstructing
past ice-dynamic changes and extrapolating ice-core records. Such studies employed airborne, ground-based or orbital RES systems on terrestrial and planetary ice bodies. In the following, we will give a brief account on RES physics and applications,



but refer the reader to available publications for further details, e.g. Bogorodsky et al. (1985); Plewes and Hubbard (2001b); Dowdeswell et al. (2008); Bingham and Siegert (2007); Pellikka and Rees (2010); Woodward and Burke (2007).

For our objective in this review, the important information derived from radargrams is the englacial layer architecture. Such layer boundaries, called internal reflection horizons (IRH), were formed at the former ice sheet surface, then advected into the ice by additional accumulation and deformed by ice flow. These IRHs are thus isochronous (Gudmandsen, 1975; Siegert, 1999), i.e., each has the same age everywhere. IRHs primarily originate from density fluctuations in the upper part and variations in dielectric conductivity (e.g., from acidity, MacGregor et al. (2012)) in deeper regions of the ice sheet. In the medium to deepest layers of the ice sheet, changes in the crystal orientation fabric can also result in reflections (Fujita et al., 1999; Eisen et al., 2003).

## 2.2 Radar products

In radioglaciology applications, out of every single survey line, a 2D cross-sectional profile of the ice sheet is produced. This product is called a *radargram* or an *echogram*. In older texts, similar profiles were called Z-scope. A radargram depicts a full profile of the cross section of the ice sheet as opposed to single traces. It is usually composed of single transmit signals and reflections. In the case of single-point measurements, they are stored as amplitude displays which are also called A-scope, similar to panel (a) and (b) of Fig.1. The aforementioned characteristics such as presence of impurities, acids, and changes in ice-crystal orientation cause reflections, and when they are laterally coherent, they appear as horizons. Every pixel within the radargram corresponds to the quantification of amplitude (or power) associated with the radar wave that is reflected by subsurface interfaces positioned at a designated range (two-way travel-time or depth) location and a spatial coordinate within the azimuthal direction.

Fig. 1 depicts different representations of a trace and a vertical section of the same profile. Panels (a) and (b) represent the an arbitrary trace with 60-ns and 600-ns pulse respectively. Panels (c) and (d) show a section of radargram with the leftmost trace shown in panels (a) and (b), and panels (e) and (f) show same radargram sections composed of differentiates traces. In most cases for older systems, where phase was lost because of rectification of the received signal, studies are done using the differentiated radargrams as they illustrate a clearer picture of the englacial architecture. In this figure, ice surface (air–ice interface), base (ice–base interface), englacial reflection and the so-called echo-free zone (EFZ, just above the bed) can be seen. The EFZ in the conventional sense was affected by different factors, e.g. system sensitivity or lack of coherent reflections owing to disturbances possibly from ice flow near the interface of ice and base (Drews et al., 2009).

Individual measurements are often noisy, typically due to the electromagnetic interference from other electronics, such as aircraft and other components in the vicinity of the instrument, as well as thermal noise. Therefore, radar traces are usually stacked to increase the signal-to-noise ratio and obtain enhanced subsurface images (Karlsson et al., 2012). In the presented Fig. 1, each plotted trace is a stack of ten consecutive traces.




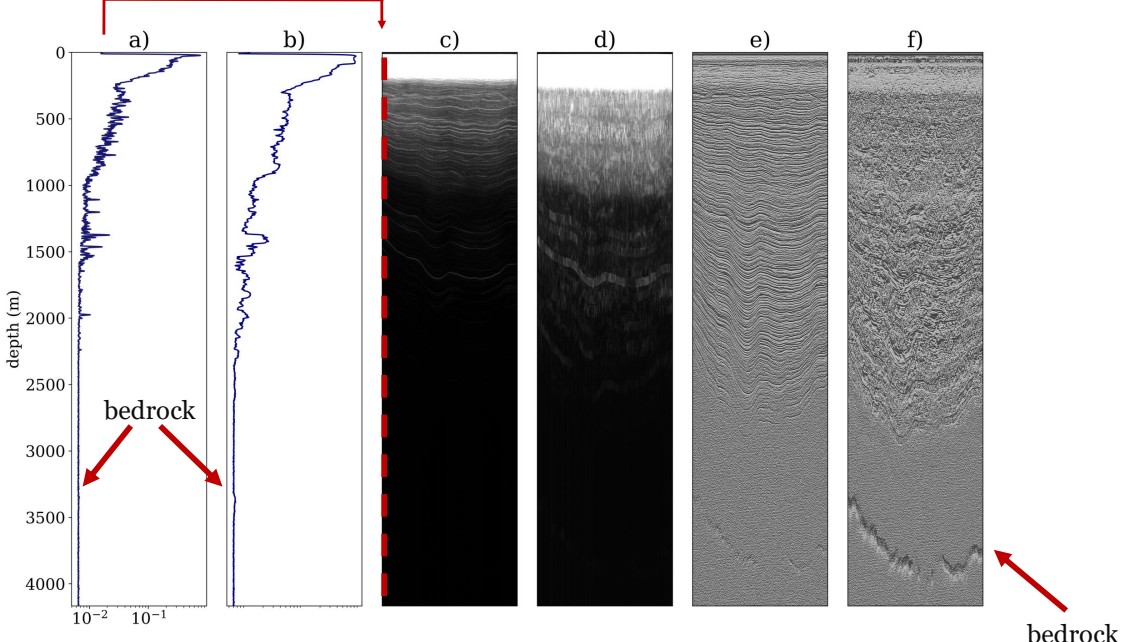

**Figure 1.** An example of a vertical section of a radargram and a single trace from it. The section is from a flight performed in 1999 between Dome Fuji and Kohnen station (Steinhage et al., 2013): (a) trace (A-scope) with 60-ns pulse; (b) trace (A-scope) with 600-ns pulse; (c) vertical section of raw radargram (Z-scope) with 60-ns pulse; (d) vertical section of raw radargram (Z-scope) with 600-ns pulses (e) vertical section of differentiated radargram (Z-scope) of panel c); (f) vertical section of differentiated radargram (Z-scope) of panel d).

## 2.3 Internal reflection horizons

In the radargram of Fig. 1, reflection signatures can be seen in different regions such as close to the surface, englacially, and
subglacially. The most general term to refer to any signal in the data which is not noise is *event*. Such events are illustrated in Fig. 2, which is a simplified a schematic of a radargram, where differences between ice layers and IRHs are depicted. The ice surface at the top (blue line) and basal reflection (black line) at the bottom of the ice are also shown. The first reflection of each transmitted pulse of an airborne survey is the reflection from the ice surface.

For the sake of facilitating analyses of radargrams, one of the common practices is to synchronize all traces form the ice-air
interface at time zero, omitting topographical variations. This flat ice surface is naturally appearing in ground-based systems, however, for airborne systems this assigning surface time to zero is a step during data processing. The black line depicts the basal reflection. The red lines in the radargram indicate IRHs. In an ice sheet, these represent the interfaces between the neighbouring ice layers of different properties, such as layers with different density or crystal orientation fabric, or they can be caused by thin individual horizons with higher conductivity, thus forming IRHs. As mentioned before, it has been shown that
IRHs are isochrones (Gudmandsen, 1975; Siegert, 1999; Fahnestock et al., 2001; Miners, 2002).





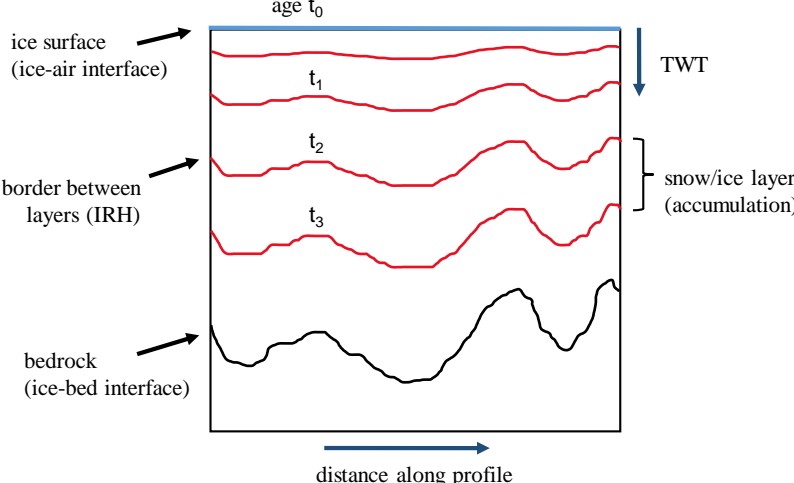

**Figure 2.** Schematic of a radargram. The blue line at the top represents the surface of the ice sheet (ice-air boundary). It is conventionally set as time zero discarding topography. The red lines are IRHs which represent the changes in the permittivity that could be present on the boundaries between different layers. The black line at the bottom is a representation of the ice base. The x-axis is the distance in the direction of the flight and the y-axis can be shown as either two-way travel-time (TWT) or depth.

## 2.4 Applications

Englacial stratigraphies deduced from RES data are more and more used to benchmark and validate models of ice dynamics (Sutter et al., 2021; Björnsson and Pálsson, 2020). A number of englacial features can be visible in radargrams, which can be studied in both a quantitative and a qualitative manner (Plewes and Hubbard, 2001a; Pellikka and Rees, 2010). Quantitative
studies take advantage of amplitude and phase of traces and are often used to derive physical properties of ice (Plewes and Hubbard, 2001b). Qualitative studies, in contrast, mostly utilize stratigraphy to infer current of past flow dynamics or boundary conditions, e.g. surface accumulation (Arcone et al., 2005) or basal melting (Bogorodsky et al., 1985). Some of the many applications of englacial stratigraphy are to study past ice stream dynamics (Keisling et al., 2014; Winter et al., 2015; Jansen et al., 2024; Carter et al., 2023), glacier-volcano interactions (Björnsson and Einarsson, 1990), meltwater drainage (Pitcher
et al., 2020), glacier hydrology and dynamics (Eisen et al., 2020), glacier response to climate shifts (Guðmundsson et al., 2009), mass balance (Kowalewski et al., 2021), glacier evolution (Aðalgeirsdóttir et al., 2011), and volcanic activities (Brandt et al., 2005b). RES is also used to identify subglacial properties, such as lakes (Bowling et al., 2019), which appear as strong and rather flat features at the bottom of the ice, owing to the high permittivity of liquid water in contrast with the overlaying ice.
For a variety of applications such as developing compilations of bedrock topography (Lythe and Vaughan, 2001; Frémand et al., 2023), synchronizing ice cores (Steinhage et al., 2013; Cavitte et al., 2016), paleoglaciological studies (Parrenin et al., 2017), ice dynamics (Jansen et al., 2024), mass balance derivation (Brandt et al., 2005a), and ice sheet modelling (Sutter et al.,




2021), the key is to have a mapped englacial stratigraphy or mapped basal surface. In the next section, we look into the most common methods that have been used to map the englacial stratigraphy.

## 3 Overview of methods

In this section, we provide a brief overview of the methods that have been applied to tracing IRH and segmenting radargrams. The subsections related to each method provide information on how it has been used for this task. We present more details on implementation in the timeline of publications in section 4.

Given the versatile application of RES across various domains, efforts to characterize features within radargrams or to map reflections have proved valuable across various fields, including contamination assessment, hydrology, archaeology, geotechnical engineering, and glaciology (Jol, 2009). Mapping englacial stratigraphy contributed to a broad range of glaciological studies. Many studies used manually tracing of horizons, but only some applied semi-automatic tracing of horizons, e.g. for investigating ice dynamics (Fahnestock et al., 2001) and subglacial lakes (Humbert et al., 2018), age-stratigraphy (Fahnestock et al., 2001; Winter et al., 2019), bed mapping and topography (Lee et al., 2014; Franke et al., 2020), synchronizing ice cores (Steinhage et al., 2013) and snow accumulation (Medley et al., 2013; Freeman et al., 2010).

Constructing an automated tracing method for RES encounters a significant challenge when dealing with closely spaced layers. This situation gives rise to numerous horizon candidates that are nearly identical but slightly offset from each other. If the algorithm mistakenly selects the wrong candidate, it may veer into adjacent horizons, leading to inaccurate tracing (Panton, 2014). This situation is more relevant when regarding deep IRHs. The IRHs in snow and firn radargrams have much less compaction as well as vertical fluctuation. This is the foremost reason for automatically identifying and differentiating deep englacial horizons to be much more challenging than near surface and basal reflections.

The methods to map the near-surface, basal or englacial architecture of the ice could be categorized on the basis of a variety of criteria. One such criterion is if a method operates semi-automatically or fully automatically. By semi-automatic, we refer to methods that require manual tweaking, interference or initialization by a user. Another category is if the proposed method does or does not include machine learning algorithms. It is also possible to categorize methods based on the depth or specific reflection that they are designed for. Some methods (mostly earlier ones) are only aimed at tracing surface and basal reflections in order to estimate ice thickness, others look into englacial events.

The complexity of tracing englacial layers is caused by:

- limitation of resolution,

- large signal-to-noise ratio,

- lack of discrete boundaries between layers,

- complex englacial structures.



We will give a short summary of the methods that have been utilized in mapping and segmenting radargrams. The provided method summaries are intended to aid understanding of the timeline of methodologies in section 4, to make readers more aware

of the underlying components or procedures of each method.

### 3.1 Cross-correlation and peak-following

Cross-correlation identifies similarities between two signals. Peak-following typically refers to a control strategy used in systems where one variable is controlled to follow the peaks or high points of another (Fahnestock et al., 2001). Stratigraphy mapping, cross-correlation and peak-following enforce and complement each other in a manner that first a peak is calculated

within a certain vertical window, which is the strongest return in the case of radargrams. Next, the cross-correlation is used to find a similar pattern in the radargram. Depending on the backscatter characteristics and spatial coherence, each method performs more efficiently in different areas of a single radargram (Fahnestock et al., 2001). This method has its roots in seismic applications, which often have been used for data processing and analysis in glaciology (Eisen et al., 2004, 2006).

### 3.2 Edge detection and thresholding

An edge in an image is considered to be the location of abrupt change in pixel intensity. One of the most prominent filters used in edge detection is the Canny operator (Canny, 1986). It is a special filter kernel that is convolved with the image and smoothes the image to remove some noise and simultaneously calculates the gradient of the image to determine locations with high spatial derivatives. The next step is to follow along the gradient and suppress pixels that are not maxima, a process called non-maxima suppression. Lastly, it is necessary to apply thresholding and remove weak edge pixels. Having been in

use for more than three decades, the Canny edge detector is still widely-used and efficient in detecting edges in a number of applications, e.g. to capture sharp breaks or discontinuities in an image (Canny, 1986). Based on these properties it was expected to be an efficient method in tracing englacial horizons, and has been applied to near surface reflections (Freeman et al., 2010; Ilisei and Bruzzone, 2014). However, it has been concluded that this detector works well only for the detection of surfaces due to presence of noise in radar and closeness and weakness of horizon boundaries (Mitchell et al., 2013a).

### 205 3.3 Active contour: Snake

A well-known computer vision method of active contours is the Snake (Kass et al., 1988). It consists of splines that are forced by external constrains and influenced by pixel intensity. In the context of active contours, a spline is a mathematical curve that is used to represent the contour or shape of an object or region of interest in an image. From there, two constraints are to be satisfied. One is for the spline to align with the high-gradient energy pixels and the other is avoidance of having discontinuities.

An energy function is defined, and the cost of the first spline is calculated. Then the energy function is minimized to find the most optimum location, in relation with the two constraints (Kass et al., 1988). In radargram applications, an active contour comprising a single particle per column is initially positioned at the uppermost portion of the radargram and subsequently sinks down until it reaches the designated horizon. The contour attains a convergence of optimization through the interplay of three





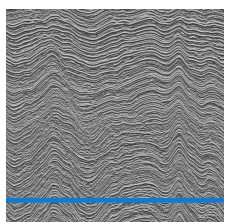 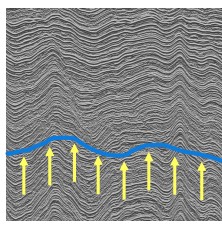 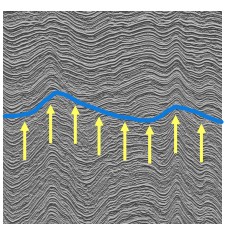 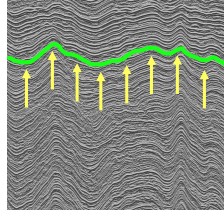

**Figure 3.** Application of a Snake or active contour method; the series from left to right show the evolution of the initial contour until it reaches the boundary.

distinct "forces": 1) a gravity-like force exerted to propel the contour in a downward direction, 2) an upward force influenced
by image edges, akin to buoyancy, and 3) a tension force operating between adjacent particles (Reid et al., 2010; Gifford et al., 2010). Active contour models have the advantage that they do not require radargrams with manually traced IRHs. The main disadvantage is that a Snake model is not able to maintain the complex topology of the evolving curve (Rahnemoonfar et al., 2017a). In terms of automatization level, on the grounds of the initial seeding and curve placement, they are mostly considered as semi-automatic methods. Fig. 3 depicts the stages of the active contour, from the initial contour to the final one reaching the
edge boundary. The arrays in the figure are some of the forces applied to the contour at each stage.

## 3.4 Active contour: Level Set Function

This approach uses Level Set Functions (LSF) and presents a significant advancement in boundary delineation and image contours (Osher and Sethian, 1988; Malladi et al., 1995). It is a scalar field that signifies the signed distance to the nearest edge or boundary (cfd, 2019). Distinguished from conventional Snake active contour model, the level set framework can work
as well with no explicit initial contour parameterization (Lin et al., 2004), making it well-suited for the intricate analysis of radargrams. Optimising a level set involves creating an energy or cost function, which governs the iterative minimization of the function to detect the object boundaries, using image attributes such as gradients and curvatures (Chan and Vese, 2001). The evolution of the initial curve is determined by a speed function, which in turn depends on factors such as image gradient, and involves a halting criteria which reduces the speed function to zero on high gradients delineating boundaries (Lin et al., 2004).
The Level Set method has also proven efficient in other domains such as semi-automatic image segmentation for medical imagery (Lin et al., 2004; Chunming Li et al., 2011).

## 3.5 Statistical Analysis

This method has been employed mostly for characterization of subsurface target classes (Ferro and Bruzzone, 2012; Ilisei and Bruzzone, 2014, 2015). Its backbone is statistical analysis of distribution of the radar signals. This is obtained by fitting several
probability distribution functions (pdf) to the histogram of samples from each target class in the radargram. The pdfs used to fit the signals are parametric models such as Rayleigh and Nakagami distributions (Ferro and Bruzzone, 2012; Ilisei and



Bruzzone, 2014, 2015). The choice of such parametric models for the fits result from their proven capability to model radar amplitude fluctuations of signal backscatter (Oliver and Quegan, 2004).

## 3.6 Layer Slope Inference

What we call here layer slope inference is in fact a combination of methods to calculate the dip angles of the horizons. It consists of: denoising using averaging techniques, thresholding to obtain the binary image from a radargram, discretizing the data horizontally to detect short segments of boundaries, eliminating the invalid objects, and finally compiling the non-uniformly distributed information on object dip (Sime et al., 2011). This method, although being robust and easy to implement, does not map IRHs. Instead, it yield estimates of the potential layer boundaries and their dips and slopes (Sime et al., 2011;
Holschuh et al., 2017).

## 3.7 Hough and Radon transforms

Hough (Hough, 1962) and Radon transforms (Radon, 1917) are very closely related to each other (van Ginkel and van Vliet, 2004). Radon (1917) introduced a method to express a function on the basis of its (integral) projections, and Radon transform is mapping of this function onto its projection. As it maps from image space to parameter space, the function that is formed in the parameter space includes peaks which correspond to shapes or edges in the image space (van Ginkel and van Vliet, 2004; Radon, 1986; Epstein, 2007). The Hough transform is similarly mapping from image space to parameter space. It was originally developed to detect straight lines in black and white images (Hough, 1962). An accumulator array is set up, with each of its elements representing the number of votes that indicate the presence of a shape or edge with corresponding parameters of that element, signifying strong evidence for the existence of that line or edge (Duda and Hart, 1972; Bailey et al., 2020).

## 3.8 Continuous Wavelet Transform

Unlike traditional methods such as gradient-based edge detection (e.g. Sobel (Sobel and Feldman, 2015), Roberts (Roberts, 1963)), which rely on discrete derivatives, continuous wavelet transform (CWT) operates by analyzing the image at multiple scales and positions simultaneously (Mallat and Hwang, 1992). Considering all the values of the translation and scale parameters is the point where CWT differs from discrete wavelet transform, making it a preferred method for detecting specific
features in images (Antoine et al., 1993). Mallat and Hwang (1992) established edge detection in a multi-scale method using wavelet transform. Locating an edge involves initially identifying the scale where the power spectrum, derived from the wavelet transform, reaches its peak. At this scale, the position of the peak in the squared CWT can be identified. CWT's advantages include multi-scale analysis for edge detection at various levels of detail and handling non-stationary signals, making it effective for complex image analysis (Mallat and Hwang, 1992; Kaspersen et al., 2001; Heric and Zazula, 2007). Another advantage is
that CWT-based methods do not necessarily require thresholding, which reduces complexity of an algorithm (Kaspersen et al., 2001).





### 3.9 Hidden Markov Model and Viterbi algorithm

The application of Hidden Markov Models (HMMs) in the context of edge detection is an approach rooted in probabilistic modeling (Ekisheva and Borodovsky, 2006). HMMs, well-known for their efficiency in capturing sequential patterns, offer a great framework for identifying edges in complex and noisy radargrams (Carrer and Bruzzone, 2017; Donini et al., 2022b). They are based on augmenting the Markov chains which describes the probabilities of sequences of random variables to compute probabilities of observable events. In case of radargrams, the observable events are pixel intensities. For edge detection, pixels within a radargram are conceptualized as hidden states, each one associated with emission probabilities indicating local intensity gradients. Transition probabilities, inferred from the gradients of neighboring pixels, represent the likelihood of going from one pixel to another, capturing the contextually-dependent edge characteristics. By optimizing the sequence of hidden states, HMMs effectively capture IRHs in radargrams (Stauffer and Grimson, 1999; Ekisheva and Borodovsky, 2006; Zhang et al., 2008; Bouguila et al., 2022; Carrer and Bruzzone, 2017).

For any task containing hidden variables, it is important to find which sequence of such hidden variables is the underlying source of the desired observation. This is called decoding. One common such algorithm used along with HMMs is the Viterbi algorithm (VA; Viterbi, 1967), a dynamic programming technique, which finds the most plausible sequence of concealed states within a Markov field, depending on a series of observations (Bouguila et al., 2022).

### 3.10 Gibbs Sampling

The Gibbs sampler (Casella and George, 1992) is a Markov Chain Monte Carlo (MCMC) method for indirectly generating random variables from a (marginal) distribution, removing the need to directly calculate the density. Every pixel or region within the image is allocated a label representing its class or segment. Through iterative sampling of labels, considering conditional probabilities in neighboring pixels or regions, Gibbs sampling facilitates the partitioning of the image into coherent segments (Casella and George, 1992; Xiao Wang and Han Wang, 2004).

### 3.11 Support Vector Machine

The support vector machine (SVM) (Vapnik et al., 1996) approach in image segmentation classifies two-class problems by maximizing the margin between classes in an n-dimensional feature space. The closest data points to the discriminating hyperplane are called support vectors, and they are pivotal in defining the discrimination function. Despite the potential existence of multiple discriminating hyperplanes, SVMs are able to identify the optimal surface, mitigating overfitting during training (Burges, 1998).

### 3.12 Deep Learning

Deep learning (DL) is a learning method which includes multiple levels of representation (LeCun et al., 2015). Representation learning is the group of methods through which a machine gets raw data as input and yields outputs that are necessary in classification or detection tasks. DL performs such tasks using multiple levels of non-linear modules, transforming these



representations from raw to higher and more abstract levels (Hinton et al., 2006; LeCun et al., 2015) and in a hierarchical manner (Zeiler and Fergus, 2013; Tomasini and Wyart, 2024). What makes DL outstanding in terms of efficiency is that such

features are not designed by humans but learned from data. DL techniques are classified into three main categories: supervised learning, semi-supervised learning and unsupervised learning. Supervised learning is related to learning from labelled data, i.e. to map the input data to some known targets, while semi-supervised learning deals with semi-labelled datasets. The advantage of semi-supervised learning is that it does not require a large amount of labelled data, but there is the danger of learning irrelevant features. The third class is unsupervised learning. As the name suggests, the learning procedure is based on finding

representations without help of known targets (Alzubaidi et al., 2021; Goodfellow et al., 2016). These capabilities have made DL methods popular in various applications including computer vision (Chen et al., 2022b), image classification (Rawat and Wang, 2017; Plested and Gedeon, 2022), object detection (Arkin et al., 2023), image (semantic) segmentation (Minaee et al., 2020; Emek Soylu et al., 2023), language translation (Maruf et al., 2021), and natural language processing (Zhu et al., 2023).

### 3.12.1   Convolutional Neural Networks

A subset of artificial neural networks (ANN), convolutional neural networks (ConvNets or CNNs; LeCun et al., 1989; Lecun et al., 1998), are at the heart of DL, and the most commonly used DL algorithm (Krizhevsky et al., 2017a). They are designed for processing data that are in the form of multiple arrays, such as language (1D), images (2D) and videos (3D) (LeCun et al., 2015). One important advantage of CNN over their predecessor is their capability to automatically identify relevant features (Gu et al., 2017), and according to Goodfellow et al. (2016) sparse interactions, parameter sharing and equivariant

representations are what makes CNNs preferable to traditional neural networks. Even though it can differ to some extent, their global architecture is composed of the same elements: an input layer, some layers of convolution and pooling, one or more fully-connected layer, an activation function and an output layer (LeCun et al., 2010; Zhao et al., 2024). Fig. 4 (Lecun et al., 1998) depicts a simplified CNN architecture for image classification. Convolutional layers are the most important components of CNNs. They contain a set of convolution kernels to perform convolution process over the input image to generate feature

maps. Their main role is to locate the feature conjunctions from the previous layer (LeCun et al., 2015). The inputs and outputs of each of the stages are called feature maps (LeCun et al., 2010). For the example of images, feature maps would be 2D arrays of a color channel of the input image. Mathematically, the filtering operations are discrete convolutions taking place between the input image and a kernel (or filter) returning feature maps as output (LeCun et al., 2015; Goodfellow et al., 2016). Pooling layers sub-sample similar feature maps to create smaller feature maps, as well as reducing the representation dimension and

constructing an invariance to small shifts (LeCun et al., 2015). Going from one layer to another, a weighted sum of the inputs form the previous layer is passed through a non-linear function. This is the role of activation function. Rectified linear unit (ReLu) (Agarap, 2019) has been the most commonly-used such function. A fully-connected layer, commonly at the end of the CNN architecture (Alzubaidi et al., 2021), is a global operation which means that each neuron is connected to all the ones from its previous layer (Zhao et al., 2024) where neuron is a computational unit which carries weighted input connections from each

layer to the other.



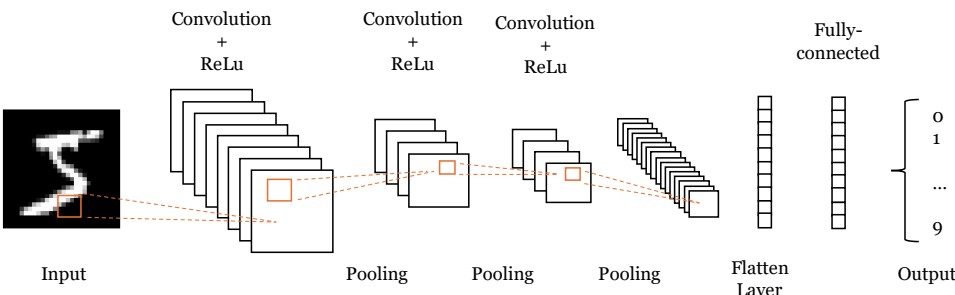

**Figure 4.** A simplified CNN architecture for an image classification task. The architecture is similar to the one of LeNet, which is the early CNN and was used for handwritten digit recognition (Lecun et al., 1998).

CNNs have been employed in a variety of fields such as computer vision (Zhao et al., 2024), speech and language processing (Pham et al., 2016), face recognition (Coskun et al., 2017), object detection (Galvez et al., 2018), to name a few. In recent years, CNNs applications have been expanded to the field of glaciology as well, for instance in calving front delineation using synthetic aperture radar (SAR) imagery (Mohajerani et al., 2019; Zhang et al., 2019), grounding line delineation (Mohajerani et al., 2021), and automatic stratigraphy mapping (e.g., Varshney et al., 2021b; Cai et al., 2022; Wang et al., 2020; Donini et al., 2022c).

### 3.12.2 Autoencoders

Autoencoders are a class of neural networks whose primary task is to copy an output from the given input (Goodfellow et al., 2016). The network has two parts. The primary part of the architecture, minimizing the input size is called the *encoder*, and the secondary part is referred to as the *decoder*. Useful features of the data are learned and those are the ones that have priority when reconstructed. In addition, since the basic idea of an autoencoder architecture is to have the same input and output dimensions, autoencoders are a good choice for segmentation tasks (Goodfellow et al., 2016; Aggarwal, 2018).

One of the most used autoencoder architectures has been the U-net architecture (Ronneberger et al., 2015). Originally designed for biomedical image segmentation applications (Li et al., 2018; Kugelman et al., 2022), in recent years its applications have spread over many different field, from remote sensing for the cryosphere (e.g., Ji et al., 2019; Mohajerani et al., 2021; Varshney et al., 2020; Loebel et al., 2022; Donini et al., 2022a), to plant root research (Smith et al., 2020), and many more applications. There are also many architectures that function on the basis of U-net (Jha et al., 2019; Zhou et al., 2020; Zhang et al., 2021a). The network architecture features a U-shaped structure with an encoder–decoder pathway, facilitating the extraction of contextual information and precise localization of object boundaries. The encoder path uses convolutional and pooling layers to downsample the input image, while extracting high-level features. Thereafter, the decoder path employs upsampling layers to recover spatial information and generate segmentation masks with details. Additionally, skip connections between corresponding encoder and decoder layers make the propagation of high-resolution features possible, to preserve spatial infor-



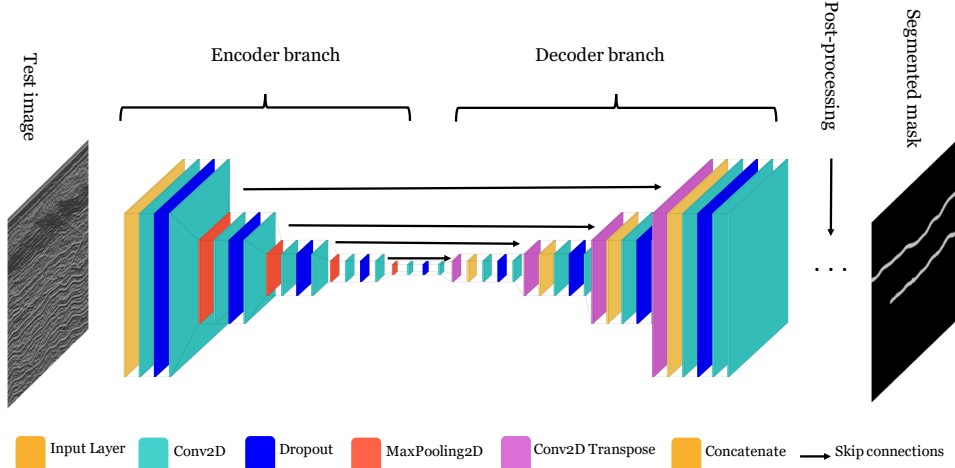

**Figure 5.** Schematic of a U-net architecture, and its conventional components, inspired by (Ronneberger et al., 2015). Left- and right-most images show a radargram and its representation with two IRHs predicted by U-net.

mation throughout the network (Ronneberger et al., 2015; Siddique et al., 2021). Fig. 5 is an example of U-net architecture for image segmentation.

### 3.12.3 Transfer Learning and pre-training

CNNs require a very large amount of training data in order to yield acceptable output (Lecun et al., 1998), and lack of sufficient annotated data for training is one of the challenges of these models. One way to overcome such challenges is transfer learning techniques. Simply stated, such techniques involve training a DL model with a proven architecture using a large amount of data, for example publicly available general purpose computer vision datasets e.g. ImageNet (Krizhevsky et al., 2017a), and fine-tuning it with the limited training data that is available for the task at hand (Weiss et al., 2016). There are a number of available CNN models such as AlexNet (Krizhevsky et al., 2017b), GoogleNet (Szegedy et al., 2014), ResNet (He et al., 2015a) that are pre-trained on large datasets such as ImageNet. It has been shown that using pre-trained models improves robustness of learning process (Hendrycks et al., 2019).

### 3.12.4 Holistically-Nested Edge Detection

Holistically-Nested Edge Detection (HED) (Xie and Tu, 2015) is an end-to-end detection system which learns types of hierarchical features crucial to understanding an image as a whole. HED is inspired by fully convolutional neural networks (Long et al., 2014) with addition of deep supervision on a VGG network (Simonyan and Zisserman, 2014). Contrary to traditional edge detection algorithms which go after abrupt changes in pixel intensity locally, HED considers edge detection as a holistic problem (global image-to-image mapping). Moreover, it uses side outputs compensating for the absence of deep supervision which is a characteristic of fully convolutional neural networks.



### 3.12.5 Multi-scale Learning

Multiscale learning (Elizar et al., 2022) offers significant advantages by employing discriminative-feature representation to improve information acquisition. This is achieved through the fusion of low- and high-resolution data as well as the integration of diverse data sources. Multi-scale learning brings about a higher level of explanation and learning through collective results at different scales. Additionally, it is a feasible method for combining with other advanced networks e.g. generative adversarial networks (Suh et al., 2022). The fundamental concept behind multi-scale feature learning involves the simultaneous construction of multiple CNN models with varied contextual input sizes. These models work in parallel, and their respective features are merged at the fully connected layer (Elizar et al., 2022). An edge detection technique can be implemented at every scale for feature detection (Yari et al., 2020). Small-scale structures are expected to include smaller features, such as local fluctuations of ice layer boundaries (Cai et al., 2022).

### 3.12.6 Recurrent Neural Networks

Unlike other kinds of neural network architectures where variables are independent of each other, Recurrent Neural Networks (RNN) (Cho et al., 2014) are designed for sequential data, such as sentences (Mirowski and Vlachos, 2015), time series (Hewamalage et al., 2021), and biological sequences (Aggarwal, 2018). Simply put, the input of each node is a combination of input and the hidden state of the same node from the previous time step (Goodfellow et al., 2016). In image segmentation applications, RNNs are applied in a way that segmentation is performed as a sequence prediction as well. This sequential processing is the main strength of RNNs for image segmentation (Salvador et al., 2019).

## 4 Timeline of the methods used

In this section, we provide a concise description of the most important and relevant studies that were done on mapping the englacial stratigraphy. For the sake of simplicity, the timeline (Fig. 6) starts in the year 2000 to focus on more modern approaches. As Wang et al. (2020) pointed out, at first the task of tracing IRHs from radargrams seems like a straightforward classic computer vision problem, however in practice it becomes obvious that it is a very challenging endeavour. Difficulties emerge from multiple reasons, such as high spatial variability, complex features, abundance of noise, unknown number of horizons and horizon discontinuity and merging.

Over the last decade, various studies have put forth the use of pattern recognition methodologies in the examination of ground-penetrating radar (GPR) signals (e.g. Delbo et al., 2000). The primary focus of these investigations lies in the identification of specific subterranean entities, such as mines, pipelines, or tanks buried at shallow depths through ground-based GPR. These entities mostly exhibit hyperbolic signatures in radargrams, a distinct contrast from most signatures in radargrams obtained from all platforms (ground, air, orbital) over ice. Consequently, we primarily abstain from including those investigations and their methodological proposals within the purview of this paper.



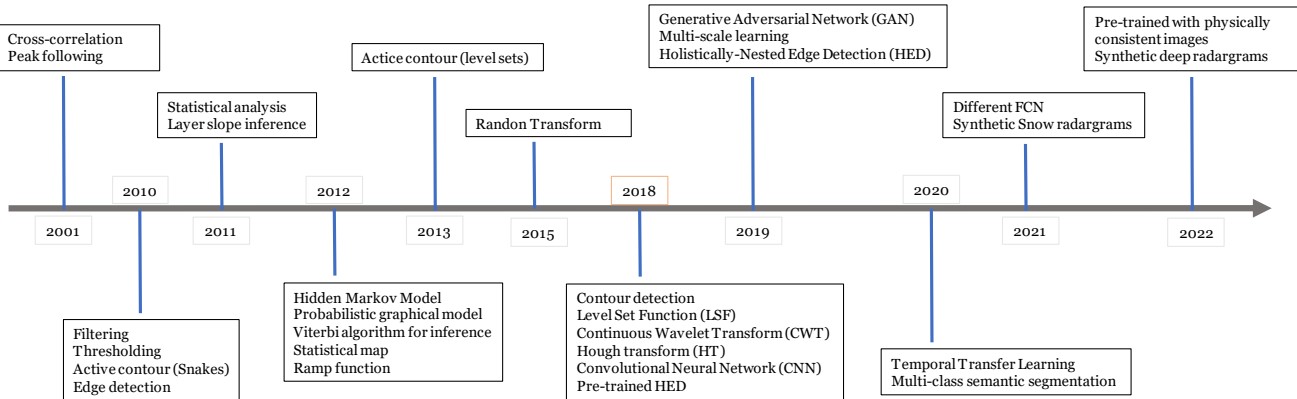

**Figure 6.** Evolution of selected methods for IRH tracing in chronological order, linked to the year each method was applied for the first time.

Automatic feature detection methods have been applied to a variety of applications of RES. Some of these include asphalt/pavement thickness measurements (Zhang et al., 2022), subsurface distress detection (Li et al., 2022), detection of structures beneath rail tracks (Peng Xu et al., 2004), concrete feature tracking (Todkar et al., 2017), mine detection (Frigui et al., 2005; Reichman et al., 2017), and tunnel lining (Zeng et al., 2023). As for ice, this task is traditionally done manually or semi-
automatically. Semi-automated methods include a certain degree of subjectivity. The two steps in which this subjectivity plays the main role are positioning of the seed points (to be connected to each other automatically) and critical evaluation of the results (Dossi et al., 2015). In recent years, there have been studies that attempted to perform this task with automated methods. Such studies include a variety of approaches such as neural networks (Reichman et al., 2017), CNN (Zhang et al., 2022), Laplace transform artificial neural networks (Szymczyk and Szymczyk, 2015), and multilayer perceptron (Sukhobok et al.,
2019), to name some. However, as a result of radar systems differing in frequencies and waveform characteristics (thus resolution and penetration depth), studies applied to GPR and RES systems over mediums other than ice, do not provide considerable insights. Therefore, the description of methods in our timeline (Fig. 6) concentrates on the studies that have studied radargrams from ice sheets and other large ice masses, whether on Earth or planetary bodies.

In a number of studies radargrams were analysed to find different segments or subsurface targets (e.g. englacial bound-
aries, EFZ, basal units) and classes of events in each radargram (e.g., Donini et al., 2019; Goldberg et al., 2020; García et al., 2021, 2023). Even though we focus on the methods for mapping englacial ice structure and tracing IRHs and/or layer boundaries, we also take a look at studies done to detect regions and targets in radar products, since those are, in terms of methodology, in close vicinity to stratigraphy mapping endeavours.

Table 1 gives a comprehensive overview of the published work for a quick look-up of published studies in tracing englacial
stratigraphy. It contains published year, method(s) applied, traces mapped, type of radar system and the regions it was applied to.

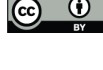



Table 1: Published research on mapping methods.

| Author(s) | Year | Method | Automation Level | Traced reflections | Radar System | Frequency range [MHz] | Applied region(s) |
|---|---|---|---|---|---|---|---|
| Gades et al. (Gades et al., 2000) | 2000 | Time-window maximum calculation | Semi-automatic | Ice-base and englacial | pulse-compression coherent radar | 3–4 | Antarctica |
| Fahnestock et al. (Fahnestock et al., 2001) | 2001 | Peak Following | Automatic | Deep and base | — | 150 | Greenland |
| Gifford et al. (Gifford et al., 2010) | 2010 | Edge-based active contour (Snake) | Semi-automatic | Surface and bed | CReSIS radar | 140-160 | Antarctica |
| Freeman et al. (Freeman et al., 2010) | 2010 | Filtering / Thresholding / Morphological operators | Semi-automatic | surface and deep | SHARAD | 15-25 | Mars NPLD |
| Reid et al. (Reid et al., 2010) | 2010 | Active contour (Snake) | Semi-automatic | Basal return | SHARAD | — | Mars NPLD |
| Ferro and Bruzzone (Ferro and Bruzzone, 2011) | 2011 | Statistical analysis | Semi-automatic | Radargram regions | SHARAD | 1-20 | Mars NPLD |
| Sime et al. (Sime et al., 2011) | 2011 | Layer slope inference | Semi-automatic | Deep IRHs | DELORES/WSB | 4/150 | Antarctica |
| Crandall et al. (Crandall et al., 2012) | 2012 | MRF+VA | Semi-automatic | Ice-base and air-ice | MCoRDS | 140-230 | Antarctica |
| Ilisei et al. (Ilisei et al., 2012) | 2012 | Statistical map | Semi-automatic | base (properties) | MCoRDS | — | Antarctica |
| Smock and Wilson (Smock and Wilson, 2012) | 2012 | VA | Semi-automatic | Surface and layer boundaries | GPR | — | — |
| Ferro and Bruzzone (Ferro and Bruzzone, 2012) | 2012 | Statistical analysis, level-set algorithm | Semi-automatic | subsurface regions | SHARAD | 1-20 | Mars NPLD |
| Ferro and Bruzzone (Ferro and Bruzzone, 2013) | 2013 | Image processing, Steger Filter | Semi-automatic | Subsurface layer boundaries | SHARAD | 1-20 | Mars NPLD |
| Karlsson et al. (Karlsson et al., 2013) | 2013 | Ramp function fitting | Semi-automatic | Bølling–Allerød transition | U Kansas coherent radar | 150 | Greenland |
| Mitchell et al. a (Mitchell et al., 2013a) | 2013 | Active contour (Snake) | Semi-automatic | Near surface (snow layer boundaries) | CReSIS snow radar | — | Antarctica |
| Mitchell et al. b (Mitchell et al., 2013b) | 2013 | Active contour (LSF) | Semi-automatic | Surface and base | CReSIS multichannel coherent radar | — | Antarctica |
| Lee et al. (Lee et al., 2014) | 2014 | MRF; Gibbs sampling | Semi-automatic | Surface and base | CReSIS Coherent Radar Depth Sounder | — | Greenland |
| Panton (Panton, 2014) | 2014 | Active contour (Snake) | Semi-automatic | deep | MCoRDS | 180-210 | Greenland |
| Ilisei and Bruzzone (Ilisei and Bruzzone, 2014) | 2014 | Statistical map, thresholding | Semi-automatic | Ice-EFZ, EFZ-base | MCoRDS | 9.5 | Antarctica |
| Ilisei and Bruzzone (Ilisei and Bruzzone, 2015) | 2015 | Statistical analysis, SVM | Semi-automatic | Regions (layers, base, noise) | MCoRDS, MCoRDS2 | 193.5 | Antarctica |
| Onana et al. (Onana et al., 2015) | 2015 | Radon transform | Semi-automatic | Near surface/firn layer boundaries | Ku-band | 13-17 | West Antarctica |
| Dossi et al. (Dossi et al., 2015) | 2015 | Complex trace analysis | Semi-automatic | Englacial | ProEx Malå Geoscience GPR | 250 | Julian Alps |
| McGregor et al. (MacGregor et al., 2015) | 2015 | Reflection slope estimation | Semi-automatic | Englacial reflections | ICORDS, ACORDS, MCRDS, MCoRDS | various frequencies | Greenland |
| Koenig et al. (Koenig et al., 2016) | 2016 | Image processing | Semi-automatic | Snow and firn boundaries | CReSIS ultra-wideband snow radar | 2-6.5 | Greenland |
| Xiong et al. (Xiong and Muller, 2016) | 2016 | Radon transform | Semi-automatic | Deep IRHs | MCoRDS | 193.9 | Greenland |
| Carrer and Bruzzone (Carrer and Bruzzone, 2017) | 2017 | HMM+VA | Semi-automatic | Deep IRHs | SHARAD | 1-20 | Mars NPLD |
| Xiong et al. (Xiong et al., 2017) | 2017 | Continuous wavelet transform + Hough transform + peak following | Automatic | Deep IRHs | MCoRDS | 150-195 | Greenland |
| Rahnemoonfar et al. a (Rahnemoonfar et al., 2017a) | 2017 | Active contour (LSF) | Semi-Automatic | Surface and base | CReSIS radar | — | Antarctica |
| Rahnemoonfar et al.b (Rahnemoonfar et al., 2017b) | 2017 | Contour detection | Semi-automatic | Surface and base | CReSIS radar | — | Antarctica |
| Xu et al. (Xu et al., 2017) | 2017 | MRF, 3D reconstruction | Semi-automatic | Air–ice and ice-rock | MCoRDS | — | Canadian Arctic Archipelago |
| Khodadadzadeh et al. (Khodadadzadeh et al., 2017) | 2017 | Statistical analysis, SVM | Automatic | Regions (layer, base, noise) | MCoRDS | 193.5 | Antarctica |
| Berger et al. (Berger et al., 2018) | 2018 | VA | Automatic | Base | MCoRDS | — | Antarctica |
| Donini et al. (Donini et al., 2018) | 2018 | HMM+VA - Fuzzy system | Semi-automatic | Lava tubes | simulated radargrams | — | Simulated Moon subsurface |
| Xu et al. (Xu et al., 2018) | 2018 | 3D ConvNet, RNN | Automatic | 3D reconstruction, air–ice, ice-bed | MCoRDS | — | Canadian Arctic Archipelago |
| Kamangir et al. (Kamangir et al., 2018) | 2018 | Pre-trained HED | Automatic | Surface and base | 2009-2016 NASA OIB Mission radar | various | — |
| Yari et al. (Yari et al., 2019) | 2019 | HED, pre-training | Automatic | Shallow IRHs | CReSIS snow radar | — | various polar regions |
| Lines et al. (Lines et al., 2019) | 2019 | Average square difference function | Semi-automatic | Deep IRHs | Radar | 900, 400 | Greenland |
| Donini et al. (Donini et al., 2019) | 2019 | SVM + RBF | Automatic | 4-class regions | MCoRDS | 195 MHz | North Greenland |
| Rahnemoonfar et al. (Rahnemoonfar et al., 2019) | 2019 | HED (mapping), GAN (synthetic radargrams generation) | Automatic | Surface and base | CReSIS radar + Synthetic | — | — |
| Cai et al. (Cai et al., 2020) | 2020 | Deep Convolution network, encoder-decoder, Resnet | Automatic | Region classification, surface and base | MCoRDS, MCoRDS2 | 9.5, 30 | Antarctica |
| Goldberg et al. (Goldberg et al., 2020) | 2020 | Linear/cubic interpolation | Semi-automatic | Surface and base | Polarimetric-radar Airborne Science INstrument (PASIN) | — | Antarctica |
| Delf et al. (Delf et al., 2020) | 2020 | Sobel-Feldman operator | Automatic | Deep IRHs | CReSIS radar | — | Antarctica |
| Yari et al. (Yari et al., 2020) | 2020 | Multi-scale learning, transfer learning | Automatic | Shallow IRHs | CReSIS radar | — | — |
| Ibikunle et al. (Ibikunle et al., 2020) | 2020 | Multi-class Neural network | Semi-automatic | Snow surface boundaries | Simulated Snow radargram | N/A | N/A |
| Keeler et al. (Keeler et al., 2020) | 2020 | Probabilistic method + Radon transform | Automation | shallow IRH | FMCW | 12.5-17.5/2-6.5 | Antarctica |
| Varshney et al. (Varshney et al., 2020) | 2020 | FCN | Automatic | Snow/firn | CReSIS radar | 2-6.5 | Greenland |



| Author(s) | Year | Method | Automation Level | Traced reflections | Radar System | Frequency range [MHz] | Applied region(s) |
|---|---|---|---|---|---|---|---|
| Wang et al. (Wang et al., 2020) | 2020 | CNN, RNN | Automatic | Deep IRHs | CReSIS radar | 2-6.5 | Greenland |
| Rahnemoonfar et al. (Rahnemoonfar et al., 2021) | 2021 | CNN | Automatic | Snow surface boundaries | CReSIS Snow radar | 2000-18000 | Greenland |
| Varshney et al. (Varshney et al., 2021b) | 2021a | FCN | Automatic | Snow layer boundaries | CReSIS snow radar | 2000-6500 | Greenland |
| Varshney et al. (Varshney et al., 2021a) | 2021b | CNN regression network | Automatic | Snow layer thickness | CReSIS radar | 2000-6500 | Greenland |
| García et al. (García et al., 2021) | 2021a | DL, unsupervised | Automatic | subsurface regions | MCoRDS2 | 193.9 | Antarctica |
| García et al. (García et al., 2021) | 2021b | CNN, domain adaptation, transfer training | Automatic | Subsurface targets | MCoRDS3 | 190 | Antarctica |
| Donini et al. (Donini et al., 2021) | 2021a | Unsupervised learning | Automatic | subsurface regions | MARSIS | 1.8, 3, 4, or 5 | Mars South Pole |
| Varshney et al. (Varshney et al., 2022) | 2022 | CNN-based Regression model | Semi-automatic | Snow layer thickness | CReSIS radar | — | Greenland |
| Cai et al. (Cai et al., 2022) | 2022 | DL, MFFN | Automatic | Surface and bottom | CReSIS radar | 9.5 | Antarctica |
| Donini et al. (Donini et al., 2022c) | 2021b | DL, U-net | Automatic | Basal layer and basal unit | MCoRDS3 | — | North Greenland, west Antarctica |
| Donini et al. (Donini et al., 2022b) | 2022 | HMM+VA, fuzzy method | Automatic | Lava tubes | Lunar Radar Sounder (LRS) | 5 | Moon |
| Dong et al. (Dong et al., 2022) | 2022 | Deep neural network | Automatic | Base and englacial | Synthetic data, CHINARE | N/A | Antarctica |
| Tang et al. (Tang et al., 2022) | 2022 | DL | Automatic | deep IRH and ice-base | HiCARS | 60 | Antarctica |
| Liu-Schiaffini et al. (Liu-Schiaffini et al., 2022) | 2022 | CNN + CCRF | Automatic | Ice-bed interface | HiCARS | 60 | East Antarctica |
| Ghosh et al. (Ghosh and Bovolo, 2022b, a) | 2022 | Transformer-based DL | Automatic | region classification | MCoRDS | 193.5 | Antarctica |
| Ghosh et al. (Ghosh and Bovolo, 2023b) | 2023 | self-supervised transformer | automatic | multiple regions (3 target classes) | MCoRDS | 193.5 | Antarctica |
| García et al. (García et al., 2023) | 2023 | DL, transfer learning | Automatic | Subsurface regions | MCoRDS1 and MCoRDS3 | — | Antarctica |
| Ibikunle et al. (Ibikunle et al., 2023) | 2023 | DL, multi-class classification | Semi-automatic | Snow layer boundaries | FMCW | — | Greenland |
| Jebeli et al. (Atefeh Jebeli et al., 2023) | 2023 | DL, U-net | Automatic | Surface and bottom | CReSIS radar | — | Greenland |
| Zalatan et al. (Zalatan and Rahnemoonfar, 2023) | 2023 | GCN - RNN | Automatic | Snow layer boundaries | CReSIS radar | 2-8 | Greenland |
| Varshney et al. (Varshney et al., 2023) | 2023 | dl-wavelet transform | automatic | snow layer boundaries | CReSIS snow radar | 125 | greenland |



Early tracing of IRHs have been performed in semi-automatic or manual methods. For instance, Dahl-Jensen et al. (1997) mapped IRHs from RES data to search for a new ice core drilling site. Eisen et al. (2004) and Steinhage et al. (2013) connected ice-core age–depth relationships by connecting the drilling locations using mapped RES data.

We present published studies classified in three main categories of methods, in subsection:

4.1: Computer vision-based methods; this category consists of filtering and thresholding methods, peak following, layer slope inference methods, transform methods such as Radon and wavelet transforms and active contour methods.

4.2: Probabilistic and statistic methods; this contains of Markov methods, Viterbi, statistical mapping, support vector machine and gibbs sampling.

4.3: Deep learning methods include CNN-based models and related architectures and modifications.

In the following subsections, we present published studies within these three main categories grouping connected works in individual paragraphs. For studies which employed multiple approaches, we have included each in the category where its main method belongs to. It should be kept in mind that categorization in separate groups serves mainly the sake of readability and coherence and does not indicate strict separation between categories.

**4.1 Applications of Traditional Computer Vision and Signal Processing Methods**

Gades et al. (2000) and Nereson et al. (2000) both have employed an semi-automatic picking routine for bed and deep horizons, respectively. Their method finds the maximum amplitude of the each of the RES traces in a prescribed time window. Their pre-processing step is the noise removal using a fourth-order Butterworth band-pass filter. Similarly, **Fahnestock et al. (2001)** used peak-following and cross-correlation and implemented it on the pulse-compressed coherent radar data. Peak following
was used to select and follow IRHs on the radargrams taken by the NASA radar system. This method is mainly automatic, however there is an operator-assisted element to ensure accuracy. Radargrams were visually improved by normalizing them from the depth of 400 m and lower in order for the operator to better identify the reflections. They remark that peak following is the method of choice for areas where peaks are distinctly visible but patterns are ambiguous. The correlation method, in contrast, is more suitable for instances where peaks are faded and patterns are more distinct. **Lines et al. (2019)** presents a
hybrid method founded upon Fahnestock et al. (2001) to track the IRHs in radargrams covering the snow and firn columns. These interactive semi-automatic methods seems to be sensitive to noise, acting as a weakness for GPR data. The method is called Average square difference function Layer Picking System (ALPS) named after the algorithm it uses for tracing IRHs. Connecting discontinuities that exist especially near crevassed terrain or close to ice sheet margin in radargrams is a weakness of this method.

In the planetary ice sheet domain, **Freeman et al. (2010)** implemented a method to detect near-surface ice layers in Mars' ice–rich Northern Polar Layered Deposits (NPLD) from radargrams of the Shallow Subsurface Radar (SHARAD) mounted on NASA's Mars Reconnaissance Orbiter. SHARAD radargrams are equipped with linear frequency modulation (LFM) which operates in the range of 15–25 MHz. SHARAD's products are often used for tracing or segmenting IRHs and regions of



ice. As a pre-processing method they use a Gaussian blur and a high-pass filter. The main techniques for detecting IRHs are
thresholding and morphological processes.

**Gifford et al. (2010)** used the edge method and edge-based active contour to pick the surface and basal reflections in an
attempt to estimate the thickness of ice sheets. To estimate the ice thickness, one would need two reflections: the first peak
represents the ice surface and the last peak representing the base. These two peaks are the strongest in their vicinity for
every individual trace. They list some of the shortcomings of traditional segmentation methods for this task such as watershed
(Beucher and Lantuejoul, 1979), level set (Chunming Li et al., 2011), and region growing (Pal and Pal, 1993) methods. The
main reason for these shortcomings is that the base reflection of an ice sheet is neither reliably continuous nor connected,
e.g. in regions with steep topography of very thick ice (> 4 km). Other obstructions are the presence of noise and the non-
uniformity of the characteristics of ice sheets because of the deep reflections and surface multiples. Furthermore, the basic
edge detection techniques suffer from a lack of continuity, stiffness and smoothness which makes them not optimal stand-alone
choices for this task. They applied their approach on the data from the University of Kansas, the Center for Remote Sensing
of Ice Sheets (CReSIS). They present a visual comparisons of the basal horizon picked by 1) a human expert, 2) edge-based
approach, and 3) active contour approach. It is concluded that the active contour method provides the best approximation of
the base, with a trade-off in time, as it takes longer for the contour to end up on the basal reflection compared to the edge
detection method. The advantage of the active contour over the edge-based approach mostly lies in the ability to bridge the
discontinuities. However, the weakness of the active contour seems to be in regions where the texture below the base is rough
or where strong IRHs or other reflection signatures occur near the ice-sheet base (Gifford et al., 2010). Automatic mapping
the air-ice and ice-bed boundaries is a task which is tried by various studies throughout the years, to calculate surface or basal
topology, or ice thickness. **Reid et al. (2010)** used two different methods to map the ice surfaces and basal reflections aiming
to estimate the ice thickness, similar as techniques presented by Gifford et al. (2010). They apply their implementation to data
from CReSIS and concluded that although edge-based methods are faster than active contour, the latter is more robust to image
artifacts and yields better continuity in traced IRHs than edge-based methods. One interesting suggestion is that the edge-based
methods can be used as the initialization for the active contour method.

**Sime et al. (2011)** used an automated finite-segment method to obtain englacial reflector dip angles, from both airborne and
ground-based radar observations. In fact, this is not a interface-picking method, as dips are horizontally integrated to produce
synthetic isochrones. Integration of slope of local layers yields first-order approximation of the layer location. The method
is a horizontal averaging technique that reduces the noise and identified the layers, but does not attempt to tracing complete
horizons. Similar approaches, e.g. Holschuh et al. (2017) on computing slope information from radargrams further improved
our understanding of the inner structure of the ice sheets.

**Ferro and Bruzzone (2013)** focus on automatic characterization of the linear features in radargrams taken from Mars
NPLD. They considered some previous studies employing the Hough transform for radargrams (Gamba and Lossani, 2000;
Pasolli et al., 2009) to be useful for detecting hyperbolae or straight lines. Nonetheless, their efficacy decreases significantly
when applied to radargrams which include features that are not straight, which is also the case with GPR products from glacier
ice. The output is a vector object described by its local width and computing features' geometrical characteristics without





further post-processing is possible. The method is composed of a combination of some image processing techniques, namely
pre-conditioning, block-matching and 3D-filtering processes (BM3D) to remove the noise, and subsequently a Steger filter
(Steger, 1998) for ridge detection. For denoising, the main filter used is the BM3D (Dabov et al., 2007) which while removing
noise, preserves linear features (Ferro and Bruzzone, 2013).

To investigate the depth of the Bølling–Allerød interstadial, an important warming event of the last deglaciation **Karlsson
et al. (2013)** presented an automatic fitting method. Having dated the transition at an ice-core drill site, they extend this
transition further away from the ice core site. They noticed that the upper half of the radargrams from CReSIS RES data from
Greenland contain much more reflections than the lower one, and subsequently fit a ramp function to represent the standard
deviation of the data. They constructed the data histogram using an optimal number of bins and reconstructed the radargram
using intensities of the mean values of each bin. They then fit a 2D version of the ramp function to the more interesting parts
of the reconstructed radargram. Finally, the location of the most of the fits is selected as the depth of the transition. They
concluded that the absence of the transition is the result of ice flow having disrupted the layering (Siegert et al., 2004) and it is
not possible to recognize layering in thin ice (Fahnestock et al., 2001).

**Mitchell et al. (2013a)** used the method proposed by Steger (1998) to find the estimated position of the curves belonging
to horizons. They combine off-the-shelf methods to estimate near-surface layers in radargrams of the snowpack. It finds points
with high probability to be part of a curvilinear structures and therefore, the features are more parallel and less fluctuating. A
user is required to initially determine the number of visible layers. The pipeline starts with edge detection to estimate the layer
location, curve point classification to estimate reflection location and finally active contour (Snake) to detect lower reflections
from the upper ones. They noted that the Canny operator is only suitable to identify near-surface surface but not deeper
reflections, due to fainter layers boundaries and inherent noise of the radargrams. **Panton (2014)** introduced two methods
that respectively infer the local slope and track a boundary from the initial estimate. This method traces IRHs by optimizing
the position of the entire IRH to improve the areas with poor radargram quality. After pre-processing, local boundary slopes
are estimated and seed points are picked by a human expert. From here, the Snake algorithm traces the boundaries from the
picked seed points. In areas where there are discontinuities in the IRHs, for instance in the presence of shear margins or
disturbing fast flow, the method yields incorrect slope fields, therefore it is not recommended for areas where stratigraphy is
visibly complex. In a similar work, **Mitchell et al. (2013b)** also employed LSF to semi-automatically estimate the surface and
basal interface from a multichannel coherent radargrams. An expert provides two initial contours for the air–ice and ice–base
boundaries. These lines then evolve at each step while a cost function is calculated and this continues until the cost function
reaches minimum using gradient descent. This method was applied to 20 radargrams and provided a comparison between this
approach and HMM approach of Crandall et al. (2012), and it was shown that level set performs between 3 and 5 times better
than HMM for air–ice and ice–base boundaries respectively. **Rahnemoonfar et al. (2017a)** also used a LSF method to detect
the topology of surface and basal boundaries. 323 radargrams from the NASA's OIB mission were tested on and iterated 800
times from the initial curve to overlap with the air–ice and ice–base boundaries. The difficulty of processing air–ice and ice–
base boundaries are classified as: i) Subglacial topography is greatly variable, ii) there are artifacts in the data, mainly a result
of electric devices around the radar as well as the aircraft itself, and iii) ice–base boundary exhibits low signal to interference





and noise ratios (SINRs). In another study, **Rahnemoonfar et al. (2017b)** detect air–ice and ice–bed inspired by Coulomb's
electrostatic law based on the projection profile of the contours. IRHs are interpreted as contours. The gray-scale intensity of
a pixel symbolizes the electrical charge associated with each particle. By establishing specific criteria that translate electrical
charges into pixel characteristics, the resulting computation of the electric field for each pixel is used to define the edges within
the radargram. To remove noise, an anisotropic difference is applied.

Semi-Automated Multi-layer Picking Algorithm (SAMPA) introduced by **Onana et al. (2015)** is an algorithm that traces
annual accumulation layers from firn radargrams from both airborne and ground-based radar systems. A Radon transform is
used to map the features which represent firn-layer boundaries, which are later traced using thresholding of the amplitude.
SAMPA was applied to radargrams from West Antarctica where the firn layers are known to be rather flat. Deeper boundaries
are more difficult than shallow ones for SAMPA to trace because of attenuation and low contrast resulting from low variance
in trace amplitude. **Xiong and Muller (2016)** mention the difficulty of tracing internal IRHs are due to presence of non-
contiguous IRHs, and relating that to local anomalies such as folds and crevasses caused by ice dynamics. The underlying
method here is also the Radon transform similar to Onana et al. (2015) to extract IRHs from radargrams of NEEM station.
However, they mention that Onana et al. (2015) extracted firn IRHs which are represented as horizontal lines in radargrams,
and in this study, the goal was to also extract deeper IRHs with larger slope bending. Therefore, they adapt SAMPA which is
based on block processing and convert it to slice processing. They present a comparison of the two methods in one radargram
and conclude that the present method functions more effectively, especially for deeper IRHs. It is worth mentioning that the
only measure of effectiveness presented is qualitative. Continuous wavelet transform as a strong signal processing tool was
used together with Hough transform (Ballard, 1981) by **Xiong et al. (2017)** to semi-automatically trace IRHs and infer the
local dips and propagate them away from the CWT seed points. The critical problem with peak following is high possibility to
fall onto neighbouring layers. Moreover, in determining which pixels/regions might constitute peaks and which might not, they
note that high-amplitude peaks do not always correspond to layers, while conversely, low-amplitude points can indeed signify
peaks. A post processing algorithm is triggered that connects the lines that belong to each other. They compare their results
with MacGregor et al. (2015) and show that more horizons were traced using CWT and Hough transform. This is related to the
fact that seed picking in McGregor's method was done manually and consequently only prominent horizons were picked. For
instance, IRHs around folds are not picked in the Radiostratigraphy and Age Structure of the Greenland Ice Sheet (RRRAG)
dataset (MacGregor et al., 2015). There is a good agreement between their result and the published RRRAG radiostratigraphy
dataset, while the average vertical difference (between the traced IRHs of the two methods) is 15 pixels which corresponds to
40 meters.

Phase information has also been useful in developing IRH tracing methods. **Dossi et al. (2015)** developed a method to
detect and trace reflections with lateral phase continuity and to assess polarities (to evaluate the materials) in GPR and seismic
data using attribute analysis (Taner et al., 1979). As any single reflection is the outcome of a series of phases with alternating
polarities, it is difficult to determine the initial phase, i.e. polarity. The method is rooted in automatic identification of reflection
phase using the cosine of the instantaneous phase and search for sub-parallel events, in other words, tracking events whose
phase continues laterally. The cosine phase of signals was employed to reconstruct the reflected wavelet's shape. Synthetic



GPR data created with GPRmax Giannopoulos (2005) were used for testing the method, in addition to real data from Alpine
glaciers and archaeological surveys. Similar to other methods, it was found that a limitation of polarity assessment is that there
exist many closely-spaced parallel reflections and these could be recognized as single events. As in intensity gradient methods,
including a thresholding step is a necessity to select only stronger events. The method is able to pick numerous short length
horizons but falls short in dealing with discontinuities, in a manner than the method picks up phases of unrelated events as
a single horizon. **MacGregor et al. (2015)** introduce two interactive semi-automatic methods to *predict* internal stratigraphy
based on phase information as well. They rely on calculating reflections' slopes, similar to the works of Sime et al. (2011)
and Panton (2014). Integrating the slope in along-track direction would yield the stratigraphy (predictions). The first method
uses the smooth horizontal phase changes along the radar track, and natural coherence of radar signals to predict reflection
morphology benefiting from recordings of coherent radar while implementing SAR techniques as a backbone (Raney, 1998).
The second method calculates the reflection slope by extracting information from the wavenumber of the Doppler centroid
in radar data. Fourier transform is computed on brief segments of radar data to examine the Doppler spectrum of englacial
reflections. **Goldberg et al. (2020)** developed an algorithm to detect basal units. They introduce a SAR processing method and
employed unique phase shift response functions to classify feature types such as englacial layers and potential basal units. It
can distinguish between them by matching model phase shift responses with pixel data.

Noise removal is the initial step for most of the methods, such as **Koenig et al. (2016)** who apply a median filter to re-
575 move noise, and detect snow surface simply by thresholding. They devised a semi-automated snow layer boundary detection
algorithm and applied it to radargrams from NASA's OIB Arctic Campaign taken from 2009–2012 surveyed by CReSIS' ultra-
wideband snow radar. Identification of peaks is facilitated by the distinction between spatial variability in travel-time–depth
domain across high and low frequencies, functioning as a high-pass filter. These identified points are subsequently linked to
form coherent IRHs by utilizing the half-maximum width of the waveform associated with each peak. After connecting the
580 picked points using spline fitting, an expert examines the picked IRHs by correcting indices, filling gaps, deleting, adding and
other corrections.

**Delf et al. (2020)** performed a inter-comparison of automated IRH tracing and layer-dip-estimation methods. In order to asses
their capabilities, two types of algorithms are implemented: those that trace IRHs and those that extract the slope or dip of the
horizons; and their capability to propagate an age–depth model is tested. They used two CReSIS Multichannel Coherent Radar
Depth Sounder (MCoRDS) datasets from Antarctic surveys for the inter-comparison study and implemented three algorithms:
i) the Automated RES Englacial Layer-tracing Package (ARESELP) implemented by Xiong et al. (2017), ii) Steger algorithm
implemented by Ferro and Bruzzone (2013), iii) Sobel-Feldman operator (also called Sobel(Sobel and Feldman, 2015)). To
assess each method, its impact on propagation of a simple age–depth relationship (Nye, 1963; Leysinger Vieli et al., 2007)
is investigated. The central conclusion is that there is a requirement for further studies and advances in such algorithms, as
even in a region with relatively simple geometry, the methods do not show promise in linking age–depth profiles between two
locations. It is important to note that all three methods performed better in dip-estimation compared to their performance in
IRH tracing.



## 4.2 Applications of Probabilistic and Statistical Methods

**Smock and Wilson (2012)** developed a layer boundary detection identification algorithm for GPR radargrams captured by
vehicle-mounted sensor arrays. Their work is an extension of Smock et al. (2011). Finding base and surface reflections in
radargrams was implemented using VA, representing radargrams as trellis graphs. To investigate subsurface layer boundaries,
one has to look for multiple disjoint paths through the trellis graphs. They propose a criterion for choosing multiple disjoint
paths called reciprocal pointer chain. However, due to lack of ground-truth, no quantitative comparison was provided.

The statistical analysis by **Ferro and Bruzzone (2011)** present a method to detect scattering areas of a radargram, namely
the basal returns. The data is from SHARAD radar in a lower frequency range of 1–20 MHz. They classify radargram regions,
such as strong layers, weak layers, no target and basal layers. For each of these classes, they analyzed statistical distribution
of their corresponding returns by means of fitting three pdfs, namely the Rayleigh, Nakagami and K pdfs. Using maximum
likelihood, the parameters of these pdfs for each class type were estimated. For evaluation, they calculated the root mean
squared error (RMSE) and the Kullback-Leibler divergence (Lin, 1991) between the normalized histogram of data and the
obtained histogram. The best fitting is shown to be the K-pdf distribution in almost all the cases. The product of the method is
used to calculate the NPLD thickness, local geology and seasonal variations. To ascertain a selection of statistical distributions
capable of characterizing amplitude variations caused by distinct subsurface categories **Ferro and Bruzzone (2012)** analyzed
radar sounder products from Mars NPLD once more with two main objectives. One is computing statistical properties of
radargrams, and the other is devising two methods for automated extraction of subsurface features. They previously found that
the signal statistical distribution pertaining to different targets can be modelled efficiently using the K-pdf distribution and
the background noise can be modelled by a Rayleigh distribution (Ferro and Bruzzone, 2011). They generate a map for the
different subsurface classes, and additionally identify and map the deepest scattering areas. They considered their approach as
a first step for a general framework for analysis of radargrams.

**Crandall et al. (2012)** proposed finding boundaries between layers of different materials (such as air, ice, rock, etc) in
radargrams as an inference problem on a statistical graphical model using Markov Random Field (MRF), HMM and VA
(Crandall et al., 2012). This inference model includes a number of constraints, such as layer boundaries being located where
there are high radargram contrasts, being continuous and not intersecting with others. The hidden variables are pixels that
belong to an IRH, and VA is supposed to return the maximum likelihood path of the HMM. They used 827 radargrams with a
size of $700 \times 900$ pixels from NASA Operation Ice Bridge (OIB) data as test data. These radargrams were manually labelled
as well, serving as ground truth. They divided their data set into a training and test set with almost 50% for each set. The
method showed better results for identification of the air–ice interface and could pick these IRHs in a rather short time. By
adding the constraints and human interaction, the error was decreased to some extent. One notable advantage of probabilistic
models is their robustness against inherent radar noise. **Lee et al. (2014)** also considered the task of tracing air–ice and ice–bed
boundaries as a probabilistic inference problem and used Markov-Chain Monte Carlo to sample from joint distribution over all
the possible layers in each radargram. The advantage of this approach is that it brings multiple possibilities to integrate over
and obtain the layer boundaries as opposed to edge detection techniques which present hard singular boundaries. IRHs are





computed from expectations of the distributions, and confidence intervals are computed from variance of samples. The used Gibbs sampling due to its ability to characterize uncertainties by calculating confidence intervals. They show that their method yields an improvement over the approach of Crandall et al. (2012). The challenges are the presence of noise, faint reflections
between boundaries, and the confusing structure that is a result of signal reflections and clutter. Random noise prevents correct picking because of serious signal distortion. **Xu et al. (2017)** aims to reconstruct a 3D structure of the ice sheet from available 2D profiles. They approach this as an inference problem and employ probabilistic graphical model, namely MRF to resolve the reconstruction. They search for a surface that minimizes a discrete energy function on a first-order MRF. This study takes advantage of 3000 radargrams for each of the seven topographic sequences which were resolved, corresponding to 50 km.
Results were compared to the extruded results of Crandall et al. (2012) and Lee et al. (2014), showing that this study presents a more robust method. **Berger et al. (2018)** worked on modifications to the previous methods (Crandall et al., 2012; Xu et al., 2017). They assume that the surface boundary is known a priori, and the focus is on the extraction of the ice–base boundary. Another useful assumption is that the interface between ice and base is single-valued in the radargram, which means that every column of the radargram can only contain one pixel belonging to this interface. By trial and error, they conclude that
the optimum result is obtained for 50 iterations. The modification makes the method more appropriate for a wide range of radargrams, based on comparison with results of Crandall et al. (2012); Lee et al. (2014); Rahnemoonfar et al. (2017a).

To calculate ice thickness and basal properties **Ilisei et al. (2012)** used the MCoRDS data to constructed a statistical map that accentuates the predominant subsurface features within the ice. The map is subsequently partitioned, facilitating the iden­tification of regions associated with basal scattering. Additionally, the method identifies the surface, thereby enabling precise
ice thickness computation. The map is segmented into distinct sectors such as base, englacial fold zones and englacial layers. It was applied to seven radargrams that have been pre-processed using the minimum variance distortionless response (MVDR) approach. Manual initialization is integrated into the procedure, rendering it a semi-automatic method. Weaknesses are pre­sented as i) results depend on the presented model of the radargram (not usable for other radar systems), and ii) the method uses spatial correlation of the subsurface features. To recognise specific ice subsurface targets and estimate their characteristics, **Ili­-**
**sei and Bruzzone (2014)** suggest a semi-automatic method based on statistical map generation. It is based on a comprehensive understanding of the statistical attributes of the radar signal and the spatial arrangement of subsurface targets. Firstly, a statis­tical analysis of the radargram is performed to yield a statistical map of it. This is followed by thresholding and segmenting this map into the areas of interest (horizons, noise, EFZ, and base). Having the same objectives as Ilisei and Bruzzone (2014), **Ilisei and Bruzzone (2015)** propose a different method to obtain the specific subsurface targets applying feature extraction and
automatic classification consecutively. Features are extracted using statistical analysis, similar to Ferro and Bruzzone (2012) and the classification task is performed using am SVM classifier (Cortes and Vapnik, 1995). After statistical analysis, in which different classes are represented by pdfs, the next step is to obtain an approximation of the location and spatial distribution of the classes, e.g. expected order of the classes and their extension in along-track direction. Once these targets and their features are known statistically, this information is preserved as input to an SVM for predicting regions in unseen radargrams. SVM is
chosen mostly thanks to its capabilities in generalization and non-linearity comprehension. Being binary classifiers, multiple SVM classifiers were used in the architecture. The technique has proven to be robust in handling radargram heterogeneity,



as a result of the integration of both statistical and machine learning methodologies. Moreover, SVM is capable of objective extraction since the same criteria for feature extraction were used for all radargrams. Furthermore, it is possible to parallelize the method, it can perform at high speed. Another attempt to classify various englacial regions was done by **Khodadadzadeh**
**et al. (2017)**. The novelty of this work lies in that the window for region classification is not fixed but its size is adaptively changed with regard to characteristics of englacial structure, in order to classify the radargram into ice, base and noise. By using this adaptive window, the method is able to extract features from a radargram by using piece-wise constant representation of the back-scattering signal in the vertical direction to adapt the window size. These extracted features are used in an SVM to perform the classifications same as Ilisei and Bruzzone (2015). They compare their results to their previous work Ilisei and
Bruzzone (2015) and observe an improvement.

Going away from classification of regions and ice-base and ice surface IRHs to tracing internal IRH, **Carrer and Bruzzone (2017)** used radargrams taken from Mars NPLD by SHARAD and implement a method to automatically trace (shallow and deep) IRHs. Their approach uses a combination of local scale HMM and VA. Detection of IRHs involves inferring the most probable boundaries within a section of the radargram, employing an algorithm to patch together local IRH locations. Further-
more, a radargram enhancement and denoising technique is introduced. One issue with VA is that when two IRHs are very close to each other, it mistakenly interprets them as one layer and might jump from one to the other while tracing. To facilitate the comparison, the same radargrams as Ferro and Bruzzone (2013) were used. Similar to Ferro and Bruzzone (2013), they investigate the upper part of the radargram as it is the shallow region near the surface where the spatial density of the IRHs is high. The high computational complexity caused by large IRH density and long azimuth acquisition is resolved by the divide
and concur approach, i.e. the radargram is divided into blocks, within which both inference and discrimination between noise and layer boundaries are conducted. For the ultimate purpose of finding an alternative place to Earth for humans, the Moon has been investigated and lava tubes have been proposed to be an optimal place for human habitat. **Donini et al. (2018)** developed a method to detect such features using analysis of radar sounder data. Their method is composed of two steps: i) extraction of linear features from radargram amplitude and phase using the method of Carrer and Bruzzone (2017) (HMM + VA) and, ii)
evaluation of features using a fuzzy-logic-based system in order to detect the tubes. Another attempt for automatically detect subsurface lava tubes on the Moon was performed by **Donini et al. (2022b)**. A fuzzy system that extracts these tubes on the basis of their geometrical features in radargrams is utilized. The method is an extension of Donini et al. (2018). It is an unsupervised method and the analysis itself is a fuzzy detection system. The steps include firstly to improve the signal to noise ratio by using a conditional density function (CDF) of the noise and the signal, then using HMM and VA to detect the lava tube
signatures. Eventually, a fuzzy logic-based system analyzes the extracted lines in the radargram.

With the aim of categorizing radrgrams into four distinct regions: ice layering, refreezing ice, base, the EFZ and thermal noise, **Donini et al. (2019)** propose a method for radargram segmentation. The class termed "refreezing ice" is referred to liquid water which creates a particular structure when it is in contact with ice and accretes to the overlying ice at the ice–bed interface (Carter et al., 2017). The labeled samples are described by a set of features that discriminate between the different
structures present in the radargram. The segmentation process involves applying a pixel-based classification using an SVM classifier with a Gaussian radial basis function (RBF) kernel. The classification is supervised, meaning that the classifier learns





the characteristics of the different classes from a set of labeled samples selected from the radargrams. The authors performed cross-validation to determine the optimal parameters for the RBF kernel of the SVM classifier. Once the classifier is trained, it assigns a label to each pixel of the radargram based on its learned characteristics. The resulting segmentation map shows the distinct regions of the radargram, categorizing the different geological structures and noise present in the data. The algorithm was applied to radargrams obtained in north Greenland and showed high accuracy in mapping the refreezing ice.

**Keeler et al. (2020)** devised a fully automatic probabilistic method to estimate the surface mass balance from radargrams. The method primarily utilizes successive peak following and Monte Carlo simulations. They present the Probabilistic Automated Isochrone Picking Routine (PAIPR), with the aim of estimating annual surface mass balance in the upper 25 m of dry firn. The data is obtained by a frequency-modulated continuous-wave (FMCW) radar, both ground-based and airborne. After amplifying signal-to-noise-ratio, layer gradient field is estimated using a local Radon transform with a moving window. The next step is to find peaks and subsequently grouping them together to form IRHs. Afterwards, they assign the likelihood of isochrones using a logistic regression algorithm. Lastly, the age–depth scale for each of the picked layer boundaries is estimated. After comparison with the method of Onana et al. (2015), it is concluded that this study presents a more robust method. Furthermore, some of the limitations of the presented method are stated, one being that it requires radargrams which are taken from spatially longer sections.

## 4.3 Applications of Deep Learning Methods

In this subsection, summaries and main points of the studies that used deep learning-based methods are presented. We would like to note that although the primary foci of some of the the works e.g. Donini et al. (2021, 2022c); Garcia et al. (2021); García et al. (2023); Ghosh and Bovolo (2022a, 2023b) lie in radargram region segmentation, they are included in this review because of their methodological relevance for the overall objective.

To perform the same task as Xu et al. (2017), i.e. 3D reconstruction of the ice sheet using 2D radargrams, **Xu et al. (2018)** present a DL-based method for tracing basal or englacial IRHs. A multi-task spatio-temporal neural network that combines 3D ConvNets and RNN is constructed. The specific RNN is called Gated Recurrent Units (GRU), commonly used for learning sequential data, which is coupled to a (Convolutional 3D) C3D network (Tran et al., 2014). Their dataset and study regions are the same as Xu et al. (2017). Comparison with Crandall et al. (2012) and Lee et al. (2014) show that this approach achieves better results, as methodologies of those studies were designed for 2D segmentation. Additionally, another comparison with Xu et al. (2017) concludes that this approach yields slightly poorer results because Xu et al. (2017) utilizes more information, i.e. additional non-visual data such as prior weak information about ice thickness information from satellite maps. However, without this information, this study yielded more accurate results. Also, the present method is much faster than their previous study (Xu et al., 2017) as the former method is resting on statistical analysis. To further evaluate their method, they ran several baselines to gauge the different components of their architecture, observing that all the components play a vital role in the final outcome.

To trace the ice-air and ice-bed horizons, **Kamangir et al. (2018)** initially applied an undecimated wavelet transform for removing speckle noise from the radargrams due to its translation invariance. The next step is a multi-step neural network to



extract the edges from the radargram. The architecture is a pre-trained HED, built from a series of convolutions. After each set of convolution there are two outcomes, one of which goes through a max pooling step and next step of the convolution (thus shrinking in size) and another one is taken as a side output. These independent networks combined with the result of each side output form the final output of the model. **Yari et al. (2019)** used NASA OIB ICE2012 data to implement a multi-layer learning

HED to map the shallow IRHs. Three experiments were performed: i) using a pre-trained model, ii) training on a synthetic radar dataset and iii) training with a normal distribution initialization. Pre-training was performed on the BSDS500 data (Martin et al., 2001) and its augmentation (Arbeláez et al., 2011). Even though transfer learning seems to be a good approach, because most neural networks are trained on optical imagery with much less amount of noise, the pre-training approach yielded poor results, or did not converge overall. The synthetic dataset uses a simple linear superposition for IRHs, and layer thickness

model generated by a smoothed Gaussian random process, produces a simplified and not very realistic dataset. Thus, training on synthetic data did not return acceptable results either. It was concluded that training a model from scratch gives the best results out of three. To synthesise radargrams, a generative adversarial network (GAN) (Goodfellow et al., 2014) was used by **Rahnemoonfar et al. (2019)** and later an HED model was trained using this synthetic dataset. They noted that such radargrams cannot completely replace the use of real radargrams for training since they do not contain all characteristics of real

radargrams such as noise and Doppler effect. Along with quantitative tests, a qualitative test was undertaken to evaluate the produced radargrams, which resulted in synthetic and real radargrams being indistinguishable to the observers. After training an HED network for tracing englacial boundaries, it was observed that network's results improved when trained with actual and synthetic data. Due to their inability to reproduce nuances of real radargrams, training solely on synthetic data cannot produce high quality results. The best results came from the experiment with an equal number of synthetic and real radargrams for both

training and testing, and the lowest came from a combination that was trained on synthetic radargrams and tested on real ones. An important conclusion is that while synthetic data are visually and statistically similar to real data, they fail to represent the physics.

**Cai et al. (2020)** developed a model performing pixel-level classification by using a deep convolutional classifier with the goal to classify the englacial regions of ice sheets. The network architecture is composed of filter processing and an

encoder–decoder. For training and validation, radar products provided by CReSIS are used. Their initial stage is to remove the noise by using bilateral filtering. The encoder contains atrous spatial pyramid pooling (ASPP), whose function is to improve classification and obtain multi-scale feature extraction and the backbone network is the ResNEt3 (He et al., 2015b). In the decoder, high and low level features are proportionally concatenated so that the low level edge information can be used. The model manages to yield the same F-measure as Kamangir et al. (2018).

**Ibikunle et al. (2020)** used a multi-class neural network and the iterative "row-block-column" approach and present their preliminary results for automatically tracing the snow layer boundaries from snow radargrams of CReSIS. They simulated the training data, which is considered to not capture all the details of real radargrams in snow, but nevertheless show some promise for training. The iterative approach starts from the known IRH using the rows below it and continues in the direction of deeper snow boundaries. Using the same iterative, **Ibikunle et al. (2023)** enhanced their previous work and trained a multi-

class classification deep neural network to identify the index for the next layer boundary in each column, using a RowBLock



approach. A "columnpatch", which is the *N* neighbouring columns of the current column under the iterative process, is used to enhance robustness and spatial awareness of the algorithm. Since this method is grounded in selecting a number of rows beneath each known IRH, it is methodologically inappropriate for deeper boundaries and suitable only for snow layer boundaries. Firstly, the deeper in the radargram, the steeper variability in the geometry of the boundaries appears. Secondly, owing to layer compaction from overburden pressure, the deeper IRHs are closer to each other, making it much more difficult to choose a suitable number for each row block.

Acknowledging that automatically tracing englacial IRHs is much more challenging than detecting the ice–base boundary, **Wang et al. (2020)** implement a CNN to detect the surface boundary, estimate layer thickness, and the number of visible IRHs. They frame the task as a tiered segmentation problem (Felzenszwalb and Veksler, 2010), and the aim is to solve this tiered labelling with the use of DL methods. An RNN refines the boundaries of the IRHs below the surface. This is a more general version of tiered segmentation as in this case the number of tiers (labels) is not known. For this reason, this new approach is one that could handle large and unknown number of labels.

**Yari et al. (2020)** implement a multi-scale learning process with an edge detection function as well as several side outputs at each level. Their aim was to track shallow IRHs in radargrams from different radar systems. There are two training paths: one uses the output of Koenig et al. (2016) for training which were later corrected by a human expert; the other path is training from scratch. They importantly indicate that tracing IRHs is a more complicated task compared to tracing the air–ice and ice–base boundaries. The second method of training gives the best results, as the shortcomings of the first method was most probably due to presence of noise.

**Varshney et al. (2020)** applied a fully convolutional Network (FCN) and perform a multi-class semantic segmentation on snow radargrams to infer the thickness of snow layers. The layers are detected and their thickness are separately calculated using an automated technique. They noted that so far methods that employed DL were focused on binary detection of each pixel, while in their approach they focus on separately detecting each IRH uniquely. The algorithm uses pixel-wise annotations for each IRH for training. This means that each pixel has a label stating if it belongs to a horizon or another. As ground truth, the output of Koenig et al. (2016) is utilized. Since the goal is to find complete snow layers, as opposed to layer boundaries, the regions of the radargrams where the labels follow a discontinuous layer boundary are cropped out. One of the disadvantages of this method is that by cropping out regions with incomplete training labels, about 50% of the training dataset is diminished. As this is done manually, it is a substantial task that requires a lot of time from the operator to perform. Three different architectures for semantic segmentation are trained: U-Net (Ronneberger et al., 2015), PSPNet (Zhao et al., 2016), and DeepLabv3+ (Chen et al., 2018), to produce multi-class results. They concluded that DeepLabv3+ yields the best results in terms of both spatial information and global textual prior, attributed to its more advanced architecture. **Varshney et al. (2021a)** uses a CNN regression network to estimate snow layer thickness. For training, the traced boundaries of Koenig et al. (2016) and the pre-processing product of Varshney et al. (2020) which are cropped for incomplete IRH have been employed. This study is built up on the work of (Varshney et al., 2020) in which the snow layer boundaries were traced and used as ground truth. **Rahnemoonfar et al. (2021)** employed CNN architectures for multi-scale learning and HED in order to perform automatic tracing of IRHs. They selected this approach to overcome the noisy nature of the radargrams and to extract both local





and global features. Training data comes from the output of Onana et al. (2015), corrected by a human expert in a previous study (Koenig et al., 2016). For another experiment, synthetic radargrams are used. A number of deep neural network architectures were employed for training, such as AlexNet (Krizhevsky et al., 2017b), VGG-net (Simonyan and Zisserman, 2014), GoogLeNet (Szegedy et al., 2014), and Resnet (He et al., 2015a). It was concluded that the best results come from training the multi-scale model on real data. Other experiments i.e. training the model on augmented benchmark dataset, and training the model on synthetic data, and traditional edge operator (Canny) results do not show comparatively satisfactory outcomes. They highlighted that while numerous renowned DL methodologies exhibit remarkable performance when applied to optical imagery, their efficacy tends to wane significantly when extended to non-optical domains, such as radargrams, as discussed in the literature (Heaven, 2019). Another important conclusion from this paper is that transfer learning is not an optimal solution for radargrams and training from scratch leads to significantly superior outcomes, the drawback being the necessity of a large number of annotated data. Further work on producing synthetic data with higher quality that could represent real data more realistically might be a suitable solution. **Varshney et al. (2021b)** also used FCN to trace snow layer boundaries. This publication is a combination of Varshney et al. (2020) and Rahnemoonfar et al. (2021), also employing the same architectures as the latter. Their training data come from Koenig et al. (2016), and are labelled as a separate class. Cropped radargrams were used as in Varshney et al. (2020), so that only continuous picked horizons prevail to reduce the effects of noise, also similar to Varshney et al. (2020), they form semantic layers from the cropped radargrams and the annotated layer boundaries. **Varshney et al. (2022)** used the regional Modèle Atmosphérique Régional (MAR) (Fettweis, 2007) to compute the surface mass balance for the past 30 years to match the corresponding stratigraphy seen in the snow radargrams, corrected with density variations in depth using Herron and Langway (1980) densification. Radargrams with manually detected and corrected layer boundaries are selected for training. These annotated labels and labels taken from MAR in the regression model of (Varshney et al., 2021a) are accounted for to estimate, learn and predict the thickness of the snow layers. **Zalatan and Rahnemoonfar (2023)** consider that two weaknesses of CNN are sensitivity to noise and their inability to perform spatio-temporal tasks. Therefore, to trace snow layer boundaries from radargrams, they utilize a recurrent graph convolutional neural network (GCN Kipf and Welling, 2016). It converts the thickness of the snow layers into temporal graphs and use them as inputs. Similar to the RNN, a long short-term memory (LSTM) is also implemented. One improvement of the model is to explain its capabilities to a higher number of traced IRHs. The model is capable of detecting five shallow IRHs. **Varshney et al. (2023)** developed a wavelet-based multi-scale DL architecture to detect layers in firn. The multi-scale backbone architecture is the one of Yari et al. (2019); Rahnemoonfar et al. (2021). They set up two multi-scale CNNs which differ in the point where to apply the wavelet transform, i.e. to the image (so-called WaveNEt) or to each side output (so-called Skip-WaveNet). These two architectures are each combined with three popular discrete wavelets. They use the OID dataset for training, and compare their results of inferred firn layer thickness with some in-situ stake measurements taken ~16 km away from some parts of the radar profile. Six experiments were performed with each of the two architectures and three wavelet transforms. Overall, they report some enhancement in detection from the Skip-WaveNet architecture with dmey discrete wavelet (Daubechies, 1992).

   **Donini et al. (2021)** took advantage of unsupervised learning for subsurface geological target information extraction. The choice of unsupervised learning is both to utilize the capability of DL (as opposed to classical machine learning and statistical





analyses) and to overcome the lack of sufficient labelled data. It is worth noting that this work is among the ones which try to detect subsurface targets in radargrams, and not necessarily layer boundaries. The method entails three steps: i) generation of a coarse segmentation map of each radargram, ii) refining this map using DL iii) further analysis of these features. Each radargram is considered to consist of two classes, i.e. background and target, and the target class is further subdivided using

the unsupervised method. The first step, after patch extraction of the radargrams, is to use the method of Ferro and Bruzzone (2012) to fit distributions to each patch to sort them into background and features classes. Similar to Garcia et al. (2021), the selected unsupervised architecture is the W-Net (Xia and Kulis, 2017). In this network, each convolutional layer learns semantic features from the background class. The normalized reconstructed error maps are exploited to gain information on the two classes. This error is small for background and large for features. The MARSIS data from Mars' South Pole is used for

evaluation. They are more successful in the **Garcia et al. (2021)** attempt to segment the radargrams into five classes (air, ice sheet, ice shelf and crevasses, base, and noise). Since Ilisei and Bruzzone (2015) required hand-labelled radargrams and input features such as entropy, and Kullback-Leibler distance (Lin, 1991), they replaced SVMs by more modern DL techniques such as CNN. However, the CNNs also require a large number of annotated training data and there are not many available annotated radargrams for training. This challenge was overcome by using a W-net (Xia and Kulis, 2017), an unsupervised, fully

convolutional autoencoder architecture. Overall, they were more successful in detecting the air and noise classes compared to others. Moving away from unsupervised learning, **García et al. (2021)** attempt to reach the same objectives as Garcia et al. (2021) by using an autoencoder CNN, and to overcome the lack of labelled training data, a pre-trained network in another domain (ImageNet (Russakovsky et al., 2015)) was used. The role of transfer learning is to adapt the pre-trained CNN weights in the radar sounder domain. Domain adaptation is established by addition of a convolutional layer in the upstream of the CNN,

so that the model can handle different properties of a radargram. Furthermore, the pre-trained CNN's last layers are removed so that is it not too specifically tailored to the features of the source domain. The kernel of the architecture is MobileNet V2 (Sandler et al., 2018). The results show the network's robustness and that the learned features from the source domain are extendable to radar sounder domain. The network seems to correctly distinguish between ice sheet and ice shelf, nevertheless the results without fine-tuning seem to be of poor quality. In another attempt to obtain the locations of geological targets, **Donini**

**et al. (2022c)** implemented a U-net architecture including ASPP for controlling the resolution of features used for training, as introduced by Guo et al. (2020). The training is a two-step process. In the first step, the network initializes parameters and extracts features by minimizing the loss between input and output. In the second step, a supervised training takes place to yield the final classes. Data augmentation has been performed to increase the data five-fold. The segmentation takes places within multiple scales and the ASPP expands the receptive feature space to host more context while using less number of parameters.

The initial weights are taken from pre-training to optimize the loss function. The U-net is also equipped with an attention gate which has the function to discard irrelevant information through the skip connections. Additionally, morphological filters are convolved to refine the outcome of the U-net. The final product of this method is a map that segments each radargram into four classes: i) englacial layers, ii) basal ice, iii) base, and iv) noise. The best outcome has been the detection of basal ice and noise classes. The strive for segmenting subsurface regions in a radargram continued by **García et al. (2023)** who published their

work on a weakly supervised approach to segment radar products, a combination of the methods in Garcia et al. (2021) and



García et al. (2021) made to segment a revised version of the classes of Donini et al. (2022c). The main driver for the choice of transfer learning was to overcome the insufficiency of labelled data and the class imbalance which is the case for radargrams. Regions of each radargram are classified into five discrete classes, i.e. free space, noise, inland ice, crevasses and englacial features and base. Two transfer learning methods are presented and the method comprises of two designs. One network (a light-weight CNN) is pre-trained in a domain other than radargrams in a supervised manner, following the idea of (Garcia et al., 2021). The second approach is a very deep network, pre-trained with radar sounder radargrams in an unsupervised manner (García et al., 2021). A set of four experiments was performed, each with and without data augmentation, transfer learning and different sizes of pre-training sets. Their results show a proof for the effectiveness of transfer learning and both designs show good performance. However, the first one (a light-weight CNN pre-trained with non-radargrams) seems to be the better choice on the grounds of being light-weight and not requiring much time and computation power, even though the deeper network's accuracy is slightly higher. Data augmentation shows enhanced accuracy for one of the datasets, the smaller and more simple one (5% higher for MCoRDSv1 data).

To automatically calculate ice-sheet thickness, **Cai et al. (2022)** constructed a method to automatically trace ice sheet surface and base boundaries. To this end, they make use of a multi-scale fusion network (MFFN) and a multi-scale convolution (MSM) module learns these representations. As the name suggests, in this method the provided ground-truth supervises the outcome at every stage. They employ a CE-focal loss function, which is a combination of cross-entropy of weight balance and cross-entropy with modulating factor. It is shown that this loss function is an optimal way to counterfeit the class imbalance of the data. 4700 and 974 radargrams were used for training and validation respectively, cropped in $400 \times 400$ pixels. The model is compared with other available DL methods such as VGG (Simonyan and Zisserman, 2014), HED, PiDiNET (Su et al., 2021), DexiNed (Soria et al., 2021) and some others and showed better precision and recall. However, they observed that the proposed method is not very efficient in detecting faint boundaries.

**Dong et al. (2022)** introduced EisNet, a deep neural network fusion architecture to trace different types of IRHs, i.e. ice–base and IRHs. EisNet is composed of a convolutional discriminator and two convolution extractors. To train the extractors, they produced a synthetic dataset, and the discriminator is pre-trained with the synthetic data while the real data is used for transfer learning. The discriminator categorizes the radargram fed to it according to presence or absence of the basal interface (ice–base boundary). If the radargram contains this interface, it goes through a convolutional encoder–decoder (same as U-net (Ronneberger et al., 2015)). And in the case of no basal interface, another convolutional encoder–decoder is chosen to perform the extraction. In testing with the real data, the shallow part near the surface and the last 522 time intervals are discarded, because they lack discernible features. Overall, they report some shortcomings in the results which are attributed to noise and obscure interface features. **Tang et al. (2022)** propose a fusion method that combines filtering methods for both noise reduction and horizon extraction. They propose a combination of these methods which is expected to enhance the quality of noise removal and horizon tracing. The filters for noise removal are Karhunen–Loeve (Karhunen, 1946; Loève, 1960), frequency–wave number domain (F–K), and F–K migration (Loève, 1977), and the neural network for noise removal is the DnCNN (Zhang et al., 2017). The basic idea is that this network is based on residual learning and learns from the residual distribution of the noise in the radargram and this distribution is subtracted from the radargram to remove the noise. After





seeing a low quality of result for horizons tracing using a U-net architecture, the idea is to merge the classical F–K and KL filtering methods for noise removal and EisNet of Dong et al. (2022) for the horizon extraction. Nevertheless, although lowering noise levels, the results of this fusion method seem to contain some discontinuity and require some post-processing for the extraction of IRHs.

A combination of CNN and probability graphical model (PGM) (Jordan, 2004; Ghahramani, 2015) was tested by **Liu-Schiaffini et al. (2022)** to automatically identify the ice–base interface. While DL networks have the ability to learn the features from complex data, they fail to understand explicit structures. On the contrary, PGMs are able to capture such structures through encoding relationships among random variables. The method has two steps: a CNN to extract the overall and large-scale structure of the ice–base interface and a continuous conditional random field (CCRF, Qin et al., 2008), a type of PGM, in

order to produce the detailed structure and distinguish the nadir ice–bed interface at a smaller scale. This two-stage learning has been performed on the University of Texas Institute for Geophysics high-capability radar sounder (HiCARS) (Schroeder et al., 2013) radargrams from East Antarctica. Quantitatively, the combined CNN + CCRF model has similar results as only the CNN, but the combined model captures different features on smaller scales. Qualitatively, the CNN + CCRF yields more continuous reflections compared to CNN's results. Thanks to its stochastic components, the combined model captures uncertainties more

effectively. Also, CNN + CCRF results oscillate much less in the vertical dimension (along the interface) compared to both CNN and human-annotated results. It was noticed that human annotations of ice–bedrock interface is misleadingly influenced by off-nadir reflections, due to noisy returns, causing humans to annotate side reflectors mistakenly instead of interface itself.

Transformer-based models, known for their ability to capture long-range sequential context and global spatial contextual prior, in contrast to traditional CNN-based methods that primarily capture local spatial context were tested on radar sounder

data as well by **Ghosh and Bovolo (2022a, b)**. Subsurface targets are characterized using a hybrid TransUNet-TransFuse architectural framework, TransSounder. The study uses the MCoRDS dataset and compares the performance of TransSounder with other architectures such as TransUNet (Chen et al., 2021), TransFuse (Zhang et al., 2021b), and U-Net. The results show that TransSounder achieved the highest overall accuracy and effectively captures global and local spatial contextual features in radargrams. In another unsupervised attempt, **Ghosh and Bovolo (2023a, b)** built upon a previous work on Self-Supervised

Transformer (Hamilton et al., 2022) called Self-Supervised Transformer with Energy-based Graph Optimization (STEGO). This work is to test the capability of this network. It was enhanced by incorporating an Expansive Network to increase the resolution of detailed visual features from a middle segmentation stage to a reconstructed signal. Each radargram is classified to three target classes/regions: ice, bedrock and noise. The modification over STEGO (Hamilton et al., 2022) is adding an expansive network in order to up-sample dense features from an intermediate segmentation head. Although not as good as

other architectures, such as U-net or that of Ghosh and Bovolo (2022b) they are more effective than the original STEGO, keeping in mind that it was not designed for radar sounder data. Overall, this shows that without any training data, it was possible to extract some meaningful semantics from the radargrams.

**Atefeh Jebeli et al. (2023)** aimed at annotating bed and surface horizons by using a two-step semi-supervised annotation (TSSA) approach which works on the backbone of ARESELP by Xiong et al. (2017) for producing training data. A U-net

architecture is trained on these data to trace the target boundaries. The U-net is combined with pre-trained VGG19 (Simonyan



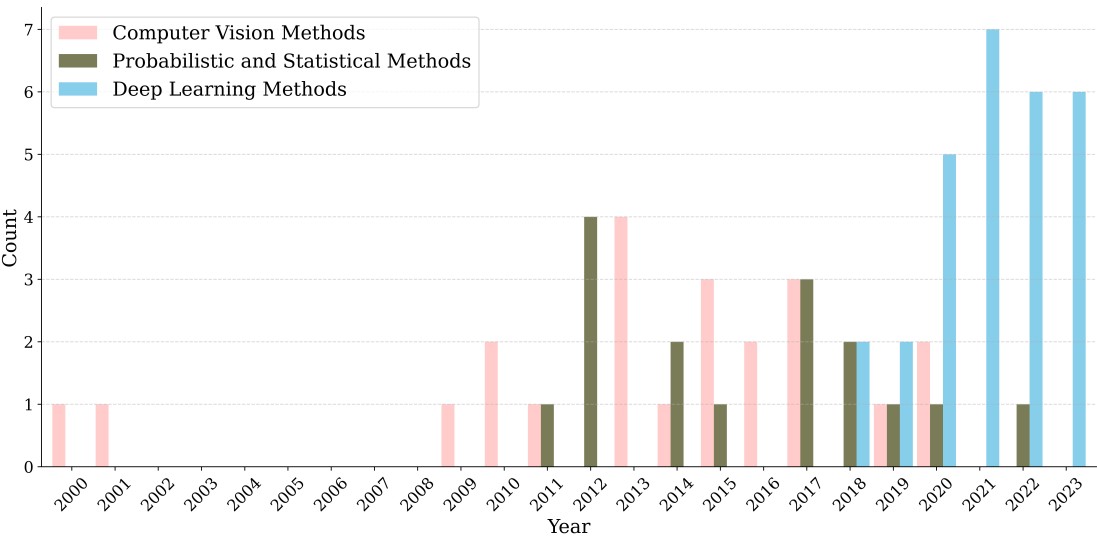

**Figure 7.** Temporal distribution of research methods: Count refers to the number of publications within this field for each year in the respective category (for brevity again restricted to the time after 2000).

and Zisserman, 2014) and Inception (Delibasoglu and Cetin, 2020) architectures, also taking advantage of data augmentation to enhance the training outcome. The designed experiments are a combination of the three architectures augmentation methods. They report that U-Net + Inception with data augmentation yield the best results.

# 5    Discussion

From the timeline of applications to map englacial stratigraphy, we note that methods started off from early-stage computer vision techniques, such as image processing, active contour, Hough, and Radon transforms, and subsequently, a shift occurred towards solutions rooted in probabilistic and statistical approaches, with methods such as HMM emerging as prominent choices. In recent years, the noticeable trend has transitioned towards the adoption of DL techniques. This trend towards the use of deep learning-based methods is evident from Fig. 7, depicting a categorization of methods that we presented in the previous section.

The increasing application of DL techniques across various fields of applications (Minaee et al., 2020; Chen et al., 2022a; Sarker, 2021) further emphasizes the likelihood of advancing methods and improving results for radargram segmentation and IRH mapping.

From an application-oriented point of view, there has been a notable transition from methods primarily focused on tracing the air–ice and ice–base boundaries in radargrams (e.g., Gifford et al., 2010; Rahnemoonfar et al., 2017a), mostly for estimating

ice thickness and basal roughness, to the increasingly emphasized task of accurately tracing englacial layers. However, most of latter have been employed on layers and horizons in snow and firn and shallower IRHs (e.g., Varshney et al., 2022; Yari et al., 2020), but much less often tested on deep IRHs. Table 2 shows a classification of the published work based on both the



| which IRH<br><br>Category | Surface and<br>bed reflection | Snow / Firn | Deep IRHs | Regions / Segments |
|---|---|---|---|---|
| **Computer vision methods** | Gades et al. 2000<br>Gifford et al. 2010<br>Reid et al. 2010<br>Mitchell et al. 2013b<br>Rahnemoonfar et al. 2017a<br>Rahnemoonfar et al. 2017b<br>Goldberg et al. 2020 | Mitchell et al. 2013a<br>Onana et al. 2015<br>Koenig et al. 2016 | Fahnestock et al. 2001<br>Freeman et al. 2010<br>Sime et al. 2011<br>Karlsson et al. 2013<br>Panton 2014<br>Dossi et al. 2015<br>McGregor et al. 2015<br>Xiong et al. 2016<br>Xiong et al. 2017<br>Lines et al. 2019<br>Delf et al. 2020 | Ferro and Bruzzone 2013 |
| **Probabilistic and Statistical methods** | Crandall et al. 2012<br>Ilsei et al. 2012<br>Lee et al. 2014<br>Xu et al. 2017<br>Berger et al. 2018 | Keeler et al. 2020 | Smock and Wilson 2012 | Ferro and Bruzzone 2011<br>Ferro and Bruzzone 2012<br>Ilsei et al. 2014<br>Ilsei et al. 2015<br>Carrer and Bruzzone 2017<br>Khodadadzadeh et al. 2017<br>Donini et al. 2018<br>Donini et al. 2019<br>Donini et al. 2021b |
| **Deep learning methods** | Kamangir et al. 2018<br>Xu et al. 2018<br>Rahnemoonfar et al. 2019<br>Cai et al. 2020 (+)<br>Cai et al. 2022<br>Dong et al. 2022 (+)<br>Liu-Schiaffini et al. 2022<br>Jebeli et al. 2023 | Yari et al. 2019<br>Yari et al. 2020<br>Ibikunle et al. 2020<br>Vashney et al. 2020<br>Rahnemoonfar et al. 2021<br>Varshney et al. 2021<br>Varshney et al. 2021<br>Varshney et al. 2022<br>Ibikunle et al. 2023<br>Zalatan et al. 2023<br>Varshney et al. 2023 | Wang et al. 2020<br>Dong et al. 2022 (+)<br>Tang et al. 2022 | Cai et al. 2020 (+)<br>Garcia et al. 2021a<br>Garcia et al. 2021b<br>Donini et al. 2021a<br>Donini et al. 2021c<br>Ghosh et al. 2022<br>Ghosh et al. 2023<br>Garcia et al. 2023 |

**Table 2.** Presentation of published works both by the mapped IRHs or region, and the category of methods used.

category of the used methods as well as IRH, boundary, or regions that those published works attempted to trace. One of the points that can be clearly noticed in this table is the inadequacy of published models for tracing deep IRHs.

The abundance and closeness of features comprising IRHs in a radargram, often in the order of a couple of wavelengths, along with possible merging and discontinuities of horizons, are the main challenges in mapping IRHs. In most other feature-extraction and image-segmentation applications in glaciology, the features and edges (if multiple are present) are usually not located at such short distances from each other as IRHs could be, for example in glacier grounding lines delineation (Mohajerani et al., 2021). This remains a complex task for any algorithm to detect different features as close as a few pixels from each other

and not merge them when there is no merging to be done, but to separately trace them in case of discontinuities, which are another phenomenon that obstructs mapping attempts (Varshney et al., 2021b). They could emerge as a result of tracing algorithm's shortcomings and inefficient tracing capabilities. Naturally-occurring discontinuities could be due to merging of



horizons as a result of decreasing separation of the actual (physical) reflectors at depth, usually below half a wavelength resp. bandwidth limit, e.g. by compaction for firn layers (Varshney et al., 2021b), ice–dynamic strain thinning (Conway et al., 2002), unconformities caused by changing environmental conditions at the surface (Siegert et al., 2004), heterogeneity of microstructures (Koenig et al., 2016) or simply decreasing surface accumulation along a profile, all the way to an erosive regime. Moreover, as the undulation of IRHs in ice sheets become more similar to basal topology in the deeper regions, steeper features are more present in the deeper parts of the ice sheet (Winter et al., 2019). Such steep features are significantly more challenging to, first, image by RES systems, and second, be differentiated in their structure, even for experts.

The DL approaches which have been applied to mapping and segmenting IRHs are among common architectures and methods that have already demonstrated success in other domains (Choudhary et al., 2022; Emek Soylu et al., 2023; Siddique et al., 2021). However, their inability to yield more satisfactory results in mapping stratigraphy so far can be associated with a number of reasons. One is the aforementioned complexity of radargrams related to the closeness of features and horizons to each other. In many cases, if a method maps an IRH even a few pixels in a wrong spatial direction, this could be a false prediction and there could be another horizon with a different age at this location. Another possible obstruction for CNN methods is class imbalance, meaning that the number of pixels corresponding to IRHs are far fewer compared to the number of pixels corresponding to the background (non-IRH) class. Unlike tasks such as mapping glacier calving front (Mohajerani et al., 2021) in which similar methodologies are performed on images with mostly one boundary in the entire image, radargrams suffer from a much higher level of imbalance between classes. Some solutions have been proposed to overcome this (Cai et al., 2022; García et al., 2023; Moqadam et al., 2024). On the other hand, CNNs are known to be data-hungry and require a large number of annotated data for training (Karimi et al., 2020; Li et al., 2019). Obviously, there seems to be a lack of a large number of annotated data for mapping deep IRHs to be used as training. In most glaciological investigations for which radargrams are annotated, only a handful of IRHs are traced, resulting in a majority of IRHs remaining unlabeled. Such annotations are unproductive for training purposes and confuse the CNN (Tang et al., 2019; Hyun et al., 2020), as some features are traced while other similar ones left untraced. In fact, this could have a counter-productive effect for CNN training (Tang et al., 2019). The useful annotations, which would be ideal for training, are radargrams in which all, or at least a large majority, of IRHs are annotated. Fig. 8 depicts an example of this.

Even though there have been numerous semi-automatic and automatic attempts on tracing surface and basal reflections, and a substantial number of attempts on snow and firn boundaries, not much work has been done in developing methods which can successfully trace deep IRHs. This lack of methods can be seen in section 4 and Table 1. There are ongoing attempts to map deep reflections along with ongoing efforts to apply machine learning, complemented by transfer approaches, to inter- and extrapolate layer characteristics across gaps (Bente and Marchant, 2024).

## 6 Conclusion and Outlook

While each of the investigations and developed algorithms presented in this review contributed to expanding comprehension of mapping the internal structure of ice sheets, the rising incidence of inconclusive and partial results across a number of datasets



Radargram          Corresponding mask

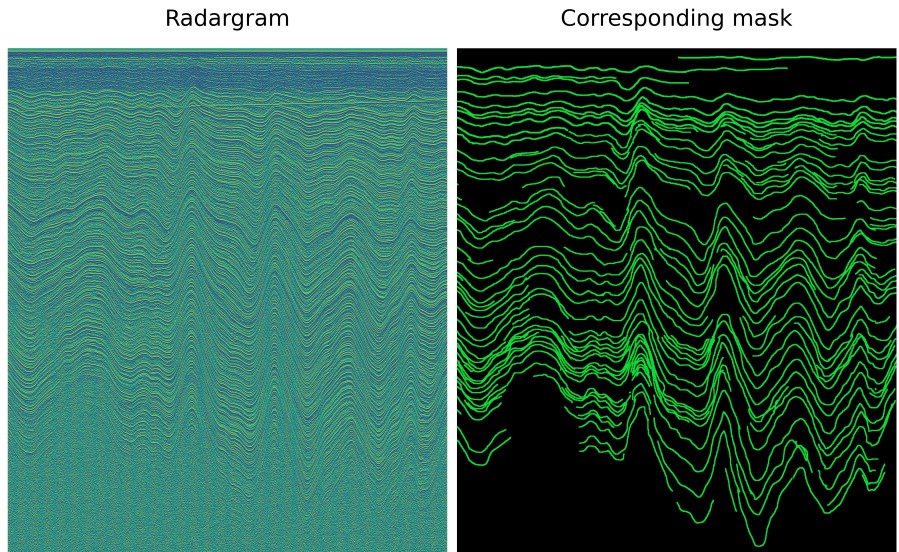

**Figure 8.** A section of a radargram (Steinhage et al., 2013) and its corresponding entirely-annotated mask.

calls for a thorough and all-encompassing approach for evaluating the methodologies and findings. Consequently, this review aims to provide a contemporary overview of advancements in this field of research over the past two decades. We assessed the employed methodologies and proposed techniques and outcomes, and delineated emerging trends and prospective trajectories for future research.

Due to the vastness of Antarctic ice sheet, there is still much data that has not yet been adequately or sufficiently analysed. Moreover, a comprehensive mapped englacial stratigraphy of the Greenland ice sheet, though major stratigraphy is available from manual tracing (MacGregor et al., 2015), and stratigraphy of mountain glaciers are still points of interest as well. Therefore, the need for a fully automated IRH tracing tool is more than evident. Using such methods, we could increase our knowledge of the age–depth relationship from ice core sites, which are point measurement, to larger spatial scales and even

synchronize age–depth relationships between ice-core sites in much more details than currently feasible (e.g., Lilien et al., 2021) and have a complete picture for larger areas between drill sites. An automated method also facilitates imaging regions of past ice-dynamic changes and varying environmental conditions. This would greatly contribute to our understanding of ice sheet's and glacier's presently visible englacial structure.

 We observe that DL-based methods have taken the lead towards solving the IRH tracing task in the past couple of years.
This can be related to the rapid advances and increasing capabilities that the DL field has been experiencing. Nevertheless, interpreting radargrams poses significant challenges, even for experienced human operators. Radargrams are often noisy and contain a multitude of closely-packed features. This complexity arises from their representation over vast depth and length scales, in particular with respect to the radar wavelengths, spanning up to several kilometers vertically and hundreds of kilometers in flight direction. Consequently, given the broad range of conditions covered by the radar, imaged features and IRHs



within radargrams can appear discontinuous, merge with each other, or be inadequately represented. These complexities make it difficult for both humans and algorithms to accurately annotate and determine the fate of IRHs and the ice–base boundary. The data acquired by radar systems and the presentation of radargrams have not changed for the past couple of decades. Although system resolution, signal-to-noise ratio and signal penetration depth increased, the visualization of IRHs did not change considerably. Current research on mapping englacial stratigraphy continues to grapple with the complexity of the task, focusing

on fundamental challenges rather than solely refining existing algorithms and methodologies.

The described challenges for automated tracing methods to work properly could be facilitated by establishing a complete pipeline for tracing IRHs, which was trained on a benchmark dataset. Such a dataset would provide the advantage that it will allow a more objective, quantifiable and hence rigorous comparison among different approaches. It has been discussed that performing comparison among different segmentation and edge detection methods would not be a valid comparison since

results could depend on specific criteria of implementation (Kaspersen et al., 2001). Therefore, method evaluation on the same dataset would be a valid and quantitative comparison. Since producing datasets such as Fig. 8 are quite cumbersome, synthetic datasets that represent the complex features and possibilities of phenomena within radargrams could be the way to go. In addition to complete labelling efforts on measured datasets, synthetic approaches can provide progress towards an intermediate solution as well. There have been a number of synthetic datasets so far (Rahnemoonfar et al., 2019; Dong et al.,

2022), but they have not yet been widely considered and used by the community. However, synthetic datasets cannot replace actual radargrams, for their lack of characteristics of real radar systems such as noise, IRH discontinuities, and merging, to name only a few. Nevertheless, since such obstructions are a matter of resources, it is highly probable that methods capable of automatically tracing IRHs will be developed in the near future.

*Author contributions.*  HM and OE conceptualized the study. HM wrote the manuscript draft. OE contributed to revising, with a focus on the

glaciological context, and editing of the manuscript.

*Competing interests.*  The contact author has declared that neither they nor the co-author have any competing interests. OE acted as an editor for TC.

*Disclaimer.*  Tools used: For improving writing style as non-native speaker the first author used the thesaurus (also known as dictionary of synonyms) (www.thesaurus.com) for looking up synonyms and ChatGPT (chatgpt.com). The usage of ChatGPT was limited to obtaining

alternatives for adverbs and prepositions such as "due to", "thus" and "according to" for less than 30 cases. No sentences were produced, paraphrased or rephrased by this or any other tools. Similarly, no AI-based tool was used to create or polish content or paraphrase paragraphs.



*Acknowledgements.* The first author HM is funded through the Helmholtz School for Marine Data Science (MarDATA), Grant No. HIDSS-0005. We greatly acknowledge the discussions with Claudius Zelenka (Univ. Kiel), Nicholas Stoll (Univ. Venice) and Steven Franke (Univ. Tübingen & AWI) during the development phase of this manuscript. We also acknowledge other discussions on this topic within the Glaciology section at AWI as well as the Glaciology group at the University of Tübingen.



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
