# Peer review of "Review article: Feature tracing in radio-echo sounding products of terrestrial ice sheets and planetary bodies"

_EGUsphere, 2024_

## Referee Comment (RC2)

Review of the article 'Review article: Feature tracing in radio-echo sounding products of terrestrial ice sheets and planetary bodies' by Hameed Moqadam and Olaf Eisen.

The paper proposes a review of the literature methods for detecting internal reflection horizons related to englacial stratigraphy and several glaciological applications to monitor the cryosphere. The structure of the paper is very complex and fragmented, making the sections contain a lot of repetitions from other sections. The English is good, but sometimes very informal, and there are typos, so it should be improved to match the journal's quality. Here are the general comments on the paper:

The paper's aim is unclear, whether to review the automatic methods for radargram analysis or to examine IRH/ice layer tracing. The paper focuses on the importance of IHR tracing (abstract, intro, and background sections). Still, it also presents methods for target detection and segmentation that are unrelated to edge/IHR/layer detection.

It is also unclear what criterion is used to select the methods the paper analyzes (the abstract considers those applied to RES data, and sec 4 claims only those to analyze radargrams). What about methods for GPR data?

The paper's headings are very confusing and do not give an overview of the section's content.

General comments on Sec 3. The criterion used to divide the data analysis methodology into different sections (sec 3.1-3.12) is unclear. Considering the large number of methods, I would expect a clustering with a clear logic. Moreover, the writing should be improved to increase fluidity. Further, the sections miss an analysis of the pros and cons of the methodologies, e.g., which are the ideal and worst conditions, which are the method's core hypothesis, how should be the radar data preprocessed, how are the radar data non-ideality handled (e.g., speckle), which are the limitations, and on which data was the method validated (e.g., planetary, terrestrial, high or low spatial resolution, adaptability to other data).

Moreover, I would expect more figures (e.g., flow charts) and formulas to understand each method's details better. Finally, I am very confused about whether the paper aims to show the methodologies for radargram automated analysis, IHR detection, or layer tracing. The paper focuses on the importance of IHR and related applications, while section 3 presents a list of automated methods used for analyzing radargrams.

General comment on Sec 4. It is unclear why Sec 3 and Sec 4 are divided and not merged, given the lack of details of Sec 3 and the repetition of concepts already explained in Sec 4. Moreover, consider using the taxonomy of sec 4 also for sec 3. Instead of Sec 3 and Sec 4, I suggest having three sections with the taxonomy of 4.1,4.2, and 4.3 that present the computer vision theory (an improved version of what is in sec3) and the application to the radar data (an improved version of what is in sec4). Moreover, the session lacks details, formulas, and images to understand how the methods work. In general, the methods' descriptions lack a discussion on the pros and cons, limitations, and if/how they can be improved. Also, the connection between paragraphs is missing, and the section looks like a list of methods without any critical analysis.

Further, the methods are applied to very different types of data (SHARAD, MCoRDS, HiCARS, accumulation data) without describing the properties of the data in terms of noise, resolutions, acquisition geometry, and so on. Consider adding information on the datasets used and their

characteristics (maybe in sec 2.2). Further, the paper's aim is unclear (IRH detection or segmentation or automatic analysis of radar data). At the end of each section (i.e., 4.1, 4.2, and 4.3), the methodologies should be discussed to understand the limitations that forced the development of novel techniques.

General comment on Sec 4.3. This section moves from segmentation to layer detection. Consider fixing an aim to the paper and sticking to it. There are few indications on the data types used to validate the methods (and how the data-specific characteristics are handled, including noise and resolutions). Moreover, consider dividing the section into subsections (e.g., DL for layer detection, DL for semantic segmentation). The section lacks i) a critical analysis of the methods, ii) information on the type of data the methods are validated, and iii) information on the computational load (given the high computational cost of the DL algorithm).

Sec 5, discussion. To better understand the validity and differences of the methods, I would expect a qualitative/quantitative comparison.

Sec 6. The paper's aim is changed to 'Consequently, this review aims to provide a contemporary overview of advancements in this field of research over the past two decades.' The section has a lot of repetition compared to sec 4 and 5. Consider merging the sections. Finally, the section misses the expected future directions in the IRH detection/ automatic analysis of radargrams.

Below are the detailed comments (Pg stands for page and Ln/ln for line).

Pg 2, ln 55, Antarctica is larger than Greenland but poorer in terms of data. There are large areas without any acquisitions.

Sec 2.1 repeats concepts already defined in the introduction. Given the length of the paper, consider removing the repetitions. Ln 96-100 lack details on the IHR formation and give partial information through examples. Consider improving this paragraph.

Pg 4, ln 103. Consider adding references for z-scope radargrams, e.g., 'Schroeder, Dustin M., et al. "Radiometric analysis of digitized Z-scope records in archival radar sounding film." Journal of Glaciology 68.270 (2022): 733-740.'

Sec 2.2, line 106. Consider removing the repetitions on the reflection generation: 'the aforementioned characteristics such as presence of impurities, acids, and changes in ice-crystal orientation cause reflections, and when they are laterally coherent, they appear as horizon'.

Sec 2.2, ln 107-110. Consider defining a radargram as a 2D matrix of $N_T$ traces and $N_S$ samples. The definition of the radargram pixels as indicating the power/amplitude is misleading. Consider that radargram may also be complex, meaning there is the amplitude and phase radargram or the imaginary and real radargram.

Figure 1.a and 1.b. Consider showing the figures in dB for a better visualization. Moreover, figs 1.a-f lack the x and y axes ticks and labels. Is the x axis of fig 1.a valid also for fig 1.b-f? If so, clarify it in the caption. Moreover, clarify that the radargrams are presumed in the caption.

Sec 2.3, ln 132-133. Repetition of the ice layer generation.

The title of sec 2.4 is misleading as the section is about the information the englacial stratigraphy provides.

Sec3, ln 156. 'In this section, we briefly overview the methods applied to tracing IRH and segmenting radargrams.' This sentence is not in line with the previous section of the paper, as the focus was only on IHR and not on the identification of ice-sheet targets (i.e., segmentation). Consider improving the sentence.

Pg 7, ln 159. The sentence 'Given the versatile application of RES across various domains,' lacks a reference.

Sec 3 ln 160-165. Repetition of concepts described in section 2.4.

Pg 7, ln 166. This sentence 'Constructing an automated tracing method for RES encounters a significant challenge when dealing with closely spaced layers.' lacks justification.

Pg7, ln 179. What about folded or interrupted layers (e.g., those in the basal area)?

Pg 7, Ln 181. Horizontal or vertical resolution? Justification?

Pg 7, Ln 182. Large SNR should be small SNR?

Pg 8, ln 163-164. Repetition on the motivations for tracing IHR.

Pg8, ln 183. 'We will give a short summary of the methods that have been utilized in mapping and segmenting radargrams. The provided method summaries are intended to aid understanding of the timeline of methodologies in section 4, to make readers more aware of the underlying components or procedures of each method.' I am unsure how this sentence relates to the previous part of the section that focuses on tracing IHR.

Sec 3.1, cross-correlation and peak following. This section lacks the claim of the strong hypothesis that the ice stratigraphy is expected to be constant and horizontal (i.e., without abrupt changes in the steepness). What about the basal region, where shadows mask the reflections?

Sec 3.2, filter. The reference Ilisei and Bruzzone does not refer to canny filtering (it's a statistical analysis of subsurface targets). The same is true for (Freeman et al., 2010), which uses morphological filters and thresholding). Consider removing the reference and description to canny filter and be more general about filtering and thresholding. Moreover, considering the complexity and hypotheses needed for thresholding, I expect at least a sentence discussing it. Finally, this section should at least refer to speckle and how it is tackled, given that speckle can be seen as an abrupt change of intensity of the pixels and thus very visible to canny filter.

Sec 3.3. improve the computer vision description of Snake.

Section 3.4. What is this reference to 'cfd, 2019'? Missing justification to the sentence 'making it well-suited for the intricate analysis Radargrams'. It is unclear why this methodology is presented even if it has not been applied to radargrams. Consider removing the section.

Sec 3.5. Statistical analysis is not a method. This section should be improved as it does not detail how the analysis works. The section is also misleading as Rayleigh and Nakagami distributions are valid only under specific hypotheses (e.g., target analyzed and data type). Finally, this section concerns segmentation/target detection, not layer tracing. How is it related to the other sections?

Sec 3.6. Consider improving the English (too informal). I would not call a method based on an LPF and thresholding robust (... this method, although being robust...). What about the speckle? In general, the section lacks methodological details. Also, this is very similar to the method in 3.2; consider joining them or explaining the difference between the sections more clearly.

Sec 3.7. There is no reference to a paper analyzing radargrams. Consider removing the section.

Sec 3.8. Missing reference to Xiong, Siting, Jan-Peter Muller, and Raquel Caro Carretero. "A new method for automatically tracing englacial layers from MCoRDS data in NW Greenland." Remote Sensing 10.1 (2017): 43. And related works. Moreover, I expect formulas to help me better understand the methodologies.

Sec 3.9. Most of the papers cited are not related to radargram analysis.

Sec 3.10. The papers cited are not related to radargram analysis.

Sec 3.11. Consider improving the section on SVM. There is plenty of work to be done for segmenting radargrams. Moreover, the description of SVM, which is very wrong (e.g., SVM is also multiclass), should be greatly improved. The meaning of this sentence 'SVMs are able to identify the optimal surface, mitigating overfitting during training' is unclear. What is the optimal surface? Also, overfitting mostly depends on the training sample numbers and representativity. Missing the motivations that pushed the community to move to deep learning (i.e., the necessity of manually designing the features).

3.12. The section is confusing; for example, I would define deep learning before explaining representation learning. Consider improving the general description of deep learning and explaining why it is important for analyzing radargrams. In the sentence 'DL performs such tasks using multiple levels of non-linear modules, transforming these representations from raw to higher and more abstract level', what does 'representations' refer to? The sentence 'The advantage of semi-supervised learning is that it does not require a large amount of labeled data, but there is the danger of learning irrelevant features' is not true. The problem is overfitting. In the sentence 'The third class is unsupervised learning. As the name suggests, the learning procedure is based on finding representations without help of known targets', known targets should be labeled images/data. Finally, this section lacks a description of the main disadvantage of DL, which is the overfitting and poor generalization capability of the network with small labeled datasets.

Sec 3.12.1. Several complex concepts are not explained while describing the so-called general architecture of the neural network. In general, the paper shows inconsistent levels of detail in the description of computer vision concepts. Basic concepts (e.g., supervised, semisupervised, unsupervised) are described in detail, while the neural network layers are just mentioned. The most critical part of DL is the training and definition of the loss function. I would expect at least to mention how the network training works.

Considering their usage in 3.12 and 3.12.1 representations and feature maps indicate the same concept. Consider being consistent.

Sec 3.12.2. Consider removing repetitions about U net to improve the fluidity of the paragraph. As highlighted in the paper, autoencoders are trained to reconstruct the network's input, i.e., the network's

output should be the same as the input. UNet is used to extract semantically meaningful features that are not like the input of the network (e.g., radargram, bio image). Consider being clearer on the paper and improving the section heading. In this direction, some sentences are misleading and lack a justification (e.g., 'In addition, since the basic idea of an autoencoder architecture is to have the same input and output dimensions, autoencoders are a good choice for segmentation task').

3.12.3. This section lacks the network adaptation for the radargram characteristics (different from those of the computer vision/optical data), e.g., 1 channel instead of 3, and not additive noise.

Sec 3.12.4. 'Moreover, it uses side outputs compensating for the absence of deep supervision, which is a characteristic of fully convolutional neural networks.' How? The meaning of the sentence 'HED considers edge detection as a holistic problem (global image-to-image mapping).' is unclear. Missing details to understand the logic of the method.

Sec 3.12.5. It is unclear if the network extracts features at different scales or analyzes data acquired over the same area with different (spatial?) resolutions. How is it done?

3.12.6. 'Simply put, the input of each node is a combination of input and the hidden state of the same node from the previous time step (Goodfellow et al., 2016).' Too informal. From the paragraph, it is not clear how RNNs work.

Sec 4. The title is misleading. This paper aims to identify englacial stratigraphy. Consider being consistent and using the same terminology—are you identifying IHR, the englacial stratigraphy? What about segmentation and target detection?

Pg16, ln 401. What is feature referring to in the sentence 'Automatic feature detection methods'? Pay attention to that 'features' indicated the output of CNN.

Pg 16, ln 408. 'Such studies include a variety of approaches such as neural networks (Reichman et al., 2017)'. Which type of NN and training are used?

Ln 410-412. I am not sure about the meaning of this sentence 'However, as a result of radar systems differing in frequencies and waveform characteristics (thus resolution and penetration depth), studies applied to GPR and RES systems over mediums other than ice, do not provide considerable insights.'. What is the difference between radargrams (acquired how? Spacecraft? aircraft?) and GPR and RES (radio echo sounding?)? Also, the abstract claims that 'we discuss a variety of methods which were developed or applied to RES data over the last decades, including image processing, statistical techniques, and deep learning approaches.'.

Ln 414-418. 'In a number of studies radargrams were analysed to find different segments or subsurface targets (e.g. englacial boundaries, EFZ, basal units) and classes of events in each radargram (e.g., Donini et al., 2019; Goldberg et al., 2020; García et al.2021, 2023). Even though we focus on the methods for mapping englacial ice structure and tracing IRHs and/or layer boundaries, we also take a look at studies done to detect regions and targets in radar products since those are, in terms of methodology, in close vicinity to stratigraphy mapping endeavours.' If this is the case, it should be claimed in the abstract, title, introduction, etc.

Ln 422-424. Repetition. This should go into the motivation for IHR identification, not here.

Ln 426. I am not sure that I would call filtering and transformation computer vision-based methods, as DL and SVM are computer vision. Consider using classical/traditional machine learning methods or something similar. The heading of the following sections should be the same as the bullet list.

Ln 487. Feature is used as geological target/IHR. Pay attention to the fact that computer vision has a different meaning. Thus, trying to be consistent in the paper.

Ln 485 (Ferro and Bruzzone (2013)) I would specify that this work is on planetary data (sharad) given that most of the other works are on aircraft data (e.g., CREeSIS-MCords). Planetary data have different types of noise and radiometric characteristics than aircraft data.

Ln 504. 'A user is required to determine the number of visible layers initially.' Manually?

Ln 508, Panton 2014. The methods are not described.

Ln 640. Consider adding the comparison to the paper 'The modification makes the method more appropriate for a wide range of radargrams, based on comparison with results of Crandall et al. (2012); Lee et al. (2014); Rahnemoonfar et al. (2017a).'

Ln 656.' After statistical analysis, different classes are represented by pdf. This is partially true. The authors designed a set of manual features (not only statistical features) that are extracted for each pixel in the radargram and then analyzed with SVM. The work was improved in Donini, Elena, et al. "An automatic approach to map refreezing ice in radar sounder data." Image and Signal Processing for Remote Sensing XXV. Vol. 11155. SPIE, 2019.

Ln 671, 'Going away from classification of regions and ice-base and ice surface IRHs to tracing internal IRH' is too informal.

Ln 714. Foci -> focus?

Ln 713 'In this subsection, summaries and main points of the studies that used deep learning-based methods are presented. We would like to note that although the primary foci of some of the the works e.g. Donini et al. (2021, 2022c); Garcia et al. (2021); García et al. (2023); Ghosh and Bovolo (2022a, 2023b) lie in radargram region segmentation, they are included in this review because of their methodological relevance for the overall objective.' I would change the paper's objective to methods for the automatic analysis of radargrams. Otherwise, explaining how to perform segmentation does not make sense if the paper's aim is layer detection.

Ln 755. The sentence 'Their initial stage is to remove the noise by using bilateral filtering.' Should emphasize that the method cannot handle the radar's noise characteristics and needs to apply strong preprocessing to mitigate speckle. This contrasts with the method Donini et al., and Garcia et al. used to manage the noisy properties within the network. The paper, in general, lacks this kind of critical analysis.

Ln 984. Some solutions have been proposed to overcome this (Cai et al., 2022; García et al., 2023; Moqadam et al., 2024). Also, Donini et al. propose a pre-training to set the network parameters to not random values.

Ln 929. The section on unsupervised segmentation of radargrams lacks the reference to Donini, Elena, et al. "Unsupervised semantic segmentation of radar sounder data using contrastive learning." Image and Signal Processing for Remote Sensing XXVIII. Vol. 12267. SPIE, 2022.

---

## Author Comment (AC1)

**Referee #1**

The authors present a very coherent and complete literature review on the various applications for detecting IRHs in radagrams, a very challenging and important problem. This is a very useful paper for anyone who wants to study IRHs and develop novel methodologies for automatically identifying them. I really enjoyed reading this review and I believe that the paper contributes to the existing literature and should be published subject to minor corrections.

- We thank the reviewer for their effort to go through our manuscript and provide the feedback.

My comments are given below:

The authors should consider mentioning at the introduction that radio-echo sounding is also often referred to as Ice Penetrating Radar (IPR) or radioglaciology.

- The authors follow the suggestions by Schlegel et al (2022, Ann. Glac.) about radar/RES terminology. But we can add IPR as well for completeness.

Line~ 40: Matlab is referenced as a software for processing radar data and automated mapping of IRHs. Although I am sure there are some Matlab scripts that can do that, Matlab in general is a high-level programming language and should not be referenced as a radar processing tool. The authors should also reference Geolitix, which is a recent commercial software that is used quite extensively by the GPR community nowadays. The unique thing of geolitix is that it is web based. Regarding Matlab, there are some packages for example "GPRlab: A ground penetrating radar data processing and analysis software based on MATLAB". But I have never used them, so I cannot comment on the quality of these tools.

- Thanks for the point. We will incorporate Geolitix, and correct the mistake with Matlab and name the relevant Toolboxes.

Line~ 60. The authors mention echograms are also known as radagrams. I would also add the term B-Scan, which is also widely used by the community.

- Thanks for this point. In 2.2, we have mentioned A-scope and Z-scope. We will include B-scan (B-scope) as well.

Line ~ 80. I would suggest the reviewers to add the well-known book of David Daniels "Ground Penetrating Radar" as a reference.

- We will add this reference.

Line ~ 85. The authors refer to radioglaciology as a remote sensing methodology. This term might exclude the ground-based in-situ systems.

- It is mentioned there as a remote sensing method to emphasize its main difference to satellite imagery. But we will make sure to clarify that ground-based systems are also RES systems.

Line ~ 85: The authors mention that the waves are reflected by changes in the "complex permittivity" of ice. I believe it would be better to use the term "dielectric properties" instead of

"complex permittivity". Dielectric properties are more inclusive and hold as special cases the complex permittivity, conductivity and magnetic permeability. It would be better also to explain the term "dielectric properties" because in paragraph 95 it is said that the main source of reflections of IRH are changes in conductivity while in previous paragraph you mention complex permittivity as the main source of reflections.

- Thank you for your meticulous point. We acknowledge that different terminologies for radioglaciology have been used throughout the decades, but we have decided to use the most conventionally-accepted ones for this review article. We will use the broader term dielectric properties.

Line ~ 90. The authors refer to similar references as they refer them on paragraph 80 for the same context. I think there is a repetition here that can be avoided.

- This is indeed a repetition, we will remove and refer to the line ~80 (2. background) section.

Line ~ 95. The abbreviation IRH is previously already defined.

- This is a repetition, we will remove that.

Line ~ 95. The authors mention that the main source of reflections of IRS is the change in conductivity, while at Line~85 they mention that reflections occur due to complex permittivity.

- The complex permittivity is more general so we change this to complex permittivity.

Line ~ 100. The authors mention that crystal orientation fabric can also result in reflections. How does this translate to dielectric properties? In the reference given by the authors (Eisen 2003) it is stated that there are evidence that suggests that changes in permittivity and not conductivity give rise to reflections in IRHs.

- The reference should be Eisen et al. 2007 (https://doi.org/10.5194/tc-1-1-2007), where it has been discussed that COF-related IRHs stem from conductivity. Changes in COF can occur along IRHs (together with conductivity) but also independently of conductivity. We will clarify this.

Line ~ 105. I would suggest the authors to use the term B-Scan as well as echogram and radargram.

- As previously stated, we will note that.

Line ~ 105. Along A-scope I would also suggest the term A-Scan.

- This will be added in parentheses.

Line ~ 115. The authors discuss Figure 1, where an example of old traces is illustrated, where traces needed to be differentiated. I think it would be better to show more recent radagrams that don't need to be differentiated.

- As this review aims to be comprehensive, we have decided to include this example to include also older systems. Not including those system would mean to exclude a lot of data.

Line 127. Typo "…. a simplified schematic of a …"

- We will correct that.

Line ~ 130. The authors state "The red lines in the radargram indicate IRHs. In an ice sheet, these represent the interfaces between the neighbouring ice layers of different properties, such as layers with different density or crystal orientation fabric, or they can be caused by thin individual horizons with higher conductivity, thus forming IRHs". From the above I understand that interfaces caused by changes in conductivity (and not permittivity) are IRHs?

- Conductivity is mentioned as an example for some of the properties. As mentioned in Line ~85, IRHs are caused by complex-valued permittivity, which includes both conductivity and permittivity. In the first part of this mentioned text, the density is mentioned which is caused by change in permittivity, but only in the top few hundred meters, only not mentioned.

Line ~ 170. Reference is needed there to support this statement.

- We will add Winter et al. 2019 (cited also in the paper) for the sentence " The IRHs in snow and firn... ". However, for the subsequent statement "This is the foremost reason …" is an observation by the authors.

Line ~ 135. This statement has been repeated many times in the text so far, and I believe that could be omitted.

- That is correct, we will remove that.

Line ~ 190. It would be good to outline the limitations and advantages for each type of methods.

- This has been mentioned for some of the methods (in section 4), however, we will add this to the methods as much as it can be relevant. This is because each method can be implemented differently and could alter in terms of limitations. We will add general remarks on the limitations, advantages and weaknesses.

Line ~ 210. "..avoidance of having discontinuities", I would add "..avoidance of having discontinuities between splines".

- Thank you for the suggestion, we will incorporate that.

Line ~ 225. Both Level Set Function and Snake are part of the Active Contour methodologies. The authors should consider adding them as subsections in the same section "Active Contour".

- Thank you for the suggestion, we will merge them into one "Active contour section".

Line ~ 250. The Radon and Hough transform are very similar. To my understanding the Hough transform is a discreet version of the Radon transform.

- Thank you for the suggestion. This is implicit in the text, but we will explicitly mention that.

Line ~ 290. This is a very generic definition of support vector machines (SVM). The original support vector machines fit hyper-planes, but with the Kernel trick SVM can deal with non-linear boundaries. Also SVM can deal with multi-class problems using strategies such as one vs one, or one vs all.

- Thank you for the suggestion. This is a detail but correct observation. We will add the note about SVMs ability to deal with multi-class problems.
  Section 3's short description of each method are meant to be to some extent generic, as the

aim is to explain the methods in general and not necessarily how they can be used for IRH tracing.

Many of the sections that define the methodologies 3.1-3.12 are flagged as AI-generated text, which often results to read very generic and definitional.

- The authors have checked with a couple of AI-detector softwares and sections 3.7 to 3.11 seem to show that some of the sentences are AI-generated. However, as clearly stated at the end, no sentence or clause were AI-generated or AI-refined. As the AI-detector are not reliable, they can detect erroneously.

Section 3.12. Flagged as AI generated. This is a very detailed and out of scope explanation of deep learning. I believe that this paragraph is not necessary. Otherwise another paragraph is needed to define what is machine learning prior to describing Support Vector Machines.

- We have considered to summarize and simplify the text in some of the subsections of section 3 to facilitate readability.
At the time that this referee's comments were out, the authors checked LuCun et al. 2015 (oi.org/10.1038/nature14539) paper with AI-detector websites and some of that text was also flagged as AI-generated. This is an indication of the immaturity of the AI-detector tools. Also, the authors checked section 3.12 with some AI-detector tools and it was not flagged as AI-generated.

Line ~ 345. The authors state that U-net is a type of autoencoder. An autoencoder outputs the same inputs i.e. is an un-supervised learning method that has the same inputs as outputs. U-net can be an auto-encoder if the inputs and outputs are the same, but U-nets can also use different inputs and outputs. The paper cited by the authors (Ronneberger et al., 2015) is the first introduction to U-net, used as a segmentation tool, therefore not as an autoencoder.

- Thank you for highlighting this important point. This is correct. U-net, although is similar to autoencoders because of its encoder-decoder structure and can be used like one, it is not an autoencoder. It differs in terms of input-output, purpose, and skip connections. We will correct the text accordingly.

Line ~ 395. The authors state "Over the last decade, various studies have put forth the use of pattern recognition methodologies in the examination of ground-penetrating radar (GPR) signals (e.g. Delbo et al., 2000)." The reference is from 2000 so it cannot support this statement.

- We will use more recent references.

Line ~ 435. Typo "….a semi-automatic picking routine…"

- We will correct that.

Line ~ 436. Typo " …the maximum amplitude of each…"

- We will correct that.

Chapter ~ 4.1. It is not clear why some references are in bold font while others are not.

- Throughout section 4, bold references are the ones that are included in Table 1, as they are the references which have performed the similar task of tracing IRHs or segmenting

radargrams. We have decided to bold those references (in their first mention) to indicate that the summary following belongs to that publication, for the interested readers to refer to for further information.

Line ~ 785. This is a very big paragraph that needs to be divided into smaller ones.

- Section 4 paragraphs are meant to be similar publications bundled to each other. However, we will consider this and make necessary changes in order to facilitate readability, such as shortening paragraphs.

In Figure 7, would it be easy to include machine learning (not deep learning) related papers as well?

- We assume the referee means to separate SVM and similar methods as Machine learning methods. We will consider that, although this will be only few papers, it might be a better categorization to separate them. Thank you for pointing it out.

The authors should use some figures from the most important cited papers to illustrate the results of the methods discussed in chapter 4. It would be good to showcase with some visual examples how these methods work.

- Including figures from some of the papers is potentially a good idea. However, considering the current length of the paper, it would not make sense to extend it even more. We hope that the description of each paper suffices to encourage the readers to check out the papers that are interesting to them. We will consider to add more figures during the revision where those might really provide added values without lengthening the manuscript too much.

---

## Author Comment (AC2)

**Referee #2**

Review of the article 'Review article: Feature tracing in radio-echo sounding products of terrestrial ice sheets and planetary bodies' by Hameed Moqadam and Olaf Eisen.

- We thank the reviewer for their effort to go through our manuscript and provide the feedback.

The paper proposes a review of the literature methods for detecting internal reflection horizons related to englacial stratigraphy and several glaciological applications to monitor the cryosphere. The structure of the paper is very complex and fragmented, making the sections contain a lot of repetitions from other sections. The English is good, but sometimes very informal, and there are typos, so it should be improved to match the journal's quality. Here are the general comments on the paper:

The paper's aim is unclear, whether to review the automatic methods for radargram analysis or to examine IRH/ice layer tracing. The paper focuses on the importance of IHR tracing (abstract, intro, and background sections). Still, it also presents methods for target detection and segmentation that are unrelated to edge/IHR/layer detection. It is also unclear what criterion is used to select the methods the paper analyzes (the abstract considers those applied to RES data, and sec 4 claims only those to analyze radargrams). What about methods for GPR data?

- Thank you for pointing this out. As the main objective of the paper is a review of methods that trace IRHs in radargrams, we have included methods that segment radargrams and find targets as well. This is due to the fact that these methods are quite similar to each other. Therefore, to keep the review article comprehensive, we have decided on including those. This is mentioned in Lines ~65, ~156 and in more details in Lines 414 – 418.
  RES (or radar) is an overarching terminology, which includes all airborne or ground-based active electromagnetic methods in the radio-frequency range. Details can be found in Schlegel et al. (2023, Ann. Glac.). We follow their terminology in our manuscript and, for the sake of brevity, will not discuss all possible namings of radar methods again.

The paper's headings are very confusing and do not give an overview of the section's content.

- Thank you for this comment, however it is not clear which of the headings are meant by this. In the revision we will revisit the headings and outline the sections and substructure more clearly.

General comments on Sec 3. The criterion used to divide the data analysis methodology into different sections (sec 3.1-3.12) is unclear. Considering the large number of methods, I would expect a clusteringwith a clear logic. Moreover, the writing should be improved to increase fluidity. Further, the sections miss an analysis of the pros and cons of the methodologies, e.g., which are the ideal and worst conditions, which are the method's core hypothesis, how should be the radar data preprocessed, how are the radar data non-ideality handled (e.g., speckle), which are the limitations, and on which data was the method validated (e.g., planetary, terrestrial, high or low spatial resolution, adaptability to other data).

- Thanks for this point. We will make the text on section 3 more simple to make it more readable.
  About stating the pros and cons, and best and worst conditions, considering both the scope and length of the paper, we have decided not to elaborate on those.
  About pre-processing methods of the radar data, as this is more dataset-dependent than method-dependent. We therefore do not consider it the scope of this review article.
  Which data were used for each of the studies are mentioned in section 4 in the individual description of the papers.

Moreover, I would expect more figures (e.g., flow charts) and formulas to understand each method's details better. Finally, I am very confused about whether the paper aims to show the methodologies for radargram automated analysis, IHR detection, or layer tracing. The paper focuses on the importance of IHR and related applications, while section 3 presents a list of automated methods used for analyzing radargrams.

- Considering the current length of the paper, adding figures for each (or even some) of the papers would make the article longer and less favourable for readers, without providing considerable added value in our opinion.
  The methods introduced in section 3 are the major method that are used for IRH tracing and radargram segmenting in the papers that are described in section 4. Nevertheless, we will consider in the revision to add a restricted number of additional figures to make the contents more clear.

General comment on Sec 4. It is unclear why Sec 3 and Sec 4 are divided and not merged, given the lack of details of Sec 3 and the repetition of concepts already explained in Sec 4. Moreover, consider using the taxonomy of sec 4 also for sec 3. Instead of Sec 3 and Sec 4, I suggest having three sections with the taxonomy of 4.1,4.2, and 4.3 that present the computer vision theory (an improved version of what is in sec3) and the application to the radar data (an improved version of what is in sec4). Moreover, the session lacks details, formulas, and images to understand how the methods work. In general, the methods' descriptions lack a discussion on the pros and cons, limitations, and if/how they can be improved. Also, the connection between paragraphs is missing, and the section looks like a list of methods without any critical analysis.

- The suggested taxonomy is very appealing and we will consider it in the revised version. Including figures and formulae, and stating pros and cons would make the article longer and are outside the scope of the paper, see above.
  Section 4's aim is to summarize the research done on IRH tracing and radargram segmenting and not suggesting ways to improve each of the methodologies. In that context we do not aim for a textbook-style of article, but a comprehensive review of what has been tried and applied already to help other scientists to focus their efforts and avoid doubling

Further, the methods are applied to very different types of data (SHARAD, MCoRDS, HiCARS, accumulation data) without describing the properties of the data in terms of noise, resolutions, acquisition geometry, and so on. Consider adding information on the datasets used and their characteristics (maybe in sec 2.2). Further, the paper's aim is unclear (IRH detection or segmentation or automatic analysis of radar data). At the end of each section (i.e., 4.1, 4.2, and 4.3),

the methodologies should be discussed to understand the limitations that forced the development of novel techniques.

- Thank you for this accurate point. However, adding details about all the used datasets is out of the scope of this paper since the scope is not about datasets but the methodologies. The aim of the section 4 is stated at the beginning of the section.

General comment on Sec 4.3. This section moves from segmentation to layer detection. Consider fixing an aim to the paper and sticking to it. There are few indications on the data types used to validate the methods (and how the data-specific characteristics are handled, including noise and resolutions). Moreover, consider dividing the section into subsections (e.g., DL for layer detection, DL for semantic segmentation). The section lacks i) a critical analysis of the methods, ii) information on the type of data the methods are validated, and iii) information on the computational load (given the high computational cost of the DL algorithm).

- The datasets are mentioned for the summarized studies. Also pre- and post-processing methods that were used for each paper are also mentioned when it seemed important or specific.
  The scope of the section is not to analyze the methods but to give the reader an overview of the method implemented in each study.
  Dividing  section 4.3 into DL for layer detection, DL for semantic segmentation is a good idea and we will consider that in the revised version.

Sec 5, discussion. To better understand the validity and differences of the methods, I would expect a qualitative/quantitative comparison.

- This is a very nice suggestion. But a qualitative/quantitative comparison of is way beyond the scope of a review article and is a good idea for writing another completely different study, as it would require an implementation of various methods to the same benchmark dataset. Obviously, that is beyond the scope of a review.

Sec 6. The paper's aim is changed to 'Consequently, this review aims to provide a contemporary overview of advancements in this field of research over the past two decades.' The section has a lot of repetition compared to sec 4 and 5. Consider merging the sections. Finally, the section misses the expected future directions in the IRH detection/ automatic analysis of radargrams.

- Thanks for the suggestion, we will consider that.

Below are the detailed comments (Pg stands for page and Ln/ln for line).

Pg 2, ln 55, Antarctica is larger than Greenland but poorer in terms of data. There are large areas without any acquisitions.

- Thanks for the suggestion, we will implement that.

Sec 2.1 repeats concepts already defined in the introduction. Given the length of the paper, consider removing the repetitions. Ln 96-100 lack details on the IHR formation and give partial information through examples. Consider improving this paragraph.

- Thanks for the suggestion, we will consider that.

Pg 4, ln 103. Consider adding references for z-scope radargrams, e.g., 'Schroeder, Dustin M., et al. "Radiometric analysis of digitized Z-scope records in archival radar sounding film." Journal of Glaciology 68.270 (2022): 733-740.'

- Thanks for the suggestion, we will consider that.

Sec 2.2, line 106. Consider removing the repetitions on the reflection generation: 'the aforementioned characteristics such as presence of impurities, acids, and changes in ice-crystal orientation cause reflections, and when they are laterally coherent, they appear as horizon'.

- Thanks for the suggestion, we will change that.

Sec 2.2, ln 107-110. Consider defining a radargram as a 2D matrix of $N\_T$ traces and $N\_S$ samples. The definition of the radargram pixels as indicating the power/amplitude is misleading. Consider that radargram may also be complex, meaning there is the amplitude and phase radargram or the imaginary and real radargram.

- Thanks for the suggestion, we will change that accordingly.

Figure 1.a and 1.b. Consider showing the figures in dB for a better visualization. Moreover, figs 1.a f lack the x and y axes ticks and labels. Is the x axis of fig 1.a valid also for fig 1.b-f? If so, clarify it in the caption. Moreover, clarify that the radargrams are presumed in the caption.

- They are the same radargram section, we will clarify this in the revised version.

Sec 2.3, ln 132-133. Repetition of the ice layer generation.

- Thanks for the suggestion, we will remove that.

The title of sec 2.4 is misleading as the section is about the information the englacial stratigraphy provides.

- Thanks for this important point. We will find a better title.

Sec3, ln 156. 'In this section, we briefly overview the methods applied to tracing IRH and segmenting radargrams.' This sentence is not in line with the previous section of the paper, as the focus was only on IHR and not on the identification of ice-sheet targets (i.e., segmentation). Consider improving the sentence.

- We will go thought the paper and make it more uniform in terms of these two applications.

Pg 7, ln 159. The sentence 'Given the versatile application of RES across various domains,' lacks a reference.

- Thanks for the suggestion, we will add a reference.

Sec 3 ln 160-165. Repetition of concepts described in section 2.4.

- Thanks for the suggestion, we will remove that.

Pg 7, ln 166. This sentence 'Constructing an automated tracing method for RES encounters a significant challenge when dealing with closely spaced layers.' lacks justification.

- This is an observation that we have. We will try to bring up better arguments for this.

Pg7, ln 179. What about folded or interrupted layers (e.g., those in the basal area)?

- This falls under the last one (complex englacial structures), but perhaps it is worth mentioning separately.

Pg 7, Ln 181. Horizontal or vertical resolution? Justification?

- We will add clarification for this.

Pg 7, Ln 182. Large SNR should be small SNR?

- Thank you for pointing it out, we will fix this mistake.

Pg 8, ln 163-164. Repetition on the motivations for tracing IHR.

- That is true, we will remove repetitions in the revised version.

Pg8, ln 183. 'We will give a short summary of the methods that have been utilized in mapping and segmenting radargrams. The provided method summaries are intended to aid understanding of the timeline of methodologies in section 4, to make readers more aware of the underlying components or procedures of each method.' I am unsure how this sentence relates to the previous part of the section that focuses on tracing IHR.

- This is a good point as the paragraph is not very coherent, we will improve that in the revised version.

Sec 3.1, cross-correlation and peak following. This section lacks the claim of the strong hypothesis that the ice stratigraphy is expected to be constant and horizontal (i.e., without abrupt changes in the steepness). What about the basal region, where shadows mask the reflections?

- Thanks for the suggestion, we will consider adding or clarifying the points.

Sec 3.2, filter. The reference Ilisei and Bruzzone does not refer to canny filtering (it's a statistical analysis of subsurface targets). The same is true for (Freeman et al., 2010), which uses morphological filters and thresholding). Consider removing the reference and description to canny filter and be more general about filtering and thresholding. Moreover, considering the complexity and hypotheses needed for thresholding, I expect at least a sentence discussing it. Finally, this section should at least refer to speckle and how it is tackled, given that speckle can be seen as an abrupt change of intensity of the pixels and thus very visible to canny filter.

- Thanks for the suggestion, we will consider a better description for Canny and include a mention of the thresholding and its significance.

Sec 3.3. improve the computer vision description of Snake.

- Thanks for the suggestion, we will consider that.

Section 3.4. What is this reference to 'cfd, 2019'? Missing justification to the sentence 'making it well-suited for the intricate analysis Radargrams'. It is unclear why this methodology is presented even if it has not been applied to radargrams. Consider removing the section.

- The reference name is a mistake and we will correct it.

Sec 3.5. Statistical analysis is not a method. This section should be improved as it does not detail how the analysis works. The section is also misleading as Rayleigh and Nakagami distributions are valid only under specific hypotheses (e.g., target analyzed and data type). Finally, this section concerns segmentation/target detection, not layer tracing. How is it related to the other sections?

- As previously stated, we are also considering target detection and segmentation methodologies. We will improve the text in 3.5.

Sec 3.6. Consider improving the English (too informal). I would not call a method based on an LPF and thresholding robust (… this method, although being robust...). What about the speckle? In general, the section lacks methodological details. Also, this is very similar to the method in 3.2; consider joining them or explaining the difference between the sections more clearly.

- Thanks for the suggestion, we will consider that.

Sec 3.7. There is no reference to a paper analyzing radargrams. Consider removing the section.

- Thanks for the suggestion, we will consider that.

Sec 3.8. Missing reference to Xiong, Siting, Jan-Peter Muller, and Raquel Caro Carretero. "A new method for automatically tracing englacial layers from MCoRDS data in NW Greenland." Remote Sensing 10.1 (2017): 43. And related works. Moreover, I expect formulas to help me better understand the methodologies.

- Thanks for the suggestion, we will add the references.

Sec 3.9. Most of the papers cited are not related to radargram analysis.

- In section 3, we describe the methods also in a general sense, that is why some of the references in this section are not related to radargram analysis.

Sec 3.10. The papers cited are not related to radargram analysis.

- In section 3, we describe the methods also in a general sense, that is why some of the references in this section are not related to radargram analysis.

Sec 3.11. Consider improving the section on SVM. There is plenty of work to be done for segmenting radargrams. Moreover, the description of SVM, which is very wrong (e.g., SVM is also multiclass), should be greatly improved. The meaning of this sentence 'SVMs are able to identify the optimal surface, mitigating overfitting during training' is unclear. What is the optimal surface? Also, overfitting mostly depends on the training sample numbers and representativity. Missing the motivations that pushed the community to move to deep learning (i.e., the necessity of manually designing the features).

- This is a good point and we will change this subsection in the revised version.

3.12. The section is confusing; for example, I would define deep learning before explaining representation learning. Consider improving the general description of deep learning and explaining why it is important for analyzing radargrams. In the sentence 'DL performs such tasks using multiple levels of non-linear modules, transforming these representations from raw to higher and more abstract level', what does 'representations' refer to? The sentence 'The advantage of semi-supervised learning is that it does not require a large amount of labeled data, but there is the danger

of learning irrelevant features' is not true. The problem is overfitting. In the sentence 'The third class is unsupervised learning. As the name suggests, the learning procedure is based on finding representations without help of known targets', known targets should be labeled images/data. Finally, this section lacks a description of the main disadvantage of DL, which is the overfitting and poor generalization capability of the network with small labeled datasets.

- This is a good point and we will change this subsection in the revised version.

Sec 3.12.1. Several complex concepts are not explained while describing the so-called general architecture of the neural network. In general, the paper shows inconsistent levels of detail in the description of computer vision concepts. Basic concepts (e.g., supervised, semisupervised, unsupervised) are described in detail, while the neural network layers are just mentioned. The most critical part of DL is the training and definition of the loss function. I would expect at least to mention how the network training works.

- We will change this subsection in the revised version.

Considering their usage in 3.12 and 3.12.1 representations and feature maps indicate the same concept. Consider being consistent.

- We will use the same terminology.

Sec 3.12.2. Consider removing repetitions about U net to improve the fluidity of the paragraph. As highlighted in the paper, autoencoders are trained to reconstruct the network's input, i.e., the network's output should be the same as the input. UNet is used to extract semantically meaningful features that are not like the input of the network (e.g., radargram, bio image). Consider being clearer on the paper and improving the section heading. In this direction, some sentences are misleading and lack a justification (e.g., 'In addition, since the basic idea of an autoencoder architecture is to have the same input and output dimensions, autoencoders are a good choice for segmentation task').

- We will change this subsection in the revised version.

3.12.3. This section lacks the network adaptation for the radargram characteristics (different from those of the computer vision/optical data), e.g., 1 channel instead of 3, and not additive noise.

- We will change this in the revised version.

Sec 3.12.4. 'Moreover, it uses side outputs compensating for the absence of deep supervision, which is a characteristic of fully convolutional neural networks.' How? The meaning of the sentence 'HED considers edge detection as a holistic problem (global image-to-image mapping).' is unclear. Missing details to understand the logic of the method.

- We will change this sub-chapter in the revised version.

Sec 3.12.5. It is unclear if the network extracts features at different scales or analyzes data acquired over the same area with different (spatial?) resolutions. How is it done?

- We will change this sub-chapter in the revised version.

3.12.6. 'Simply put, the input of each node is a combination of input and the hidden state of the same node from the previous time step (Goodfellow et al., 2016).' Too informal. From the paragraph, it is not clear how RNNs work.

- We will change this sub-chapter in the revised version.

Sec 4. The title is misleading. This paper aims to identify englacial stratigraphy. Consider being consistent and using the same terminology—are you identifying IHR, the englacial stratigraphy? What about segmentation and target detection?

- As said previously, other similar methods are considered as well.

Pg16, ln 401. What is feature referring to in the sentence 'Automatic feature detection methods'? Pay attention to that 'features'indicated the output of CNN.

- Noted, we will consider that.

Pg 16, ln 408. 'Such studies include a variety of approaches such as neural networks (Reichman et al., 2017)'. Which type of NN and training are used?

- We will modify the paragraph, as stated before.

Ln 410-412. I am not sure about the meaning of this sentence 'However, as a result of radar systems differing in frequencies and waveform characteristics (thus resolution and penetration depth), studies applied to GPR and RES systems over mediums other than ice, do not provide considerable insights.'.

- This is the reason why we do not include radar studies over mediums other than ice.

What is the difference between radargrams (acquired how? Spacecraft? aircraft?) and GPR and RES (radio echo sounding?)? Also, the abstract claims that 'we discuss a variety of methods which were developed or applied to RES data over the last decades, including image processing, statistical techniques, and deep learning approaches.'.

- The radargram differences fall under the categories of datasets, which is out of the scoepe of this review article.

Ln 414-418. 'In a number of studies radargrams were analysed to find different segments or subsurface targets (e.g. englacial boundaries, EFZ, basal units) and classes of events in each radargram (e.g., Donini et al., 2019; Goldberg et al., 2020; García et al.2021, 2023). Even though we focus on the methods for mapping englacial ice structure and tracing IRHs and/or layer boundaries, we also take a look at studies done to detect regions and targets in radar products since those are, in terms of methodology, in close vicinity to stratigraphy mapping endeavours.' If this is the case, it should be claimed in the abstract, title, introduction, etc.

- We will consider adding this.

Ln 422-424. Repetition. This should go into the motivation for IHR identification, not here.

- We consider this repetition and remove them.

Ln 426. I am not sure that I would call filtering and transformation computer vision-based methods, as DL and SVM are computer vision. Consider using classical/traditional machine learning methods or something similar. The heading of the following sections should be the same as the bullet list.

- We will consider better titles.

Ln 487. Feature is used as geological target/IHR. Pay attention to the fact that computer vision has a different meaning. Thus, trying to be consistent in the paper.

- As previously mentioned, we will make sure to be clear on this.

Ln 485 (Ferro and Bruzzone (2013)) I would specify that this work is on planetary data (sharad) given that most of the other works are on aircraft data (e.g., CREeSIS-MCords). Planetary data have different types of noise and radiometric characteristics than aircraft data.

- We will consider to add this note.

Ln 504. 'A user is required to determine the number of visible layers initially.' Manually?

- As mentioned in the original paper, in their training data, an expert counted the visible IRHs.

Ln 508, Panton 2014. The methods are not described.

- The method is mentioned, i.e. snake.

Ln 640. Consider adding the comparison to the paper 'The modification makes the method more appropriate for a wide range of radargrams, based on comparison with results of Crandall et al. (2012); Lee et al. (2014); Rahnemoonfar et al. (2017a).'

- We will consider to add this suggestion.

Ln 656.' After statistical analysis, different classes are represented by pdf. This is partially true. The authors designed a set of manual features (not only statistical features) that are extracted for each pixel in the radargram and then analyzed with SVM. The work was improved in Donini, Elena, et al. "An automatic approach to map refreezing ice in radar sounder data." Image and Signal Processing for Remote Sensing XXV. Vol. 11155. SPIE, 2019.

- We will consider this.

Ln 671, 'Going away from classification of regions and ice-base and ice surface IRHs to tracing internal IRH' is too informal.

- Thanks for pointing this out.

Ln 714. Foci -> focus?

- We will correct this.

Ln 713 'In this subsection, summaries and main points of the studies that used deep learning-based methods are presented. We would like to note that although the primary foci of some of the the works e.g. Donini et al. (2021, 2022c); Garcia et al. (2021); García et al. (2023); Ghosh and Bovolo (2022a, 2023b) lie in radargram region segmentation, they are included in this review because of their methodological relevance for the overall objective.' I would change the paper's objective to

methods for the automatic analysis of radargrams. Otherwise, explaining how to perform segmentation does not make sense if the paper's aim is layer detection.

- This point has been mentioned several times already and we will add this to the abstract and introduction.

Ln 755. The sentence 'Their initial stage is to remove the noise by using bilateral filtering.' Should emphasize that the method cannot handle the radar's noise characteristics and needs to apply strong preprocessing to mitigate speckle. This contrasts with the method Donini et al., and Garcia et al. used to manage the noisy properties within the network. The paper, in general, lacks this kind of critical analysis.

- The scope of this review article is giving the readers an overview of all the methods that haven been implemented for IRH tracing and radargram segmentation. The suggestions is quite detailed, but we will consider to add some information in this respect. However, our intention is to point interested readers to the published material so that they refer to those when they need more insights.

Ln 984. Some solutions have been proposed to overcome this (Cai et al., 2022; García et al., 2023; Moqadam et al., 2024). Also, Donini et al. propose a pre-training to set the network parameters to not random values.

- We will consider adding the paper as well. This paper has been cited other times in out manuscript as well.

Ln 929. The section on unsupervised segmentation of radargrams lacks the reference to Donini, Elena, et al. "Unsupervised semantic segmentation of radar sounder data using contrastive learning." Image and Signal Processing for Remote Sensing XXVIII. Vol. 12267. SPIE, 2022.

- We will consider adding this paper. However, as it is not open access it is difficult to find out more about the paper. Also we could not access it through our institution's access either.

---

## Author Response (AR2)

**Author's response, Referee 1**

Hameed Moqadam

December 19, 2024

The authors present a very coherent and complete literature review on the various applications for detecting IRHs in radagrams, a very challenging and important problem. This is a very useful paper for anyone who wants to study IRHs and develop novel methodologies for automatically identifying them. I really enjoyed reading this review and I believe that the paper contributes to the existing literature and should be published subject to minor corrections.

- We thank the reviewers for their effort to read our manuscript, appreciate the rating of referee 1 and provide our response below.

My comments are given below: The authors should consider mentioning at the introduction that radio-echo sounding is also often referred to as Ice Penetrating Radar (IPR) or radioglaciology.

- The authors follow the suggestions by Schlegel et al (2022, Ann. Glac.) on radar/RES terminology, however, for completeness we have added IPR once.

Line 40: Matlab is referenced as a software for processing radar data and automated mapping of IRHs. Although I am sure there are some Matlab scripts that can do that, Matlab in general is a high-level programming language and should not be referenced as a radar processing tool. The authors should also reference Geolitix, which is a recent commercial software that is used quite extensively by the GPR community nowadays. The unique thing of geolitix is that it is web based. Regarding Matlab, there are some packages for example "GPRlab: A ground penetrating radar data processing and analysis software based on MATLAB". But I have never used them, so I cannot comment on the quality of these tools.

- We have added Geolitix to the softwares, and corrected the mentioned Matlab toolbox.

Line 60. The authors mention echograms are also known as radagrams. I would also add the term B-Scan, which is also widely used by the community.

- In 2.2, we had mentioned A-scope and Z-scope, and we have included B-scan (B-scope) as well.

Line 80. I would suggest the reviewers to add the well-known book of David Daniels "Ground Penetrating Radar" as a reference.

- This reference has been added.

Line 85. The authors refer to radioglaciology as a remote sensing methodology. This term might exclude the ground-based in-situ systems.

- It is mentioned there as a remote sensing method to emphasize its main difference to satellite imagery. However, we have added clarification that ground-based systems are also RES systems.

Line 85: The authors mention that the waves are reflected by changes in the "complex permittivity" of ice. I believe it would be better to use the term "dielectric properties" instead of "complex permittivity". Dielectric properties are more inclusive and hold as special cases the complex permittivity, conductivity and magnetic permeability. It would be better also to explain the term "dielectric properties" because in paragraph 95 it is said that the main source of reflections of IRH are changes in conductivity while in previous paragraph you mention complex permittivity as the main source of reflections.

- Thank you for your meticulous point. Acknowledging that different terminologies for radioglaciology have been used throughout the decades, but we have decided to use the most conventionally-accepted ones for this review article, thus we have used the broader term dielectric properties.

Line 90. The authors refer to similar references as they refer them on paragraph 80 for the same context. I think there is a repetition here that can be avoided.

- This repetition has been removed and readers are referred to the line 80 (2. background section).

Line 95. The abbreviation IRH is previously already defined.

- This repetition has been removed.

Line 95. The authors mention that the main source of reflections of IRS is the change in conductivity, while at Line 85 they mention that reflections occur due to complex permittivity.

- As most isochronal IRH are from conductivity changes, we have kept the text as it was.

Line 100. The authors mention that crystal orientation fabric can also result in reflections. How does this translate to dielectric properties? In the reference given by the authors (Eisen 2003) it is stated that there are evidence that suggests that changes in permittivity and not conductivity give rise to reflections in IRHs.

- The reference was changed to Eisen et al. 2007 (https://doi.org/10.5194/tc-1-1-2007), where it has been discussed that COF-related IRHs stem from conductivity. It has been clarified that changes in COF can occur along IRHs (together with conductivity), but also independent of conductivity.

Line 105. I would suggest the authors to use the term B-Scan as well as echogram and radargram.

- This has been included.

Line 105. Along A-scope I would also suggest the term A-Scan.

- This has been included.

Line 115. The authors discuss Figure 1, where an example of old traces is illustrated, where traces needed to be differentiated. I think it would be better to show more recent radargrams that don't need to be differentiated.

- We have decided to keep this figure the same. This review aims to be comprehensive, therefore we have decided to use this example to include also older systems. Excluding those systems would result in exclusion of a lot of data.

Line 127. Typo ".... a simplified schematic of a ..."

- This has been corrected.

Line 130. The authors state "The red lines in the radargram indicate IRHs. In an ice sheet, these represent the interfaces between the neighbouring ice layers of different properties, such as layers with different density or crystal orientation fabric, or they can be caused by thin individual horizons with higher conductivity, thus forming IRHs". From the above I understand that interfaces caused by changes in conductivity (and not permittivity) are IRHs?

- Conductivity is mentioned as an example for some of the properties. As explained in Line 85, IRHs are caused by complex-valued permittivity, which covers both conductivity and permittivity. In the first part, density is mentioned to cause changes in permittivity, but only in the top few hundred meters of the ice sheet.

Line 170. Reference is needed there to support this statement.

- We have added Winter et al. 2019 (cited also in the paper) for the sentence " The IRHs in snow and firn... ". However, for the subsequent statement "This is the foremost reason ..." is an observation by the authors.

Line 135. This statement has been repeated many times in the text so far, and I believe that could be omitted.

- We have removed that.

Line 190. It would be good to outline the limitations and advantages for each type of methods.

- This has been mentioned for some of the methods (in section 4), however, we have added such details to the method descriptions as much as it can be relevant. This is because each method can be implemented differently and could alter in terms of limitations.

Line 210. "..avoidance of having discontinuities", I would add "..avoidance of having discontinuities between splines".

- This has been corrected.

Line 225. Both Level Set Function and Snake are part of the Active Contour methodologies. The authors should consider adding them as subsections in the same section "Active Contour".

- Thank you for the suggestion, it was implemented.

Line 250. The Radon and Hough transform are very similar. To my understanding the Hough transform is a discreet version of the Radon transform.

- Thank you for the suggestion, it was implemented.

Line 290. This is a very generic definition of support vector machines (SVM). The original support vector machines fit hyper-planes, but with the Kernel trick SVM can deal with non-linear boundaries. Also SVM can deal with multi-class problems using strategies such as one vs one, or one vs all.

- Some subsections are rewritten in this section, including SVM. Section 3's short description of each method are meant to be to some extent generic, as the aim is to explain the methods in general and not necessarily how they can be used for IRH tracing.

Many of the sections that define the methodologies 3.1-3.12 are flagged as AI-generated text, which often results to read very generic and definitional.

- The authors have checked with a couple of AI-detector softwares and sections 3.7 to 3.11 seem to show that some of the sentences are AI-generated. However, as clearly stated at the end, no sentence or clause were AI-generated or AI-refined. As the AI-detectors are not reliable, they can detect erroneously. Nevertheless, this entire subsection (previously 3.12 and now 3.11) is revisited and edited or rewritten.

Section 3.12. Flagged as AI generated. This is a very detailed and out of scope explanation of deep learning. I believe that this paragraph is not necessary. Otherwise another paragraph is needed to define what is machine learning prior to describing Support Vector Machines.

- Point mentioned above. At the time that this referee's comments were out, the first author checked LuCun et al. 2015 (doi.org/10.1038/nature14539) paper with AI-detector websites and some of that text was also flagged as AI-generated, despite originating from a time when such tools were not yet available. This is an indication of the immaturity of the AI-detector tools. Also, at the time of receiving this comment, the first author checked section 3.12 with some AI-detector tools and it was not flagged as AI-generated.

Line 345. The authors state that U-net is a type of autoencoder. An autoencoder outputs the same inputs i.e. is an un-supervised learning method that has the same inputs as outputs. U-net can be an auto-encoder if the inputs and outputs are the same, but U-nets can also use different inputs and outputs. The paper cited by the authors (Ronneberger et al., 2015) is the first introduction to U-net, used as a segmentation tool, therefore not as an autoencoder.

- This section is rewritten, and only for U-Net.

Line 395. The authors state "Over the last decade, various studies have put forth the use of pattern recognition methodologies in the examination of ground-penetrating radar (GPR) signals (e.g. Delbo et al., 2000)." The reference is from 2000 so it cannot support this statement.

- We have added a more recent reference.

Line 435. Typo "....a semi-automatic picking routine..."

- This has been corrected.

Line 436. Typo " ...the maximum amplitude of each..."

- This has been corrected.

Chapter 4.1. It is not clear why some references are in bold font while others are not.

- Throughout section 4, bold references are the ones that are included in Table 1, as they are the references which have performed the similar task of tracing IRHs or segmenting radargrams. We have decided to bold those references (in their first mention) to indicate that the summary following belongs to that publication, for the interested readers to refer to for further information.

Line 785. This is a very big paragraph that needs to be divided into smaller ones.

- Section 4 paragraphs are meant to be similar publications bundled to each other. Nevertheless, to increase readability, we have separated more related parts of longer paragraphs, but nevertheless we tried to avoid fragemtation.

In Figure 7, would it be easy to include machine learning (not deep learning) related papers as well?

- We assume the referee means to separate SVM and similar methods as Machine Learning methods. Since this would be only few publications, it might be a better categorization to separate them. Thank you for pointing this out.

The authors should use some figures from the most important cited papers to illustrate the results of the methods discussed in chapter 4. It would be good to showcase with some visual examples how these methods work.

- Including figures from some of the papers is potentially a reasonable idea. However, considering the current length of the paper, it would not make sense to extend it even more and we rather hope that interested readers would refer to the cited publications. We hope that the description of each paper suffices to encourage the readers to check out the papers that are interesting to them.

**Author's response, Referee 2**

Hameed Moqadam

December 19, 2024

Review of the article 'Review article: Feature tracing in radio-echo sounding products of terrestrial ice sheets and planetary bodies' by Hameed Moqadam and Olaf Eisen.

- We thank the reviewers for their effort to read and evaluate our manuscript and provide the response to the remarks below.

The paper proposes a review of the literature methods for detecting internal reflection horizons related to englacial stratigraphy and several glaciological applications to monitor the cryosphere. The structure of the paper is very complex and fragmented, making the sections contain a lot of repetitions from other sections. The English is good, but sometimes very informal, and there are typos, so it should be improved to match the journal's quality. Here are the general comments on the paper:

The paper's aim is unclear, whether to review the automatic methods for radargram analysis or to examine IRH/ice layer tracing. The paper focuses on the importance of IHR tracing (abstract, intro, and background sections). Still, it also presents methods for target detection and segmentation that are unrelated to edge/IHR/layer detection. It is also unclear what criterion is used to select the methods the paper analyzes (the abstract considers those applied to RES data, and sec 4 claims only those to analyze radargrams). What about methods for GPR data?

- Thank you for pointing this out. Indeed the main objective of the paper is a review of methods that trace IRHs in radargrams, as already stated in the title. To keep the review article comprehensive, we have included methods that segment radargrams and find targets as well due to the fact that these methods are quite similar to each other. This is mentioned in Lines 65, 156 and in more details in Lines 414 – 418. RES (or radar) is an overarching terminology, which includes all airborne or ground-based active electromagnetic methods in the radio-frequency range. Details can be found in Schlegel et al. (2023, Ann. Glac.). We follow their terminology in our manuscript and, for the sake of brevity, will not discuss all possible namings of radar methods (more than a dozen we are aware of) again.

- In the revised abstract, we have mentioned the inclusion of publications that focus also on methods for tasks other than IRH tracing.

The paper's headings are very confusing and do not give an overview of the section's content.

- Thank you for this comment, however it is not clear which of the headings are meant by this. We have revisited the headings and changed some of them, hoping that they are more clear now and readily understandable to the reader.

General comments on Sec 3. The criterion used to divide the data analysis methodology into different sections (sec 3.1-3.12) is unclear. Considering the large number of methods, I would expect a clustering with a clear logic. Moreover, the writing should be improved to increase fluidity. Further, the sections miss an analysis of the pros and cons of the methodologies, e.g., which are the ideal and worst conditions, which are the method's core hypothesis, how should be the radar data preprocessed, how are the radar data non-ideality handled (e.g., speckle), which are the limitations, and on which data was the method validated (e.g., planetary, terrestrial, high or low spatial resolution, adaptability to other data).

- Thanks for this point. We will make the text on section 3 more simple to make it more readable. On stating the pros and cons, and best and worst conditions, considering both the scope and length of the paper, we have decided not to elaborate on those. About pre-processing methods of the radar data, as this is more dataset-dependent than method-dependent. We therefore do not consider it in the scope of this review article. Which data were used for each of the studies are mentioned in section 4 in the individual description of the papers.

Moreover, I would expect more figures (e.g., flow charts) and formulas to understand each method's details better. Finally, I am very confused about whether the paper aims to show the methodologies for radargram automated analysis, IHR detection, or layer tracing. The paper focuses on the importance of IHR and related applications, while section 3 presents a list of automated methods used for analyzing radargrams.

- Considering the current length of the paper, adding figures for each (or even only for some) of the papers would make the article longer and less favorable for readers, without providing considerable added value in our opinion. The description of each paper in chapter 4 are meant to encourage the readers to read the original publications that are of interest to them. The methods introduced in section 3 are the major method that are used for IRH tracing and radargram segmenting in the papers that are described in section 4.

General comment on Sec 4. It is unclear why Sec 3 and Sec 4 are divided and not merged, given the lack of details of Sec 3 and the repetition of concepts already explained in Sec 4. Moreover, consider using the taxonomy of sec 4 also for sec 3. Instead of Sec 3 and Sec 4, I suggest having three sections with the taxonomy of 4.1,4.2, and 4.3 that present the computer vision theory (an improved version of what is in sec3) and the application to the radar data (an improved version of what is in sec4). Moreover, the session lacks details, formulas, and images to understand how the methods work. In general, the methods' descriptions lack a discussion on the pros and cons, limitations, and if/how they can be improved. Also, the connection between paragraphs is missing, and the section looks like a list of methods without any critical analysis.

- The suggested taxonomy is very appealing, but we have decided to keep the methods and publications separate, since readers might only be interested in learning about what methods have been used in certain publications or what datasets have been used, instead of going deep into the methodology. We consider this rather a matter of personal preferences than objective advantages.

- Including figures and formulae, and stating pros and cons would make the article even longer. Such details are outside the scope of the paper, see above, but could rather be the contents of a dedicated textbook. In this context, Section 4's aim is to summarize the research done on IRH tracing and radargram segmenting and not suggesting ways to improve each of the methodologies. We do not aim for a textbook-style of article, but a comprehensive review with easily (and quickly) accessible overview of what has already been investigated and applied to help other scientists to focus their efforts and avoid doubling.

Further, the methods are applied to very different types of data (SHARAD, MCoRDS, HiCARS, accumulation data) without describing the properties of the data in terms of noise, resolutions, acquisition geometry, and so on. Consider adding information on the datasets used and their characteristics (maybe in sec 2.2). Further, the paper's aim is unclear (IRH detection or segmentation or automatic analysis of radar data). At the end of each section (i.e., 4.1, 4.2, and 4.3), the methodologies should be discussed to understand the limitations that forced the development of novel techniques.

- Thank you for this accurate point. However, adding details about all the used datasets is out of the scope of this paper since the scope is not about datasets but the methodologies. The aim of the section 4 is stated at the beginning of that section.

General comment on Sec 4.3. This section moves from segmentation to layer detection. Consider fixing an aim to the paper and sticking to it. There are few indications on the data types used to validate the methods (and how the data-specific characteristics are handled, including noise and resolutions). Moreover, consider dividing the section into subsections (e.g., DL for layer detection, DL

for semantic segmentation). The section lacks i) a critical analysis of the methods, ii) information on the type of data the methods are validated, and iii) information on the computational load (given the high computational cost of the DL algorithm).

- The datasets are mentioned for the summarized studies. Also pre- and post-processing methods that were used for each paper are also briefly mentioned when it seemed relevant or specific. As already mentioned, the scope of the section is not to analyze the methods but to give the reader an overview of the method implemented in each study. Analyzing each methods, comparing them and evaluating them beyond what the original authors already provide would be an herculian effort way beyond what we intend to provide and can achieve, as this would require benchmark test and alike. Taking the comparison with other intiatives, that would rather be a community effort leading to a method intercomparison project (MIP).

Sec 5, discussion. To better understand the validity and differences of the methods, I would expect a qualitative/quantitative comparison.

- This is a good suggestion. But a qualitative/quantitative comparison of is much beyond the scope of a review article and is a good idea for doing a completely different study, as it would require an implementation of various methods to the same benchmark dataset, see also our response to the previous comment. Obviously, that is beyond the scope of a review.

Sec 6. The paper's aim is changed to 'Consequently, this review aims to provide a contemporary overview of advancements in this field of research over the past two decades.' The section has a lot of repetition compared to sec 4 and 5. Consider merging the sections. Finally, the section misses the expected future directions in the IRH detection/ automatic analysis of radargrams.

- As this is the conclusion section, we have mentioned some of the important points once more. Moreover, the challneges are enumerated and a suggestions such as improved synthetic datasets, and model intercomparison are discussed.

Below are the detailed comments (Pg stands for page and Ln/ln for line).
Pg 2, ln 55, Antarctica is larger than Greenland but poorer in terms of data. There are large areas without any acquisitions.

- We have mentioned this.

Sec 2.1 repeats concepts already defined in the introduction. Given the length of the paper, consider removing the repetitions. Ln 96-100 lack details on the IHR formation and give partial information through examples. Consider improving this paragraph.

- We have modified the text.

Pg 4, ln 103. Consider adding references for z-scope radargrams, e.g., 'Schroeder, Dustin M., et al. "Radiometric analysis of digitized Z-scope records in archival radar sounding film." Journal of Glaciology 68.270 (2022): 733-740.'

- We have added this reference.

Sec 2.2, line 106. Consider removing the repetitions on the reflection generation: 'the aforementioned characteristics such as presence of impurities, acids, and changes in ice-crystal orientation cause reflections, and when they are laterally coherent, they appear as horizon'.

- We have removed the repetition.

Sec 2.2, ln 107-110. Consider defining a radargram as a 2D matrix of $N_T$ traces and $N_S$ samples. The definition of the radargram pixels as indicating the power/amplitude is misleading. Consider that radargram may also be complex, meaning there is the amplitude and phase radargram or the imaginary and real radargram.

- Thanks for the suggestion, we considered this. However, as a geophysical wave method, much like seismology, we decided to stick to the conventional terminology, which uses traces as pairs of traveltime and amplitude. From amplitude, magnitude (or power) and phase can be calculated, as well as real and imaginary parts. The suggested separation is more common for particular radar applications and processing techniques.

Figure 1.a and 1.b. Consider showing the figures in dB for a better visualization. Moreover, figs 1.a f lack the x and y axes ticks and labels. Is the x axis of fig 1.a valid also for fig 1.b-f? If so, clarify it in the caption. Moreover, clarify that the radargrams are presumed in the caption.

- We have clarified this in the caption.

Sec 2.3, ln 132-133. Repetition of the ice layer generation.

- We have removed the repetition about IRH origin.

The title of sec 2.4 is misleading as the section is about the information the englacial stratigraphy provides.

- The title of the subsection has been modified.

Sec3, ln 156. 'In this section, we briefly overview the methods applied to tracing IRH and segmenting radargrams.' This sentence is not in line with the previous section of the paper, as the focus was only on IHR and not on the identification of ice-sheet targets (i.e., segmentation). Consider improving the sentence.

- The sentence is improved and should give more clarification to the reader.

Pg 7, ln 159. The sentence 'Given the versatile application of RES across various domains,' lacks a reference.

- The reader is referred to the subsection on applications of mapped stratigraphy.

Sec 3 ln 160-165. Repetition of concepts described in section 2.4.

- The sentences are removed.

Pg 7, ln 166. This sentence 'Constructing an automated tracing method for RES encounters a significant challenge when dealing with closely spaced layers.' lacks justification.

- This is the authors' observation inferred from a number of publications and actually also represents the community agreement although this was never formally articulated in a citable way (the closest one being the white paper (see Bingham et al, https://egusphere.copernicus.org/preprints/2024/egusphere-2024-2593/).

Pg7, ln 179. What about folded or interrupted layers (e.g., those in the basal area)?

- This falls under the last one (complex englacial structures), we have mention those as examples.

Pg 7, Ln 181. Horizontal or vertical resolution? Justification?

- We have clarified this in the revised version.

Pg 7, Ln 182. Large SNR should be small SNR?

- Thank you for pointing it out, we have fixed this mistake.

Pg 8, ln 163-164. Repetition on the motivations for tracing IHR.

- We have remove the repetitions.

Pg8, ln 183. 'We will give a short summary of the methods that have been utilized in mapping and segmenting radargrams. The provided method summaries are intended to aid understanding of the timeline of methodologies in section 4, to make readers more aware of the underlying components or procedures of each method.' I am unsure how this sentence relates to the previous part of the section that focuses on tracing IHR.

- We have shortened the sentence which eliminates the obscurity.

Sec 3.1, cross-correlation and peak following. This section lacks the claim of the strong hypothesis that the ice stratigraphy is expected to be constant and horizontal (i.e., without abrupt changes in the steepness). What about the basal region, where shadows mask the reflections?

- Thanks for the suggestion, we have added explanation to clarifying the points.

Sec 3.2, filter. The reference Ilisei and Bruzzone does not refer to canny filtering (it's a statistical analysis of subsurface targets). The same is true for (Freeman et al., 2010), which uses morphological filters and thresholding). Consider removing the reference and description to canny filter and be more general about filtering and thresholding. Moreover, considering the complexity and hypotheses needed for thresholding, I expect at least a sentence discussing it. Finally, this section should at least refer to speckle and how it is tackled, given that speckle can be seen as an abrupt change of intensity of the pixels and thus very visible to canny filter.

- Thanks for the remark and suggestion. We have removed the citation and modified description for Canny and included a mention of the thresholding and its significance.

Sec 3.3. improve the computer vision description of Snake.

- We have revisited the descriptions of this chapter.

Section 3.4. What is this reference to 'cfd, 2019'? Missing justification to the sentence 'making it well-suited for the intricate analysis Radargrams'. It is unclear why this methodology is presented even if it has not been applied to radargrams. Consider removing the section.

- The reference name is corrected.

Sec 3.5. Statistical analysis is not a method. This section should be improved as it does not detail how the analysis works. The section is also misleading as Rayleigh and Nakagami distributions are valid only under specific hypotheses (e.g., target analyzed and data type). Finally, this section concerns segmentation/target detection, not layer tracing. How is it related to the other sections?

- As previously stated, we are also considering target detection and segmentation methodologies. We are aware that this is not a standalone method, however it has been used under certain conditions in this applicaiton, that is why we have decided to include it. Readers could refer to the cited publications for more details.

Sec 3.6. Consider improving the English (too informal). I would not call a method based on an LPF and thresholding robust (... this method, although being robust...). What about the speckle? In general, the section lacks methodological details. Also, this is very similar to the method in 3.2; consider joining them or explaining the difference between the sections more clearly.

- We have clarified the description.

Sec 3.7. There is no reference to a paper analyzing radargrams. Consider removing the section.

- As we have not mentioned publications that used specific methods for the subchapters of section 3, we have not added them here. However, Radon and Hough have been used in some of the publicaitons (Xiong et al. 2016, 2017, Onana 2015).

Sec 3.8. Missing reference to Xiong, Siting, Jan-Peter Muller, and Raquel Caro Carretero. "A new method for automatically tracing englacial layers from MCoRDS data in NW Greenland." Remote Sensing 10.1 (2017): 43. And related works. Moreover, I expect formulas to help me better understand the methodologies.

- We have mentioned this publication both in Table 1 and in section 4.

Sec 3.9. Most of the papers cited are not related to radargram analysis.

- In section 3, we describe the methods also in a general sense, that is why some of the references in this section are not related to radargram analysis. Radargram-related citations are mention in Table 1 and in section 4.

Sec 3.10. The papers cited are not related to radargram analysis.

- In section 3, we describe the methods also in a general sense, that is why some of the references in this section are not related to radargram analysis.

Sec 3.11. Consider improving the section on SVM. There is plenty of work to be done for segmenting radargrams. Moreover, the description of SVM, which is very wrong (e.g., SVM is also multiclass), should be greatly improved. The meaning of this sentence 'SVMs are able to identify the optimal surface, mitigating overfitting during training' is unclear. What is the optimal surface? Also, overfitting mostly depends on the training sample numbers and representativity. Missing the motivations that pushed the community to move to deep learning (i.e., the necessity of manually designing the features).

- SVM description is fixed.

3.12. The section is confusing; for example, I would define deep learning before explaining representation learning. Consider improving the general description of deep learning and explaining why it is important for analyzing radargrams. In the sentence 'DL performs such tasks using multiple levels of non-linear modules, transforming these representations from raw to higher and more abstract level', what does 'representations' refer to? The sentence 'The advantage of semi-supervised learning is that it does not require a large amount of labeled data, but there is the danger of learning irrelevant features' is not true. The problem is overfitting. In the sentence 'The third class is unsupervised learning. As the name suggests, the learning procedure is based on finding representations without help of known targets', known targets should be labeled images/data. Finally, this section lacks a description of the main disadvantage of DL, which is the overfitting and poor generalization capability of the network with small labeled datasets.

- This subsection is also rewritten.

Sec 3.12.1. Several complex concepts are not explained while describing the so-called general architecture of the neural network. In general, the paper shows inconsistent levels of detail in the description of computer vision concepts. Basic concepts (e.g., supervised, semisupervised, unsupervised) are described in detail, while the neural network layers are just mentioned. The most critical part of DL is the training and definition of the loss function. I would expect at least to mention how the network training works.

- This subsection is also rewritten.

Considering their usage in 3.12 and 3.12.1 representations and feature maps indicate the same concept. Consider being consistent.

- This subsection is also rewritten also with this in mind.

Sec 3.12.2. Consider removing repetitions about U net to improve the fluidity of the paragraph. As highlighted in the paper, autoencoders are trained to reconstruct the network's input, i.e., the network's output should be the same as the input. UNet is used to extract semantically meaningful features that are not like the input of the network (e.g., radargram, bio image). Consider being clearer on the paper and improving the section heading. In this direction, some sentences are misleading and lack a justification (e.g., 'In addition, since the basic idea of an autoencoder architecture is to have the same input and output dimensions, autoencoders are a good choice for segmentation task').

- This entire subsection (previously 3.12 and now 3.11) was revisited and edited or rewritten.

3.12.3. This section lacks the network adaptation for the radargram characteristics (different from those of the computer vision/optical data), e.g., 1 channel instead of 3, and not additive noise.

- This entire subsection (previously 3.12 and now 3.11) was revisited and edited or rewritten.

Sec 3.12.4. 'Moreover, it uses side outputs compensating for the absence of deep supervision, which is a characteristic of fully convolutional neural networks.' How? The meaning of the sentence 'HED considers edge detection as a holistic problem (global image-to-image mapping).' is unclear. Missing details to understand the logic of the method.

- This entire subsection (previously 3.12 and now 3.11) was revisited and edited or rewritten.

Sec 3.12.5. It is unclear if the network extracts features at different scales or analyzes data acquired over the same area with different (spatial?) resolutions. How is it done?

- This entire subsection (previously 3.12 and now 3.11) was revisited and edited or rewritten.

3.12.6. 'Simply put, the input of each node is a combination of input and the hidden state of the same node from the previous time step (Goodfellow et al., 2016).' Too informal. From the paragraph, it is not clear how RNNs work.

- This entire subsection (previously 3.12 and now 3.11) was revisited and edited or rewritten.

Sec 4. The title is misleading. This paper aims to identify englacial stratigraphy. Consider being consistent and using the same terminology—are you identifying IHR, the englacial stratigraphy? What about segmentation and target detection?

- The title of the section was modified.

Pg16, ln 401. What is feature referring to in the sentence 'Automatic feature detection methods'? Pay attention to that 'features'indicated the output of CNN.

- We have replaced feature with "edge" to avoid misunderstanding with output of a CNN.

Pg 16, ln 408. 'Such studies include a variety of approaches such as neural networks (Reichman et al., 2017)'. Which type of NN and training are used?

- The text was corrected.

Ln 410-412. I am not sure about the meaning of this sentence 'However, as a result of radar systems differing in frequencies and waveform characteristics (thus resolution and penetration depth), studies applied to GPR and RES systems over mediums other than ice, do not provide considerable insights.'.

- This is the reason why we do not include radar studies over mediums other than ice.

What is the difference between radargrams (acquired how? Spacecraft? aircraft?) and GPR and RES (radio echo sounding?)? Also, the abstract claims that 'we discuss a variety of methods which were developed or applied to RES data over the last decades, including image processing, statistical techniques, and deep learning approaches.'.

- The radargram is a representation of the data, no matter of the platform of acquisition, Differences in radargrams from different systems fall under the categories of datasets, a comparison of which is out of the scope of this review article, as we focus on ML methods. The methods mentioned in the abstract are explained in Section 3.

Ln 414-418. 'In a number of studies radargrams were analysed to find different segments or subsurface targets (e.g. englacial boundaries, EFZ, basal units) and classes of events in each radargram (e.g., Donini et al., 2019; Goldberg et al., 2020; García et al.2021, 2023). Even though we focus on the methods for mapping englacial ice structure and tracing IRHs and/or layer boundaries, we also take a look at studies done to detect regions and targets in radar products since those are, in terms of methodology, in close vicinity to stratigraphy mapping endeavours.' If this is the case, it should be claimed in the abstract, title, introduction, etc.

- The abstract was corrected.

Ln 422-424. Repetition. This should go into the motivation for IHR identification, not here.

- This is indeed a repetition and has been removed.

Ln 426. I am not sure that I would call filtering and transformation computer vision-based methods, as DL and SVM are computer vision. Consider using classical/traditional machine learning methods or something similar. The heading of the following sections should be the same as the bullet list.

- Filtering and edge detection methods are a subset of classical computer vision methods. The heading of the subsequent subsection has been modified accordingly.

Ln 487. Feature is used as geological target/IHR. Pay attention to the fact that computer vision has a different meaning. Thus, trying to be consistent in the paper.

- Thank you, this is indeed true. However, in this case we think that it is sufficiently clear that "feature" here refers to visual edges in radargrams as opposed to its usage in computer vision.

Ln 485 (Ferro and Bruzzone (2013)) I would specify that this work is on planetary data (sharad) given that most of the other works are on aircraft data (e.g., CREeSIS-MCords). Planetary data have different types of noise and radiometric characteristics than aircraft data.

- Thank you very much for this point. It has been added.

Ln 504. 'A user is required to determine the number of visible layers initially.' Manually?

- This is not clarified in the publication, it can be assumed that this is done manually, but we do not add that point to the text due to uncertainty of this matter.

Ln 508, Panton 2014. The methods are not described.

- It is mentioned that Snake technique has been used.

Ln 640. Consider adding the comparison to the paper 'The modification makes the method more appropriate for a wide range of radargrams, based on comparison with results of Crandall et al. (2012); Lee et al. (2014); Rahnemoonfar et al. (2017a).'

- Since the comparison is mentioned, we have added one remark and interested readers can be referred to the original publication.

Ln 656.' After statistical analysis, different classes are represented by pdf. This is partially true. The authors designed a set of manual features (not only statistical features) that are extracted for each pixel in the radargram and then analyzed with SVM. The work was improved in Donini, Elena, et al. "An automatic approach to map refreezing ice in radar sounder data." Image and Signal Processing for Remote Sensing XXV. Vol. 11155. SPIE, 2019.

- Thank you for pointing this out, we have modified the text.

Ln 671, 'Going away from classification of regions and ice-base and ice surface IRHs to tracing internal IRH' is too informal.

- The sentence was modified.

Ln 714. Foci -¿ focus?

- It has been corrected.

Ln 713 'In this subsection, summaries and main points of the studies that used deep learning-based methods are presented. We would like to note that although the primary foci of some of the the works e.g. Donini et al. (2021, 2022c); Garcia et al. (2021); García et al. (2023); Ghosh and Bovolo (2022a, 2023b) lie in radargram region segmentation, they are included in this review because of their methodological relevance for the overall objective.' I would change the paper's objective to methods for the automatic analysis of radargrams. Otherwise, explaining how to perform segmentation does not make sense if the paper's aim is layer detection.

- This has been explained and modified in the abstract.

Ln 755. The sentence 'Their initial stage is to remove the noise by using bilateral filtering.' Should emphasize that the method cannot handle the radar's noise characteristics and needs to apply strong preprocessing to mitigate speckle. This contrasts with the method Donini et al., and Garcia et al. used to manage the noisy properties within the network. The paper, in general, lacks this kind of critical analysis.

- The scope of this review article is giving the readers an overview of all the methods that haven been implemented for IRH tracing and radargram segmentation. The suggestions is quite detailed, and our intention is to point interested readers to the already published material so that they refer to those when they need more insights.

Ln 984. Some solutions have been proposed to overcome this (Cai et al., 2022; García et al., 2023; Moqadam et al., 2024). Also, Donini et al. propose a pre-training to set the network parameters to not random values.

- As these were examples, we had not intended to add all the publications. However, this reference was also added.

Ln 929. The section on unsupervised segmentation of radargrams lacks the reference to Donini, Elena, et al. "Unsupervised semantic segmentation of radar sounder data using contrastive learning." Image and Signal Processing for Remote Sensing XXVIII. Vol. 12267. SPIE, 2022.

- As it is not open access it is difficult to find out more about the paper. Also we could not access it through our institution's access either.